# RMNP: Row-Momentum Normalized Preconditioning for Scalable Matrix-Based Optimization

**Shenyang Deng**[* 1]  **Zhuoli Ouyang**[* 1]  **Tianyu Pang**[1]  **Zihang Liu**[2 3]  **Ruochen Jin**[1]  **Shuhua Yu**[4]  **Yaoqing Yang**[1]

## Abstract

Preconditioned adaptive methods have gained significant attention for training deep neural networks, as they capture rich curvature information of the loss landscape . The central challenge in this field lies in balancing preconditioning effectiveness with computational efficiency of implementing the preconditioner. Among recent advances, MUON stands out by using Newton-Schulz iteration to obtain preconditioned updates without explicitly constructing the preconditioning matrix. Despite its advantages, the efficiency of MUON still leaves room for further improvement. In this paper, we introduce RMNP (Row Momentum Normalized Preconditioning), an optimizer that replaces Newton-Schulz iteration with a simple row-wise($d_{\text{in}}$) $\ell_2$ normalization operation, motivated by the empirically observed diagonal block structure of the Transformer layer-wise Hessian. We empirically verified that orthogonalization and row-wise(on input dim) $\ell_2$ normalization are asymptotically equivalent in the case of the transformer. This substitution reduces the per-iteration computational complexity from $\mathcal{O}(mn \cdot \min(m,n))$ to $\mathcal{O}(mn)$ for an $m \times n$ weight matrix while maintaining comparable optimization performance. Theoretically, we establish convergence guarantees for RMNP in the non-convex setting that match recent results for MUON optimizers, achieving the minimax optimal complexity. Extensive experiments on large language model pretraining show that RMNP delivers competitive optimization performance compared with MUON while substantially reducing preconditioning wall-clock time. Our code is available at this link.

---

[*]Equal contribution [1]Dartmouth College, Hanover, NH, USA [2]International Computer Science Institute, Berkeley, CA, USA [3]University of California, Berkeley, CA, USA [4]Meta, USA. Correspondence to: Yaoqing Yang <yaoqing.yang@dartmouth.edu>.

*Proceedings of the 43rd International Conference on Machine Learning*, Seoul, South Korea. PMLR 306, 2026. Copyright 2026 by the author(s).

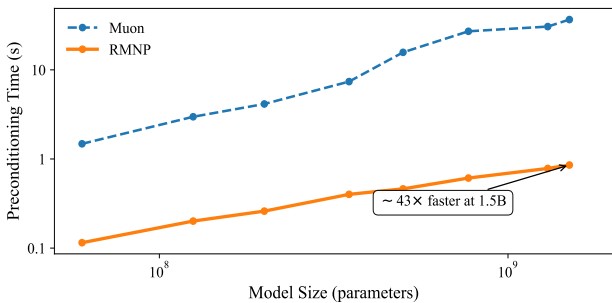

*Figure 1.* Time overhead comparison. The figure illustrates the wall-clock time for 100 computation steps for preconditioning process of RMNP versus MUON.

## 1. Introduction

Adaptive algorithms such as Duchi et al. (2011); Tieleman & Hinton (2017); Kingma & Ba (2014); Loshchilov & Hutter (2019) have achieved remarkable success in deep learning optimization. These methods employ diagonal preconditioning (Duchi et al., 2011), which scales each parameter independently based on historical gradient information. However, this diagonal structure ignores correlations among parameters, limiting the optimizer's ability to handle ill-conditioned problems with complex parameter interactions. This creates a fundamental gap between practical diagonal methods and the theoretically optimal full-matrix preconditioning.

Recent work has revisited matrix-based preconditioning to address these limitations. In particular, studies on full Gauss-Newton methods (Abreu et al., 2025) demonstrate that utilizing complete curvature information can lead to qualitatively improved convergence behavior in large language models. However, directly applying updates of the form $w_t = w_{t-1} - H_t^{-1} d_t$ remains computationally prohibitive: if the preconditioner $H_t$ is constructed using the full Hessian, the computational overhead becomes extreme and scales poorly with model size. Consequently, practical optimizer design has focused on structured approximations with first-order information to balance performance with efficiency.

Classic methods such as K-FAC (Martens & Grosse, 2015), PSGD (Li, 2017), and SHAMPOO (Gupta et al., 2018)

achieve this balance through structured matrix preconditioners that approximate curvature with lower-dimensional factors. SHAMPOO, for example, employs a Kronecker-factored preconditioner:

$$H = L \otimes R \tag{1}$$

where $L$ and $R$ are smaller matrices capturing row and column correlations, respectively. This factorization preserves essential curvature information while dramatically reducing computational demands. Subsequent works including K-BFGS (Ren et al., 2021) and ASGO (An et al., 2026) further refine this approach with sparse or low-rank updates to minimize memory overhead.

More recently, methods such as MUON (Jordan et al., 2024) have introduced an alternative perspective on matrix-based adaptivity (see Algorithm 1). Rather than explicitly forming the full preconditioner $H^{-1}$, MUON employs Newton-Schulz iterations to implicitly compute the preconditioned updates $H_t^{-1} d_t$ through matrix polynomials, enabling matrix-level adaptation without direct inversion. Subsequent refinements further improve the stability and efficiency of this approach (Tian & Parikh, 2022; Vyas et al., 2025; Si et al., 2025; Liu et al., 2026). Overall, these methods move beyond element-wise diagonal preconditioning by incorporating structured off-diagonal curvature information, aiming to achieve a **better trade-off between optimization performance and computational cost**. However, despite this conceptual advancement, the reliance on iterative matrix polynomial evaluations in MUON incurs a computational complexity of $\mathcal{O}(mn \cdot \min(m, n))$ for an $m \times n$ weight matrix, which can become a dominant bottleneck as model dimensions grow.

---

**Algorithm 1** MUON (Jordan et al., 2024)

**Require:** $\eta_t > 0, \beta \in [0, 1), W_0 \in \mathbb{R}^{m \times n}$, loss $f$,
1: $V_0 \leftarrow \mathbf{0}_{m \times n}$
2: **for** $t = 1$ to $T$ **do**
3: $\quad G_t \leftarrow \nabla f(W_t; \xi^t)$ (gradient estimator)
4: $\quad V_t \leftarrow \beta V_{t-1} + (1 - \beta) G_t$
5: $\quad D_t \leftarrow \mathrm{NS}_5(V_t)$, equal to $D_t = (V_t V_t^T)^{-\frac{1}{2}} \cdot V_t$
6: $\quad W_{t+1} \leftarrow W_t - \eta_t D_t$
7: **end for**

---

In this paper, we show that the computational complexity of MUON can be further reduced without sacrificing its matrix-level adaptivity. Specifically, motivated by recent empirical and theoretical findings on the structure of Transformer Hessians (Zhang et al., 2024; Dong et al., 2025), we introduce the Row Momentum Normalized Preconditioned Optimizer (RMNP, Algorithm 2). RMNP achieves optimization performance comparable to MUON while substantially reducing the preconditioning overhead, with a per-iteration complexity of $\mathcal{O}(mn)$. We further benchmark the

---

**Algorithm 2** RMNP

**Require:** $\eta_t > 0, \beta \in [0, 1), W_0 \in \mathbb{R}^{m \times n}$, loss $f$,
1: $V_0 \leftarrow \mathbf{0}_{m \times n}$
2: **for** $t = 1$ to $T$ **do**
3: $\quad G_t \leftarrow \nabla f(W_t; \xi^t)$ (gradient estimator)
4: $\quad V_t \leftarrow \beta V_{t-1} + (1 - \beta) G_t$
5: $\quad D_t \leftarrow \mathrm{RN}(V_t)$, equal to: $D_t = (\mathrm{diag}(V_t V_t^T))^{-\frac{1}{2}} V_t$
6: $\quad W_{t+1} \leftarrow W_t - \eta_t D_t$
7: **end for**

---

*Table 1.* Comparison of Convergence Results. $L_F, L_*$ denotes the corresponding smoothness coefficient and $\|\nabla f\|_F, \|\nabla f\|_*$ the corresponding convergence criterion.

| Work | Smooth | Conv. | Complexity |
|---|---|---|---|
| | | MUON | |
| (Chang et al., 2025) | $L_F$ | $\|\nabla f\|_*$ | $O(m^2 L \sigma^2 \Delta \epsilon^{-4})$ |
| (Kim & Oh, 2026) | $L_*$ | $\|\nabla f\|_*$ | $O(m L_* \sigma^2 \Delta \epsilon^{-4})$ |
| (Chang et al., 2025) | $L_*$ | $\|\nabla f\|_*$ | $O(m L_* \sigma^2 \Delta \epsilon^{-4})$ |
| | | RMNP | |
| Thm. 4.5 | $L_F$ | $\|\nabla f\|_F$ | $O(m^2 L_F \sigma^2 \Delta \epsilon^{-4})$ |
| Thm. 4.7 | $L_F$ | $\|\nabla f\|_{1,2}$ | $O(m^2 L_F \sigma^2 \Delta \epsilon^{-4})$ |
| Thm. 4.9 | $L_{\infty,2}$ | $\|\nabla f\|_{1,2}$ | $O(m L_{\infty,2} \sigma^2 \Delta \epsilon^{-4})$ |

wall-clock time of both optimizers under identical settings, as shown in Figure 1, demonstrating an order-of-magnitude reduction in preconditioning cost.

Mechanistically, RMNP replaces the Newton-Schulz iteration in MUON with a simple row-wise $\ell_2$ normalization. In Section 2.1, we provide a mathematical interpretation of this operation from a preconditioning perspective, showing that it corresponds to a further structured approximation of K-FAC aligned with the observed block-diagonal dominance of Transformer curvature. We also discuss how RMNP differs from related approaches and provide practical hyperparameter recommendations. Furthermore, we provide non-convex convergence guarantees for RMNP. As summarized in Table 1, our results match the best-known theoretical guarantees for MUON (Chang et al., 2025; Kim & Oh, 2026). This complexity result achieves the information-theoretic minimax optimality (Arjevani et al., 2023). For more related work, please refer to the Appendix A.

Our key contributions are summarized as follows:

- **Structure-Aware Preconditioning with Lower Time Complexity.** We propose RMNP, a matrix-based adaptive optimizer that replaces Newton–Schulz iterations in MUON with a row-wise $\ell_2$ normalization operation motivated by the observed block-diagonal dominance of Transformer curvature. This design preserves matrix-level adaptivity while reducing the per-iteration computational complexity from $\mathcal{O}(mn \cdot \min(m, n))$ to $\mathcal{O}(mn)$.

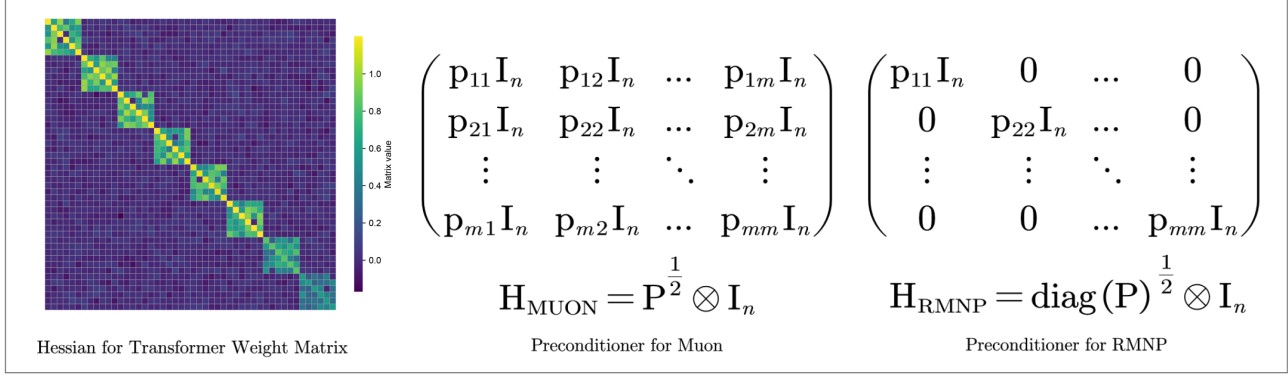

*Figure 2.* Comparison among Transformer layerwise Hessian, Preconditioner for MUON, and Preconditioner for RMNP. The figure of Transformer layerwise Hessian is conceptual, the real case can be widely found in (Zhang et al., 2025; 2024; Dong et al., 2025). $P = V_t V_t^T$, $m$ and $n$ are the number of rows and columns of the weight matrix, respectively. In this paper, the vectorization corresponding to Kronecker product is carried out by using rows for vectorization. In Section 2.2 we further verified through experiments that the MUON preconditioner has such a certain diagonal dominance property.

- **Empirical Analysis and Evaluation on Large Language Models.** We empirically validate the diagonal dominance properties of the MUON preconditioner that underlie our design hypothesis. We also conduct comparative experiments across various model architectures spanning multiple scales. Our results demonstrate that RMNP consistently matches or exceeds the optimization performance of MUON while achieving up to an order-of-magnitude reduction in preconditioning wall-clock time.

- **Non-Convex Convergence Guarantees.** We establish convergence analysis for RMNP under the non-convex smooth setting. Our theoretical results provide convergence guarantees that are on par with the current state-of-the-art theory for MUON, ensuring the robustness of our proposed method despite the reduced complexity. We also show that our results achieve information-theoretic minimax optimal complexity, despite the substantially reduced computational cost.

## 2. Method

### 2.1. RMNP Preconditioner

Recent work reveals that layer-wise Hessians of Transformers exhibit row-wise block-diagonal dominance (Zhang et al., 2024). As illustrated in Figure 2 (left), diagonal blocks—corresponding to interactions among parameters within the same row—have significantly larger magnitudes than off-diagonal blocks formed by cross-row interactions. This empirical finding is theoretically proven by Dong et al. (2025) under specific configurations.

Under this condition, the effective curvature of the loss is primarily concentrated on these diagonal blocks. By Lemma 4 in Gupta et al. (2018), the MUON preconditioner can be characterized in the following form:

$$H_{\text{MUON}} = (V_t V_t^T)^{\frac{1}{2}} \otimes I_n \tag{2}$$

where $V_t$ denotes the momentum matrix and $n$ is the column dimension, $t$ is training step index.

Building on these observations (Zhang et al., 2024), we hypothesize that the dominant curvature information resides in the row-wise diagonal blocks, while cross-row interactions contribute negligibly. This motivates approximating the preconditioner by retaining only diagonal blocks and zeroing out off-diagonal blocks, as shown in Figure 2, yielding the RMNP preconditioner:

$$H_{\text{RMNP}} = \left(\text{diag}(V_t V_t^T)\right)^{\frac{1}{2}} \otimes I_n \tag{3}$$

where $\text{diag}(\cdot)$ extracts diagonal elements to form a diagonal matrix: $[\text{diag}(M)]_{ii} = M_{ii}$ and $[\text{diag}(M)]_{ij} = 0$ for $i \neq j$. This structure preserves only the row-wise blocks because $(V_t V_t^T)_{ii}$ captures interactions within the $i$-th row of $V_t$, while the Kronecker product $\text{diag}(\cdot) \otimes I_n$ applies this scaling independently to each row.

The resulting preconditioned update $\text{diag}(V_t V_t^T)^{-\frac{1}{2}} V_t$ reduces to row-wise $\ell_2$ normalization:

$$\left[\left(\text{diag}(V_t V_t^T)\right)^{-\frac{1}{2}} V_t\right]_{i,:} = \frac{V_{t,i:}}{\sqrt{(V_t V_t^T)_{ii}}} = \frac{V_{t,i:}}{\|V_{t,i:}\|_{\ell_2}} \tag{4}$$

where $V_{t,i:}$ denotes the $i$-th row of $V_t$ and $\|V_{t,i:}\|_{\ell_2} = \sqrt{(V_t V_t^T)_{ii}}$. This dramatically reduces computational complexity compared to MUON's Newton-Schulz iteration. The above conjecture is equivalent to implying that the Gram matrix $V_t V_t^\top$ exhibits a certain diagonal dominance property. In the following subsection, we empirically verify this property of the MUON preconditioner.

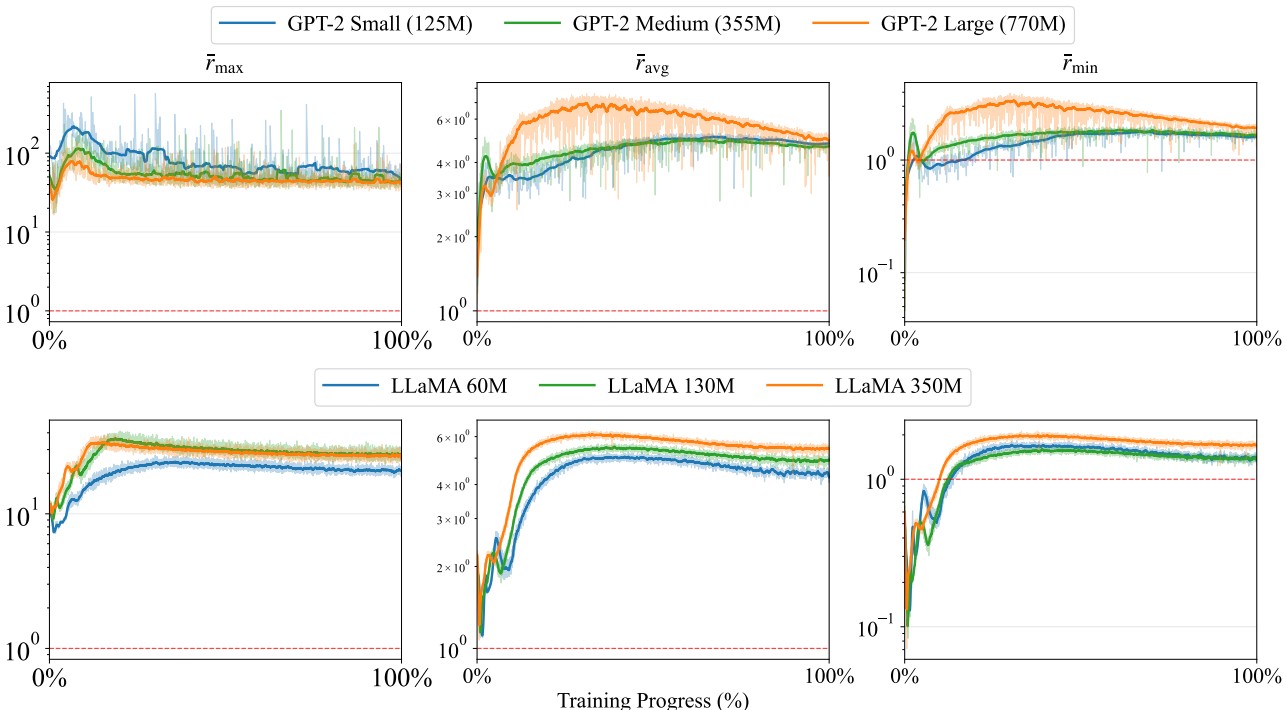

*Figure 3.* Cross-architecture, cross-scale comparison of the global diagonal dominance ratios $\bar{r}_{avg}, \bar{r}_{min}, \bar{r}_{max}$ (columns). Top row: GPT-2 Small (125M), Medium (355M), and Large (770M) on OpenWebText. Bottom row: LLaMA 60M, 130M, and 350M on C4. The x-axis is rescaled to the relative training progress (%) so that all model scales within a row align on a shared horizontal range; the y-axis is in log scale. Transparent curves: raw values; solid curves: smoothed with window size 50. Red dashed line: $y = 1$ threshold. Per-parameter ratios for three representative matrices are reported in Appendix Figures 6 (GPT-2) and 7 (LLaMA).

## 2.2. Analysis of MUON Preconditioner

To investigate the properties of the preconditioner, we analyze the Gram matrix $V_t V_t^T \in \mathbb{R}^{m \times m}$, constructed from the matrix parameter $V_t \in \mathbb{R}^{m \times n}$ at step $t$. We define a row-wise metric $r_i$ to quantify the ratio of the diagonal element to the average magnitude of off-diagonal entries in the $i$-th row:

$$
\begin{aligned}
r_i &\triangleq \frac{(V_t V_t^T)_{ii}}{\frac{1}{m-1} \sum_{j \neq i} \left| (V_t V_t^T)_{ij} \right|} \\
&= \frac{\|V_{t,i:}\|_2^2}{\frac{1}{m-1} \sum_{j \neq i} |V_{t,i:}(V_{t,j:})^T|},
\end{aligned} \tag{5}
$$

where $(V_t V_t^T)_{ij}$ denotes the entry at row $i$ and column $j$ of the Gram matrix. Based on these row-wise ratios, we introduce the following three aggregate metrics to evaluate the global diagonal dominance across all rows of the matrix. We define *average diagonal dominance ratio* ($r_{avg}$), *minimum diagonal dominance ratio* ($r_{min}$) and *maximum*

*diagonal dominance ratio* ($r_{max}$) as follows:

$$
r_{avg} = \frac{1}{m} \sum_{i=1}^{m} r_i, \tag{6}
$$

$$
r_{min} = \min_{i \in \{1, \ldots, m\}} r_i, \tag{7}
$$

$$
r_{max} = \max_{i \in \{1, \ldots, m\}} r_i. \tag{8}
$$

Regarding the interpretation, values of $r_i > 1$ indicate that the diagonal element dominates the average off-diagonal magnitude in row $i$, suggesting stronger diagonal dominance. Values approaching 1 suggest that the diagonal element is comparable to the average off-diagonal magnitude, while values significantly greater than 1 indicate that $V_t V_t^T$ closely approximates a diagonal matrix structure.

To validate our method empirically, we track these metrics across all matrix parameters during training. We visualize the evolution of the global statistics ($\bar{r}_{avg}, \bar{r}_{min}, \bar{r}_{max}$), which average the three per-parameter ratios across all matrix parameters in the network, across GPT-2 Small/Medium/Large on OpenWebText and LLaMA 60M/130M/350M on C4 in Figure 3. Per-parameter ratios $r_{avg}, r_{min}, r_{max}$ for three representative matrices at each scale are reported in Appendix Figures 6 (GPT-2) and 7 (LLaMA). The experimental setup

follows that of the main pre-training experiments; see Appendix E.1 for training hyperparameters and Appendix C for implementation details.

As shown in Figure 3, both GPT-2 and LLaMA exhibit strong diagonal dominance throughout training: across scales, $\overline{r}_{\min}$ stays comfortably above the $y = 1$ threshold, $\overline{r}_{\mathrm{avg}}$ consistently exceeds 5, and $\overline{r}_{\max}$ reaches tens. Larger models exhibit progressively stronger dominance across all three statistics, confirming that the row-wise block-diagonal dominance of the MUON preconditioner is not an isolated phenomenon but a systematic property that persists across architectures and scales. Per-parameter ratios for three representative matrices in each model (Appendix Figures 6 and 7) show the same pattern at the individual-matrix level.

**Discussion with Recent Row Normalization Optimizers.** Zhang et al. (2025) is the first work to introduce row-wise normalization into optimizer design, assigning a single learning rate per row (i.e., per output neuron) of each weight matrix to drastically cut Adam's memory while matching its performance, and Pethick et al. (2025) subsequently proposed the abstract LMO framework that unifies many modern optimizers as steepest descent under a chosen norm. Follow the LMO framework, a number of papers derive row- or column-normalized optimizers from this viewpoint. SRON (Wen et al., 2025) applies row-wise normalization to plain SGD, motivated by row-level gradient disparities in attention. SCALE (Glentis et al., 2025) shows that column-wise normalization (which is along the $d_{\mathrm{in}}$ dimension, consistent with the normalization axis of the aforementioned works) plus last-layer momentum is a minimal modification to SGD that matches Adam. SWAN (Ma et al., 2025) combines row-wise standardization with gradient whitening as a stateless preprocessing. MNGD (Scetbon et al., 2025) generalizes this via an alternating scheme enforcing multiple norms simultaneously. MANO (Gu & Xie, 2026) recasts row normalization as Riemannian optimization on a rotational Oblique manifold. MOGA (Xu et al., 2026) derives row/column normalization from mean-normalized operator norms, yielding width-independent smoothness and $\mu$P-style learning-rate transfer. NORMUON (Li et al., 2025b) extends MUON with per-neuron adaptive learning rates and a memory-efficient distributed implementation, further improving MUON's training efficiency and scalability.

**Why steepest-descent analyses cannot explain NN-specific benefits.** As illustrated in Figure 4, all of the above works expect Zhang et al. (2025) analyze their algorithms benefit through the steepest-descent lens (It mainly refers to the abstract LMO framework.), which inherently considers only the *worst-case problem for the algorithm* within a broad problem class such as nonconvex $L$-smooth, and therefore provides only a floor guarantee. While such a guar-

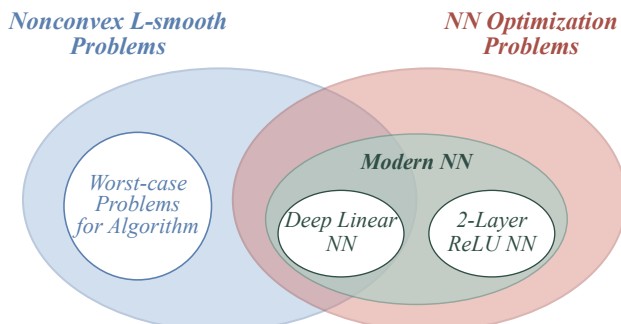

*Figure 4.* Worst-case problems for an algorithm may not capture the essential properties of NN optimization problems.

antee is meaningful in its own right, and we provide a similar result in our paper, *it cannot explain why a particular norm is specifically well-suited to neural-network optimization*: the analysis is agnostic to the actual loss landscape, so any norm choice looks equally justifiable at the worst-case level. To understand why these algorithms actually work on NNs, one has to examine the concrete problem structure itself. **Our analysis therefore departs from the steepest-descent viewpoint and starts from the curvature structure of neural networks. Motivated by recent work on the Hessian structure(Zhang et al., 2025; 2024; Dong et al., 2025) of neural networks, we verify that full orthogonalization and row $\ell_2$-normalization exhibit a high-dimensional asymptotic equivalence for Transformers.**

## 3. Main Experimental Results

In this section, we demonstrate that RMNP achieves competitive optimization performance while maintaining high computational efficiency. We first show that RMNP reduces the preconditioning computational cost by an order of magnitude compared to MUON, demonstrating its scalability advantages. We then evaluate RMNP against ADAMW and MUON, two prevalent optimizers for training large language models, on the GPT-2 and LLaMA model series. GPT-2 models are trained on OpenWebText (Gokaslan et al., 2019) and FineWeb-Edu-100B (Penedo et al., 2024), while LLaMA models are trained on C4 (Raffel et al., 2020).

### 3.1. Experimental Setup

**MUON**  Following the setup in Jordan et al. (2024); Liu et al. (2025), we employ a mixed update strategy where matrix parameters are optimized using MUON and non-matrix parameters using ADAMW. We introduce two distinct learning rate hyperparameters, $\mathrm{lr}_{\mathrm{AdamW}}$ and $\mathrm{lr}_{\mathrm{Matrix}}$, both following a cosine annealing schedule with a 10% warmup period.

**RMNP**  For RMNP, we align our experimental setup with the MUON protocol described above. We employ an almost

identical mixed update strategy, applying RMNP to matrix parameters and ADAMW to non-matrix parameters. Similarly, we utilize two learning rates, lr$_{\text{AdamW}}$ and lr$_{\text{Matrix}}$, both subject to a cosine annealing schedule with a 10% warmup. Consistent with the baseline settings, we fix the ADAMW hyperparameters ($\beta = (0.9, 0.95)$, weight decay 0.1) and exclusively tune the learning rate for the matrix optimizer, lr$_{\text{Matrix}}$, during the search process.

**ADAMW** For the ADAMW setup, we follow the standard setup in Yuan et al. (2025) for training GPT-2, and He et al. (2026) for LLaMA. We set $\beta = (0.9, 0.95)$ and weight decay 0.1, and a cosine annealing schedule with 10% warm up which consistent with the ADAMW configuration used in the mixed update strategy above.

**GPT-2 Pre-Training on OpenWebText** Experiments on GPT-2 are conducted based on the implementation of Yuan et al. (2025), using the OpenWebText dataset (Gokaslan et al., 2019) and the GPT-2 tokenizer. We pretrain three scales of GPT-2 models: small (125M parameters), medium (355M parameters), and large (770M parameters). For model configurations, we set the dropout rate to 0.0 and disable biases. Training hyperparameters are listed in Tables 5 and 8 in Appendix E.2. We also evaluate on FineWeb-Edu-100B (Penedo et al., 2024; Lozhkov et al., 2024) across four GPT-2 scales (Small, Medium, Large, and XLarge (1.5B)); see Appendix E.2 for configurations and results.

**LLaMA Pre-Training on C4** Experiments on LLaMA are conducted on the C4 dataset (Raffel et al., 2020). We pretrain four scales of LLaMA models: LLaMA-60M, LLaMA-130M, LLaMA-350M, and LLaMA-1B. Training hyperparameters are listed in Table 8 in Appendix E.2.

*Table 2.* Efficiency comparison between MUON and RMNP's preconditioning cost on GPT-2 models. Time measured over 100 steps with batch size 16 on a single RTX Pro 6000 GPU.

| Size | Time Cost (s) | | Speedup ($\times$) |
|------|------|------|------|
| | MUON | RMNP | |
| 60M | 1.480 | **0.115** | 12.9 |
| 125M | 2.975 | **0.201** | 14.8 |
| 200M | 4.140 | **0.260** | 15.9 |
| 355M | 7.380 | **0.401** | 18.4 |
| 500M | 15.720 | **0.462** | 34.0 |
| 770M | 27.070 | **0.611** | 44.3 |
| 1.3B | 30.570 | **0.783** | 39.0 |
| 1.5B | 36.650 | **0.855** | 42.9 |

### 3.2. Preconditioning Time Cost

Since RMNP and MUON primarily differ in their choice of preconditioner, where MUON applies Newton–Schulz or-

thogonalization whereas RMNP uses row normalization, we benchmark the preconditioner-operator overhead of RMNP against MUON. Specifically, we report the per-iteration time attributable to the preconditioner operator (*Step Time*) and the cumulative time over 100 iterations (*Total Time*). Experiments are run on GPT-2 models ranging from 60M to 1.5B parameters with a batch size of 16. See Appendix D.1 for detailed model configurations.

As shown in Table 2, RMNP achieves significant speedup over MUON across all model sizes. The row normalization in RMNP is approximately 13–44$\times$ faster than the Newton-Schulz orthogonalization in MUON. This result underscores RMNP's computational efficiency. More importantly, as model size grows and Newton–Schulz orthogonalization increasingly becomes the dominant bottleneck in end-to-end training throughput, RMNP's lightweight preconditioner offers a more scalable alternative, indicating strong potential for training at very large scale. For example, in Table 2, for GPT-2 60M, MUON's preconditioning cost per 100 steps is only 1.48 seconds, and RMNP provides a 12.9$\times$ speedup. However, for GPT-2 1.5B, the preconditioning cost per 100 steps increases to 36.65 seconds, while RMNP achieves a 42.9$\times$ speedup. See Appendix D for detailed results including memory usage.

### 3.3. Pretraining Performance

**RMNP consistently outperforms MUON and ADAMW in GPT-2 experiments.** As shown in Figure 5, across the Small, Medium, and Large settings, while efficiently reducing the preconditioner-operator overhead, RMNP still delivers more competitive results than both baselines in terms of evaluation perplexity: on the Small setting it improves over MUON by 0.04 and over ADAMW by 1.37; on the Medium setting the improvements are 0.07 and 1.49; and on the Large setting they are 0.24 and 0.84, respectively. This consistent pattern suggests that RMNP's efficiency gains in Table 2 do not come at the expense of optimization quality; instead, it preserves strong optimization behavior while reducing preconditioning overhead, yielding a favorable speed–accuracy trade-off across model scales under a standard large-model training protocol. Our GPT-2 experiments on OpenWebText match the results reported in Yuan et al. (2025). We conduct an extensive hyperparameter grid search for both MUON and RMNP; see Table 9 and 10 in Appendix E. Results on FineWeb-Edu-100B further confirm this trend (Appendix E.2). Per-step training and validation loss curves for all GPT-2 scales on both datasets are reported in Appendix F (Figures 10–16); the corresponding gradient clip-rate trajectories are shown in Appendix F.7. The advantage of RMNP also persists under a 2$\times$ extended training budget (Appendix E.3, Table 14).

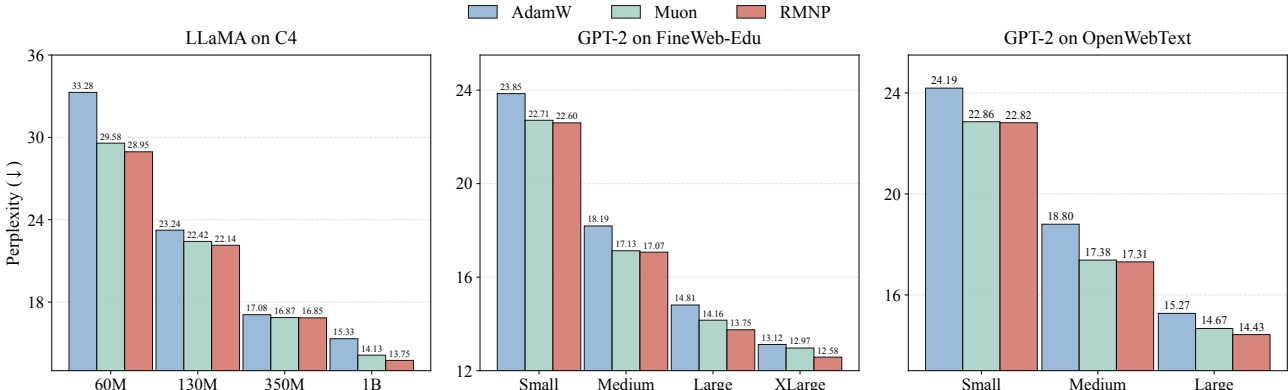

*Figure 5.* Final validation perplexity (↓) across three pretraining settings. **Left:** LLaMA on C4 – 60M trained with 1B tokens, 130M with 2B tokens, 350M with 6B tokens, and 1B with 9B tokens. **Middle:** GPT-2 on FineWeb-Edu-100B – Small (125M), Medium (355M), Large (770M), and XLarge (1.5B). **Right:** GPT-2 on OpenWebText – Small (125M) trained with 5B tokens, Medium (355M) with 10B tokens, and Large (770M) with 20B tokens. RMNP attains the lowest perplexity in every cell. Numeric values are reported in Table 17.

**RMNP consistently outperforms MUON and ADAMW in LLaMA experiments.** As shown in Figure 5, RMNP consistently achieves comparable perplexity to MUON across all model sizes, while maintaining a slight performance edge. Specifically, RMNP demonstrates modest improvements over the baseline: on the LLaMA-60M setting, it decreases perplexity by 0.63 compared to MUON and 4.33 compared to ADAMW; on the LLaMA-130M setting, the gain is 0.28 over MUON and 1.10 compared to ADAMW; and on the LLaMA-350M setting, the improvement is 0.02 over MUON. This pattern suggests that RMNP is able to fully match the optimization quality of MUON without the heavy preconditioning overhead, effectively delivering efficiency gains without sacrificing performance. It is worth noting that we perform a systematic hyperparameter grid search for both MUON and RMNP; see Table 11, 12 and 13 in Appendix E. Per-step training and validation loss curves for all four LLaMA scales are reported in Appendix F.4 (Figures 17–20). We also study the effect of applying the matrix optimizer to the LM-head and embedding parameters in Appendix E.4 (Tables 15 and 16); a final-perplexity summary across all settings is provided in Appendix F.1.

**Compatibility with distributed training.** Unlike Muon's Newton-Schulz orthogonalization that requires all-gathering the full matrix across devices, RMNP's row normalization only requires a complete row. This enables column-wise parameter sharding under PyTorch FSDP2 without cross-device synchronization, making RMNP a suitable for large-scale distributed training in PyTorch (Liang et al., 2025) where memory-efficient sharding is essential.

## 4. Non-Convex Convergence

In this section, we present the convergence analysis of our proposed method under the non-convex smooth setting. Our

setup is consistent with many existing analyses for adaptive algorithms (Chen et al., 2022; Xie et al., 2025; Li et al., 2025a; Chang et al., 2025; Kim & Oh, 2026), assuming only the smoothness of the loss function, alongside unbiased stochastic gradients and bounded second moments, as detailed in Section 4.2.

Recent work reveals that optimizers can achieve provable advantages under specific geometric structures beyond standard $\ell_2$ or Frobenius smoothness. For instance, Xie et al. (2025) discuss benefits of ADAM under $\|\cdot\|_{\ell_\infty}$-smoothness with convergence measured in $\|\cdot\|_{\ell_1}$, while Chang et al. (2025); Kim & Oh (2026) establish advantages of MUON under $\|\cdot\|_2$-smoothness with convergence measured in nuclear norm. Similarly, we identify the geometric structure under which RMNP achieves provable benefits. We establish three convergence results: under the standard $\|\cdot\|_F$-smoothness assumption, we prove convergence in both the Frobenius norm sense (Theorem 4.5) and the $\|\cdot\|_{1,2}$ norm sense (Theorem 4.7). More importantly, under the $\|\cdot\|_{\infty,2}$-smoothness assumption, we establish improved convergence guarantees in the $\|\cdot\|_{1,2}$ norm sense (Theorem 4.9), revealing that RMNP similarly benefits from its matched geometric structure.

### 4.1. Notation

Let $W \in \mathbb{R}^{m \times n}$ denote the parameter matrix, where $W_{i,:} \in \mathbb{R}^n$ denotes the $i$-th row. The matrix inner product is $\langle Z, W \rangle = \text{Tr}(Z^\top W)$. We use the Frobenius norm $\|W\|_F = \sqrt{\sum_{i,j} W_{i,j}^2}$, the mixed norm $\|W\|_{1,2} = \sum_{i=1}^m \|W_{i,:}\|_2$, and the norm $\|W\|_{\infty,2} = \max_{i=1,\dots,m} \|W_{i,:}\|_2$. These satisfy the duality $|\langle A, B \rangle| \leq \|A\|_{1,2}\|B\|_{\infty,2}$. We use $\mathbb{E}[\cdot]$ to denote the expectation and $\mathbb{E}_t[\cdot \mid \mathcal{F}_{t-1}]$ to denote the conditional expectation given $\mathcal{F}_{t-1}$. Since the input dimension is generally smaller than

the output dimension in neural networks, without loss of generality, we assume $m < n$ throughout this discussion.

## 4.2. Assumptions

**Assumption 4.1** (Lipschitz Gradient). The gradient of $f : \mathbb{R}^{m \times n} \to \mathbb{R}$ is Lipschitz continuous in one of the following norms:

**(a)** *Frobenius norm:* There exists $L_F > 0$ such that for all $W, W' \in \mathbb{R}^{m \times n}$,

$$\|\nabla f(W) - \nabla f(W')\|_F \le L_F \|W - W'\|_F.$$

**(b)** $(1, 2)$*-norm with respect to* $(\infty, 2)$*-norm:* There exists $L_{\infty,2} > 0$ such that for all $W, W' \in \mathbb{R}^{m \times n}$,

$$\|\nabla f(W) - \nabla f(W')\|_{(1,2)} \le L_{\infty,2} \|W - W'\|_{\infty,2}.$$

**Assumption 4.2** (Unbiased Gradient Estimator). For all $t$ and $W_t$,

$$\mathbb{E}_t[G_t \mid \mathcal{F}_{t-1}] = \mathbb{E}_t[\nabla f(W_t; \xi^t) \mid \mathcal{F}_{t-1}] = \nabla f(W_t).$$

**Assumption 4.3** (Bounded Gradient Variance). There exists a constant $\sigma > 0$ such that for all $t$ and $W_t$,

$$\mathbb{E}_t[\|G_t - \nabla f(W_t)\|_F^2 \mid \mathcal{F}_{t-1}] \le \frac{\sigma^2}{B},$$

where $B$ is the batch size (i.e., the number of samples used to compute $G_t$).

**Assumption 4.4** (Lower Bound). $f$ is bounded below with $f^* = \inf_W f(W)$. Define $\Delta = f(W_0) - f^*$.

## 4.3. Main Results

We now present our main theoretical results, which establish convergence guarantees for RMNP under different smoothness assumptions and convergence criteria. Our analysis reveals how the choice of matrix norms—both in the smoothness assumption and in the convergence measure—affects the sample complexity.

**Theorem 4.5** ($\|\cdot\|_F$- Lipschitz). *Under Assumptions 4.1(a), 4.2, 4.3, and 4.4, if Algorithm 2 uses constant $\eta_t = \eta$ and momentum $\beta \in [0, 1)$, then*

$$\frac{1}{T} \sum_{t=1}^{T} \mathbb{E}\left[\|\nabla f(W_t)\|_F\right]$$
$$\le \frac{\Delta}{T\eta} + (\sqrt{m} + 1)\left[\left(1 - \frac{1}{T}\right)\frac{L_F \eta \sqrt{m} \beta}{1 - \beta}\right.$$
$$\left. + \frac{\sigma}{\sqrt{B}}\sqrt{\frac{1 - \beta}{1 + \beta}}\right] + \frac{L_F \eta m}{2}. \quad (9)$$

*Remark* 4.6 (Complexity for Theorem 4.5). If we set $B = 1$, $\eta = \sqrt{\frac{(1-\beta)\Delta}{L_F m T}}$, and $1 - \beta = \min\left\{\frac{\sqrt{L_F \Delta}}{(\sqrt{m}+1)\sigma\sqrt{T}}, 1\right\}$, then the bound in (9) yields

$$\frac{1}{T} \sum_{t=1}^{T} \mathbb{E}\left[\|\nabla f(W_t)\|_F\right]$$
$$\le O\left(\sqrt[4]{\frac{m^2 L_F \Delta \sigma^2}{T}} + \sqrt{\frac{L_F m \Delta}{T}}\right.$$
$$\left. + \frac{m\sigma^2}{\sqrt{L_F \Delta T}}\right).$$

Thus, we can find an $\epsilon$-stationary point (in Frobenius norm) of $f$ with a complexity of $O(m^2 L_F \sigma^2 \Delta \epsilon^{-4})$, exhibiting an $O(m^2)$ dimension dependence.

The detail proof of Theorem 4.5 could be found in Appendix B.4. While Theorem 4.5 establishes convergence in the Frobenius norm, matrix optimization problems often involve alternative matrix norms that better capture the underlying structure. Our next result analyzes convergence in the $\|\cdot\|_{1,2}$ norm under the same Frobenius smoothness assumption, demonstrating that RMNP achieves comparable complexity guarantees across different convergence measures.

**Theorem 4.7** ($\|\cdot\|_{1,2}$-Convergence under $\|\cdot\|_F$-Lipschitz). *Under Assumptions 4.1(a), 4.2, 4.3, and 4.4, if Algorithm 2 uses constant $\eta_t = \eta$ and momentum $\beta \in [0, 1)$, then*

$$\frac{1}{T} \sum_{t=1}^{T} \mathbb{E}\left[\|\nabla f(W_t)\|_{1,2}\right]$$
$$\le \frac{\Delta}{T\eta} + 2\left[\left(1 - \frac{1}{T}\right)\frac{L_F \eta m \beta}{1 - \beta}\right.$$
$$\left. + \frac{\sqrt{m}\sigma}{\sqrt{B}}\sqrt{\frac{1 - \beta}{1 + \beta}}\right] + \frac{L_F \eta m}{2}. \quad (10)$$

*Remark* 4.8 (Complexity for Theorem 4.7). If we set $B = 1$, $\eta = \sqrt{\frac{(1-\beta)\Delta}{L_F m T}}$, and

$$1 - \beta = \min\left\{\frac{\sqrt{L_F \Delta}}{2\sqrt{m}\sigma\sqrt{T}}, 1\right\},$$

then the bound in (10) yields

$$\frac{1}{T} \sum_{t=1}^{T} \mathbb{E}\left[\|\nabla f(W_t)\|_{1,2}\right]$$
$$\le O\left(\sqrt[4]{\frac{m^2 L_F \Delta \sigma^2}{T}} + \sqrt{\frac{L_F m \Delta}{T}}\right.$$
$$\left. + \frac{m\sigma^2}{\sqrt{L_F \Delta T}}\right).$$

Thus, we can find an $\epsilon$-stationary point (in $\|\cdot\|_{1,2}$ norm) of $f$ with a complexity of $O(m^2 L_F \sigma^2 \Delta \epsilon^{-4})$, exhibiting an $O(m^2)$ dimension dependence.

The detail proof of Theorem 4.7 could be found in Appendix B.4. The preceding results demonstrate that under Frobenius smoothness, both convergence measures achieve $O(m^2)$ complexity. However, when the objective function exhibits a different geometric structure—specifically, when the gradient is Lipschitz continuous with respect to the $\|\cdot\|_{\infty,2}$ norm—RMNP's row normalization operation can exploit this structure more effectively. Our final result establishes a significantly improved complexity bound in this setting.

**Theorem 4.9** ($\|\cdot\|_{1,2}$-Lipschitz). *Under Assumptions 4.1(b), 4.2, 4.3, and 4.4, if Algorithm 2 uses constant $\eta_t = \eta$ and momentum $\beta \in [0, 1)$, then*

$$
\frac{1}{T}\sum_{t=1}^{T} \mathbb{E}\left[\|\nabla f(W_t)\|_{1,2}\right]
$$

$$
\leq \frac{\Delta}{T\eta} + 2\left[\left(1-\frac{1}{T}\right)\frac{L_{\infty,2}\eta\beta}{1-\beta}\right.
$$

$$
\left. + \frac{\sqrt{m}\sigma}{\sqrt{B}}\sqrt{\frac{1-\beta}{1+\beta}}\right] + \frac{L_{\infty,2}\eta}{2}. \tag{11}
$$

*Remark* 4.10 (Complexity for Theorem 4.9). If we set $B = 1$, $\eta = \sqrt{\frac{(1-\beta)\Delta}{L_{\infty,2}T}}$, and $1-\beta = \min\left\{\frac{\sqrt{L_{\infty,2}\Delta}}{2\sqrt{m}\sigma\sqrt{T}}, 1\right\}$, then the bound in (11) yields

$$
\frac{1}{T}\sum_{t=1}^{T} \mathbb{E}\left[\|\nabla f(W_t)\|_{1,2}\right]
$$

$$
\leq O\left(\sqrt[4]{\frac{mL_{\infty,2}\Delta\sigma^2}{T}} + \sqrt{\frac{L_{\infty,2}\Delta}{T}}\right.
$$

$$
\left. + \frac{\sqrt{m}\sigma^2}{\sqrt{L_{\infty,2}\Delta T}}\right).
$$

Thus, we can find an $\epsilon$-stationary point (in $\|\cdot\|_{1,2}$ norm) of $f$ with a complexity of $O(mL_{\infty,2}\sigma^2\Delta\epsilon^{-4})$.

### 4.4. Comparison with Related Work

We now compare our theoretical results with recent work on MUON optimizers, as summarized in Table 1. Similar to MUON, RMNP demonstrates geometry-dependent advantages: different smoothness assumptions lead to different convergence guarantees. Under Frobenius norm smoothness (Assumption 4.1(a)), both Theorem 4.5 and Theorem 4.7 achieve a sample complexity of $O(m^2 L_F \sigma^2 \Delta \epsilon^{-4})$, matching the recent results for MUON (Chang et al., 2025). More importantly, under the $\|\cdot\|_{\infty,2}$-smoothness assumption (Assumption 4.1(b)), Theorem 4.9 achieves an improved com-

plexity of $O(mL_{\infty,2}\sigma^2\Delta\epsilon^{-4})$, representing a quadratic improvement in dimension dependence from $O(m^2)$ to $O(m)$. This mirrors MUON's improvement under nuclear norm smoothness, where convergence of $\|\nabla f\|_*$ also achieves $O(m)$ complexity (Chang et al., 2025). Although the geometric structures differ—MUON exploits nuclear norm geometry while RMNP exploits $\|\cdot\|_{1,2}$ geometry—both methods achieve the same $O(m)$ dimension dependence in their respective favorable settings. This improvement stems from RMNP's row normalization operation, which naturally aligns with the row-wise structure present in the $\|\cdot\|_{1,2}$ geometry.

## 5. Conclusion

In this paper, we introduced **RMNP** (Row Momentum Normalized Preconditioning), an efficient optimizer that significantly advances preconditioned adaptive methods for deep neural network training. Motivated by the diagonal block dominance structure observed in Transformer Hessians, RMNP replaces the Newton-Schulz iteration in MUON with a simple row-wise $\ell_2$ normalization operation, reducing the per-iteration computational complexity from $\mathcal{O}(mn \cdot \min(m,n))$ to $\mathcal{O}(mn)$—an order of magnitude improvement.

Our contributions span three key dimensions. **Algorithmically**, RMNP achieves substantial efficiency gains, delivering 10-40× speedup **on the preconditioning process** over MUON across model scales from 60M to 1.5B parameters while maintaining comparable memory usage. **Empirically**, extensive experiments on GPT-2 (125M, 355M, 770M, and 1.5B on FineWeb-Edu-100B) and LLaMA (60M, 130M, 350M, and 1B) demonstrate that RMNP consistently matches or outperforms both MUON and AdamW in terms of final performance. Our empirical analysis validates the diagonal dominance property of the MUON preconditioner, providing strong support for RMNP's design principle. We also provide practical hyperparameter recommendations, showing that $\mathrm{lr}_{\mathrm{Matrix}}$ is the primary factor influencing performance. **Theoretically**, we establish rigorous convergence guarantees in the non-convex setting that match recent results for MUON optimizers. As summarized in Table 1, RMNP achieves sample complexity of $O(m^2 L_F \sigma^2 \Delta \epsilon^{-4})$ under Frobenius smoothness and an improved $O(mL_{\infty,2}\sigma^2\Delta\epsilon^{-4})$ under $\|\cdot\|_{\infty,2}$-smoothness, and both result's complexity achieving information-theoretic minimax optimality (Arjevani et al., 2023).

By effectively balancing preconditioning effectiveness with computational efficiency, RMNP provides a more scalable preconditioning approach that becomes particularly advantageous when MUON's preconditioning process emerges as a computational bottleneck in large-scale training scenarios.

## Acknowledgments

We thank our collaborators, colleagues, and funding agencies. This work is supported by the DARPA AIQ program, the U.S. Department of Energy under Award Number DE-SC0025584, the Allocation Year 2026 DOE Mission Science Award, Dartmouth College, and Lambda AI. We also thank the three ICML reviewers, the Area Chair, and Yushun Zhang for their valuable feedback and discussions on our paper. We have incorporated their suggestions to refine the work and include additional interesting findings and supporting evidence.

## Impact Statement

This paper presents work whose goal is to advance the field of Machine Learning by developing more computationally efficient optimization methods. There are many potential societal consequences of our work, none which we feel must be specifically highlighted here. The reduced computational requirements may lower barriers to training large models, with impacts that depend on downstream applications.

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

# A. Related Work

**Preconditioned Optimization Algorithms**    Preconditioned optimization methods aim to reshape the gradient by incorporating curvature information, thereby accelerating convergence in ill-conditioned problems. Early approaches such as ADAGRAD (Duchi et al., 2011) and RMSPROP (Tieleman & Hinton, 2017) employ diagonal preconditioning that adapts to the per-coordinate geometry of gradients. While computationally efficient, diagonal preconditioners fail to capture parameter correlations that naturally arise in neural network training. To address this limitation, matrix-based preconditioning methods have been developed. K-FAC (Martens & Grosse, 2015) approximates the Fisher information matrix using Kronecker-factored structure, exploiting the layer-wise organization of neural networks. PSGD (Li, 2017; 2022) introduces Lie group preconditioners that maintain geometric properties during optimization. SHAMPOO (Gupta et al., 2018) generalizes preconditioning to tensor spaces, maintaining separate preconditioners for each dimension through Kronecker factorization. Recent work has further improved upon SHAMPOO, with SOAP (Vyas et al., 2025) stabilizing it through Adam-style updates, while distributed implementations (Shi et al., 2023) enable scaling to large models. Extensions such as K-BFGS (Ren et al., 2021) and ASGO (An et al., 2026) explore sparse or low-rank updates to reduce memory overhead. More recently, MUON (Jordan et al., 2024; Liu et al., 2025) employs orthogonalization via Newton-Schulz iteration as a form of preconditioning for matrix parameters. Several variants have emerged, including ADAMUON (Si et al., 2025) which combines MUON with Adam-style adaptivity, and HTMUON (Pang et al., 2026) which generalizes to the steepest descent under a Schatten-q norm constraint. COSMOS (Liu et al., 2026) which introduces hybrid mechanisms for memory-efficient training. Studies on full Gauss-Newton methods (Abreu et al., 2025) demonstrate that complete second-order information can substantially improve convergence, motivating the search for practical approximations that balance computational cost with optimization effectiveness. Deng et al. (2026a) show that, without preconditioning, first-order methods such as SGD suffer a pathological "suspicious alignment" phenomenon under ill-conditioned Hessians with noise, where optimization becomes confined to a particular parameter subspace.

**Hessian Properties of Neural Networks**    Understanding the structure of the Hessian matrix is crucial for designing effective optimization algorithms, as the geometric properties of loss landscapes strongly influence training dynamics (Li et al., 2018). Early spectral analysis (Sagun et al., 2016; 2017) revealed that neural network Hessians exhibit a characteristic eigenvalue spectrum: a bulk of near-zero eigenvalues with a small number of isolated large outliers. Subsequent work (Ghorbani et al., 2019) observed that gradients predominantly align with these outlier eigenvectors during training. Wu et al. (2020) further demonstrated that layer-wise Hessians can be approximated using Kronecker factorization, explaining their persistent low-rank structure. Theoretical analyses (Singh et al., 2021; Liao & Mahoney, 2021) have provided rigorous explanations for these phenomena, deriving exact formulas for Hessian rank and connecting eigenvalue structure to data properties. Most relevant to our work, Zhang et al. (2024) made a significant discovery: the layer-wise Hessian of Transformers exhibits row-wise block-diagonal dominance, where diagonal blocks (corresponding to within-row parameter interactions) have significantly larger magnitudes than off-diagonal blocks (cross-row interactions). This observation has been further investigated by Dong et al. (2025), who provide theoretical characterizations of this structured dominance pattern. This row-wise block structure directly motivates our algorithm design, suggesting that row-level preconditioning may suffice to capture essential curvature information while maintaining computational efficiency. Deng et al. (2026b) how that the bulk-and-spike bifurcation of the Hessian spectrum arises from network depth rather than data imbalance, proving in deep linear networks that the ratio between dominant and bulk eigenvalues scales linearly with depth.

**Convergence Analysis of Adaptive Algorithms**    Theoretical understanding of adaptive optimization algorithms in non-convex settings has advanced significantly in recent years. For first-order adaptive methods, Chen et al. (2022) established convergence guarantees for ADAM in the non-convex setting, while Li et al. (2025a) analyzed RMSPROP and its momentum extension, proving $O(\sqrt{d}T^{1/4})$ convergence rates measured in $\ell_1$ norm. A key recent development is the recognition that different optimizers achieve provable advantages under specific geometric structures. Xie et al. (2025) demonstrate that ADAM exploits $\|\cdot\|_{\ell_\infty}$-smoothness geometry, achieving improved convergence when measured in the dual $\|\cdot\|_{\ell_1}$ norm. This geometry-dependent analysis has been extended to matrix optimization: Chang et al. (2025) and Kim & Oh (2026) establish convergence of MUON under nuclear norm smoothness, showing $O(m)$ complexity compared to the $O(m^2)$ complexity under Frobenius smoothness. These results reveal that matching the optimizer structure to the problem geometry yields substantial complexity improvements beyond what standard Euclidean or Frobenius analysis would suggest. Information-theoretic lower bounds (Arjevani et al., 2023) establish that $\epsilon^{-4}$ sample complexity is optimal for finding $\epsilon$-stationary points in the non-convex stochastic setting, providing fundamental limits for algorithm design.

# B. Proof of Theorem

## B.1. Notation

We first recall our notation here, Let $W \in \mathbb{R}^{m \times n}$ denote the parameter matrix, where $W_{i,:} \in \mathbb{R}^n$ denotes the $i$-th row. The matrix inner product is $\langle Z, W \rangle = \mathrm{Tr}(Z^\top W)$. We use the Frobenius norm $\|W\|_F = \sqrt{\sum_{i,j} W_{i,j}^2}$, the mixed norm $\|W\|_{1,2} = \sum_{i=1}^m \|W_{i,:}\|_2$, and the norm $\|W\|_{\infty,2} = \max_{i=1,\dots,m} \|W_{i,:}\|_2$. These satisfy the duality $|\langle A, B \rangle| \leq \|A\|_{1,2} \|B\|_{\infty,2}$. We use $\mathbb{E}[\cdot]$ to denote the expectation and $\mathbb{E}_t[\cdot \mid \mathcal{F}_{t-1}]$ to denote the conditional expectation given $\mathcal{F}_{t-1}$.

## B.2. Assumptions

*Assumption* (Lipschitz Gradient). The gradient of $f : \mathbb{R}^{m \times n} \to \mathbb{R}$ is Lipschitz continuous in one of the following norms:

**(a)** *Frobenius norm:* There exists a constant $L_F > 0$ such that for all $W, W' \in \mathbb{R}^{m \times n}$,

$$\|\nabla f(W) - \nabla f(W')\|_F \leq L_F \|W - W'\|_F.$$

**(b)** *Mixed norm:* There exists a constant $L_{\infty,2} > 0$ such that for all $W, W' \in \mathbb{R}^{m \times n}$,

$$\|\nabla f(W) - \nabla f(W')\|_{1,2} \leq L_{\infty,2} \|W - W'\|_{\infty,2}.$$

*Assumption* (Unbiased Estimator). For all iterations $t$ and all parameter values $W_t$,

$$\mathbb{E}_t[G_t \mid \mathcal{F}_{t-1}] = \nabla f(W_t).$$

*Assumption* (Bounded Variance). There exists a constant $\sigma > 0$ such that for all iterations $t$ and all parameter values $W_t$,

$$\mathbb{E}_t[\|G_t - \nabla f(W_t)\|_F^2 \mid \mathcal{F}_{t-1}] \leq \frac{\sigma^2}{B},$$

where $B$ is the batch size.

*Assumption* (Lower Bound). The objective function $f$ is bounded below with $f^* = \inf_{W \in \mathbb{R}^{m \times n}} f(W) > -\infty$. We define the initial optimality gap as $\Delta = f(W_0) - f^*$.

## B.3. Proof of Lemmas

**Lemma B.1.** *Let $V \in \mathbb{R}^{m \times n}$ be any matrix and define $D = RN(V)$. Then:*

1. $\|D\|_F = \sqrt{m}$,

2. $\langle V, D \rangle = \sum_{i=1}^m \|V_{i,:}\|_2 \geq \|V\|_F$.

*Proof.* By definition, $D_{i,:} = V_{i,:}/\|V_{i,:}\|_2$, so $\|D_{i,:}\|_2 = 1$ for all $i$. Thus

$$\|D\|_F^2 = \sum_{i=1}^m \|D_{i,:}\|_2^2 = m.$$

For the inner product,

$$\langle V, D \rangle = \sum_{i=1}^m \sum_{j=1}^n V_{i,j} \cdot \frac{V_{i,j}}{\|V_{i,:}\|_2}$$
$$= \sum_{i=1}^m \frac{\|V_{i,:}\|_2^2}{\|V_{i,:}\|_2}$$
$$= \sum_{i=1}^m \|V_{i,:}\|_2.$$

For the inequality, let $a_i = \|V_{i,:}\|_2 \geq 0$. Squaring both sides of $\sum_i a_i \geq \sqrt{\sum_i a_i^2}$, we need

$$\left(\sum_{i=1}^m a_i\right)^2 \geq \sum_{i=1}^m a_i^2.$$

This holds since $(\sum_i a_i)^2 = \sum_i a_i^2 + 2\sum_{i<j} a_i a_j \geq \sum_i a_i^2$. $\qquad\square$

**Lemma B.2.** *Let $V \in \mathbb{R}^{m \times n}$ be any matrix and define $D = RN(V)$. Then:*

1. $\|D\|_{\infty,2} = 1$,

2. $\langle V, D \rangle = \|V\|_{1,2}$.

*Proof.* By definition of row normalization, $D_{i,:} = V_{i,:}/\|V_{i,:}\|_2$ for all $i$, which gives $\|D_{i,:}\|_2 = 1$ for all $i$. Therefore,

$$\|D\|_{\infty,2} = \max_{i=1,\dots,m} \|D_{i,:}\|_2 = 1.$$

For the inner product, we have

$$
\begin{aligned}
\langle V, D \rangle &= \sum_{i=1}^m \sum_{j=1}^n V_{i,j} \cdot \frac{V_{i,j}}{\|V_{i,:}\|_2} \\
&= \sum_{i=1}^m \frac{\|V_{i,:}\|_2^2}{\|V_{i,:}\|_2} \\
&= \sum_{i=1}^m \|V_{i,:}\|_2 \\
&= \|V\|_{1,2}.
\end{aligned}
$$

$\qquad\square$

**Lemma B.3.** *Under Assumption 4.1(a), for any $W, W' \in \mathbb{R}^{m \times n}$,*

$$f(W') \leq f(W) + \langle \nabla f(W), W' - W \rangle + \frac{L_F}{2}\|W' - W\|_F^2.$$

**Lemma B.4.** *Under Assumption 4.1(a), for any iteration $t$,*

$$f(W_t) - f(W_{t+1}) \geq \eta \langle \nabla f(W_t), D_t \rangle - \frac{L_F \eta^2 m}{2}.$$

*Proof.* We apply Lemma B.3 with $W = W_t$ and $W' = W_{t+1} = W_t - \eta D_t$:

$$
\begin{aligned}
f(W_{t+1}) &\leq f(W_t) + \langle \nabla f(W_t), W_{t+1} - W_t \rangle + \frac{L_F}{2}\|W_{t+1} - W_t\|_F^2 \\
&= f(W_t) + \langle \nabla f(W_t), -\eta D_t \rangle + \frac{L_F}{2}\| - \eta D_t\|_F^2 \\
&= f(W_t) - \eta \langle \nabla f(W_t), D_t \rangle + \frac{L_F \eta^2}{2}\|D_t\|_F^2 \\
&= f(W_t) - \eta \langle \nabla f(W_t), D_t \rangle + \frac{L_F \eta^2}{2} \cdot m,
\end{aligned}
$$

where the last equality uses $\|D_t\|_F = \sqrt{m}$ from Lemma B.1(i). Rearranging gives

$$f(W_t) - f(W_{t+1}) \geq \eta \langle \nabla f(W_t), D_t \rangle - \frac{L_F \eta^2 m}{2}.$$

$\qquad\square$

**Lemma B.5.** *Let $E_t = V_t - \nabla f(W_t)$. Then*

$$\langle \nabla f(W_t), D_t \rangle \geq \|\nabla f(W_t)\|_F - (\sqrt{m} + 1)\|E_t\|_F.$$

*Proof.* We decompose the inner product by writing $\nabla f(W_t) = V_t - E_t$:

$$\langle \nabla f(W_t), D_t \rangle = \langle V_t - E_t, D_t \rangle$$
$$= \langle V_t, D_t \rangle - \langle E_t, D_t \rangle.$$

By Lemma B.1(ii), we have

$$\langle V_t, D_t \rangle = \sum_{i=1}^{m} \|V_{t,i,:}\|_2$$
$$\geq \|V_t\|_F.$$

For the error term, by the Cauchy-Schwarz inequality,

$$|\langle E_t, D_t \rangle| \leq \|E_t\|_F \|D_t\|_F$$
$$= \|E_t\|_F \cdot \sqrt{m},$$

where we used Lemma B.1(i).

Since $V_t = \nabla f(W_t) + E_t$, the reverse triangle inequality gives

$$\|V_t\|_F = \|\nabla f(W_t) + E_t\|_F$$
$$\geq \|\nabla f(W_t)\|_F - \|E_t\|_F.$$

Combining all inequalities:

$$\langle \nabla f(W_t), D_t \rangle = \langle V_t, D_t \rangle - \langle E_t, D_t \rangle$$
$$\geq \|V_t\|_F - |\langle E_t, D_t \rangle|$$
$$\geq \|V_t\|_F - \sqrt{m}\|E_t\|_F$$
$$\geq (\|\nabla f(W_t)\|_F - \|E_t\|_F) - \sqrt{m}\|E_t\|_F$$
$$= \|\nabla f(W_t)\|_F - (1 + \sqrt{m})\|E_t\|_F$$
$$= \|\nabla f(W_t)\|_F - (\sqrt{m} + 1)\|E_t\|_F.$$

$\square$

**Lemma B.6.** *Define the stochastic noise $\xi_t = G_t - \nabla f(W_t)$ for $t \geq 1$, which satisfies $\mathbb{E}_t[\xi_t \mid \mathcal{F}_{t-1}] = 0$ by Assumption 4.2. Then*

$$\sum_{t=1}^{T} \mathbb{E}[\|E_t\|_F] \leq (T-1)\frac{L_F \eta \sqrt{m} \beta}{1 - \beta} + T\frac{\sigma}{\sqrt{B}}\sqrt{\frac{1 - \beta}{1 + \beta}}.$$

*Proof.* From the momentum update rule $V_t = \beta V_{t-1} + (1 - \beta)G_t$ and the definition $E_t = V_t - \nabla f(W_t)$, we derive:

$$E_t = V_t - \nabla f(W_t)$$
$$= \beta V_{t-1} + (1 - \beta)G_t - \nabla f(W_t).$$

We add and subtract $\beta \nabla f(W_{t-1})$ and $(1 - \beta)\nabla f(W_t)$:

$$E_t = \beta V_{t-1} - \beta \nabla f(W_{t-1}) + \beta \nabla f(W_{t-1})$$
$$+ (1 - \beta)G_t - (1 - \beta)\nabla f(W_t) + (1 - \beta)\nabla f(W_t) - \nabla f(W_t)$$
$$= \beta(V_{t-1} - \nabla f(W_{t-1})) + \beta(\nabla f(W_{t-1}) - \nabla f(W_t))$$
$$+ (1 - \beta)(G_t - \nabla f(W_t))$$
$$= \beta E_{t-1} + \beta(\nabla f(W_{t-1}) - \nabla f(W_t)) + (1 - \beta)\xi_t.$$

Assuming $E_0 = 0$ (since $V_0 = 0$), we expand this recursion by repeated application. For $t \geq 1$, we can show by induction that

$$E_t = \sum_{j=1}^{t-1} \beta^{t-j}(1-\beta)\xi_j + \sum_{j=1}^{t-1} \beta^{t-j}(\nabla f(W_j) - \nabla f(W_{j+1})).$$

For the base case $t = 1$, we have $E_1 = \beta \cdot 0 + \beta(\nabla f(W_0) - \nabla f(W_1)) + (1-\beta)\xi_1$, which matches the formula when both sums are empty (upper limit $j = 0$). For the inductive step, assume the formula holds for $t - 1$. Then:

$$
\begin{aligned}
E_t &= \beta E_{t-1} + \beta(\nabla f(W_{t-1}) - \nabla f(W_t)) + (1-\beta)\xi_t \\
&= \beta \left[ \sum_{j=1}^{t-2} \beta^{t-1-j}(1-\beta)\xi_j + \sum_{j=1}^{t-2} \beta^{t-1-j}(\nabla f(W_j) - \nabla f(W_{j+1})) \right] \\
&\quad + \beta(\nabla f(W_{t-1}) - \nabla f(W_t)) + (1-\beta)\xi_t \\
&= \sum_{j=1}^{t-2} \beta^{t-j}(1-\beta)\xi_j + \beta(1-\beta)\xi_t \\
&\quad + \sum_{j=1}^{t-2} \beta^{t-j}(\nabla f(W_j) - \nabla f(W_{j+1})) + \beta(\nabla f(W_{t-1}) - \nabla f(W_t)) \\
&= \sum_{j=1}^{t-1} \beta^{t-j}(1-\beta)\xi_j + \sum_{j=1}^{t-1} \beta^{t-j}(\nabla f(W_j) - \nabla f(W_{j+1})).
\end{aligned}
$$

By the triangle inequality,

$$\|E_t\|_F \leq \left\| \sum_{j=1}^{t-1} \beta^{t-j}(1-\beta)\xi_j \right\|_F + \left\| \sum_{j=1}^{t-1} \beta^{t-j}(\nabla f(W_j) - \nabla f(W_{j+1})) \right\|_F.$$

For the gradient difference term, by the triangle inequality and Assumption 4.1(a),

$$\left\| \sum_{j=1}^{t-1} \beta^{t-j} (\nabla f(W_j) - \nabla f(W_{j+1})) \right\|_F$$

$$\leq \sum_{j=1}^{t-1} \beta^{t-j} \| \nabla f(W_j) - \nabla f(W_{j+1}) \|_F$$

$$\leq \sum_{j=1}^{t-1} \beta^{t-j} L_F \| W_j - W_{j+1} \|_F$$

$$= \sum_{j=1}^{t-1} \beta^{t-j} L_F \| W_j - (W_j - \eta D_j) \|_F$$

$$= \sum_{j=1}^{t-1} \beta^{t-j} L_F \eta \| D_j \|_F$$

$$= \sum_{j=1}^{t-1} \beta^{t-j} L_F \eta \sqrt{m}$$

$$= L_F \eta \sqrt{m} \sum_{j=1}^{t-1} \beta^{t-j}$$

$$= L_F \eta \sqrt{m} \sum_{k=1}^{t-1} \beta^{k}$$

$$= L_F \eta \sqrt{m} \cdot \beta \frac{1 - \beta^{t-1}}{1 - \beta}$$

$$\leq L_F \eta \sqrt{m} \cdot \frac{\beta}{1 - \beta}.$$

Summing over $t = 1, \ldots, T$ and changing the order of summation:

$$\sum_{t=1}^{T} \left\| \sum_{j=1}^{t-1} \beta^{t-j} (\nabla f(W_j) - \nabla f(W_{j+1})) \right\|_F$$

$$\leq L_F \eta \sqrt{m} \sum_{t=1}^{T} \sum_{j=1}^{t-1} \beta^{t-j}$$

$$= L_F \eta \sqrt{m} \sum_{j=1}^{T-1} \sum_{t=j+1}^{T} \beta^{t-j}$$

$$= L_F \eta \sqrt{m} \sum_{j=1}^{T-1} \sum_{k=1}^{T-j} \beta^{k}$$

$$= L_F \eta \sqrt{m} \sum_{j=1}^{T-1} \beta \frac{1 - \beta^{T-j}}{1 - \beta}$$

$$\leq L_F \eta \sqrt{m} \sum_{j=1}^{T-1} \frac{\beta}{1 - \beta}$$

$$= (T - 1) \frac{L_F \eta \sqrt{m} \beta}{1 - \beta}.$$

For the noise term, since $\mathbb{E}_j[\xi_j \mid \mathcal{F}_{j-1}] = 0$ and the noises are conditionally independent,

$$
\mathbb{E}\left[\left\|\sum_{j=1}^{t-1} \beta^{t-j}(1-\beta)\xi_j\right\|_F^2\right]
$$

$$
= \mathbb{E}\left[\left\langle \sum_{j=1}^{t-1} \beta^{t-j}(1-\beta)\xi_j, \sum_{k=1}^{t-1} \beta^{t-k}(1-\beta)\xi_k \right\rangle\right]
$$

$$
= \sum_{j=1}^{t-1}\sum_{k=1}^{t-1} \beta^{t-j}\beta^{t-k}(1-\beta)^2 \mathbb{E}[\langle \xi_j, \xi_k \rangle]
$$

$$
= \sum_{j=1}^{t-1} \beta^{2(t-j)}(1-\beta)^2 \mathbb{E}[\|\xi_j\|_F^2]
$$

$$
\leq \sum_{j=1}^{t-1} \beta^{2(t-j)}(1-\beta)^2 \cdot \frac{\sigma^2}{B}
$$

$$
= \frac{\sigma^2}{B}(1-\beta)^2 \sum_{j=1}^{t-1} \beta^{2(t-j)}
$$

$$
= \frac{\sigma^2}{B}(1-\beta)^2 \sum_{k=1}^{t-1} \beta^{2k}
$$

$$
\leq \frac{\sigma^2}{B}(1-\beta)^2 \sum_{k=0}^{\infty} \beta^{2k}
$$

$$
= \frac{\sigma^2}{B}(1-\beta)^2 \cdot \frac{1}{1-\beta^2}
$$

$$
= \frac{\sigma^2}{B}(1-\beta)^2 \cdot \frac{1}{(1-\beta)(1+\beta)}
$$

$$
= \frac{\sigma^2}{B} \cdot \frac{1-\beta}{1+\beta}.
$$

By Jensen's inequality,

$$
\mathbb{E}\left[\left\|\sum_{j=1}^{t-1} \beta^{t-j}(1-\beta)\xi_j\right\|_F\right] \leq \sqrt{\mathbb{E}\left[\left\|\sum_{j=1}^{t-1} \beta^{t-j}(1-\beta)\xi_j\right\|_F^2\right]} \leq \frac{\sigma}{\sqrt{B}}\sqrt{\frac{1-\beta}{1+\beta}}.
$$

Summing over $t = 1, \ldots, T$:

$$
\sum_{t=1}^{T} \mathbb{E}\left[\left\|\sum_{j=1}^{t-1} \beta^{t-j}(1-\beta)\xi_j\right\|_F\right] \leq T\frac{\sigma}{\sqrt{B}}\sqrt{\frac{1-\beta}{1+\beta}}.
$$

Combining both bounds:

$$
\sum_{t=1}^{T} \mathbb{E}[\|E_t\|_F] \leq (T-1)\frac{L_F\eta\sqrt{m}\beta}{1-\beta} + T\frac{\sigma}{\sqrt{B}}\sqrt{\frac{1-\beta}{1+\beta}}.
$$

$\square$

**Lemma B.7.** *Under Assumption 4.1(b), for any iteration t,*

$$
f(W_t) - f(W_{t+1}) \geq \eta\langle \nabla f(W_t), D_t \rangle - \frac{L_{\infty,2}\eta^2}{2}.
$$

*Proof.* We apply Lemma B.3 with $W = W_t$ and $W' = W_{t+1} = W_t - \eta D_t$. However, we need to be careful as Lemma B.3 is stated in terms of Frobenius norm. For the $\|\cdot\|_{\infty,2}$ case, we use the fundamental theorem of calculus directly:

$$
\begin{aligned}
f(W_{t+1}) - f(W_t) &= \int_0^1 \langle \nabla f(W_t + s(W_{t+1} - W_t)), W_{t+1} - W_t \rangle ds \\
&= \int_0^1 \langle \nabla f(W_t - s\eta D_t), -\eta D_t \rangle ds \\
&= -\eta \langle \nabla f(W_t), D_t \rangle \\
&\quad - \eta \int_0^1 \langle \nabla f(W_t - s\eta D_t) - \nabla f(W_t), D_t \rangle ds.
\end{aligned}
$$

For the second integral, by the duality $|\langle A, B \rangle| \le \|A\|_{1,2}\|B\|_{\infty,2}$ and Assumption 4.1(b),

$$
\begin{aligned}
&|\langle \nabla f(W_t - s\eta D_t) - \nabla f(W_t), D_t \rangle| \\
&\le \|\nabla f(W_t - s\eta D_t) - \nabla f(W_t)\|_{1,2}\|D_t\|_{\infty,2} \\
&\le L_{\infty,2}\|W_t - s\eta D_t - W_t\|_{\infty,2}\|D_t\|_{\infty,2} \\
&= L_{\infty,2} \cdot s\eta\|D_t\|_{\infty,2} \cdot \|D_t\|_{\infty,2} \\
&= L_{\infty,2} s\eta\|D_t\|_{\infty,2}^2 \\
&= L_{\infty,2} s\eta \cdot 1,
\end{aligned}
$$

where we used Lemma B.2(i).

Therefore,

$$
\begin{aligned}
\left| \int_0^1 \langle \nabla f(W_t - s\eta D_t) - \nabla f(W_t), D_t \rangle ds \right| &\le \int_0^1 L_{\infty,2} s\eta \, ds \\
&= L_{\infty,2}\eta \int_0^1 s \, ds \\
&= L_{\infty,2}\eta \cdot \frac{1}{2} \\
&= \frac{L_{\infty,2}\eta}{2}.
\end{aligned}
$$

Combining these results:

$$
f(W_{t+1}) - f(W_t) \ge -\eta \langle \nabla f(W_t), D_t \rangle - \frac{L_{\infty,2}\eta^2}{2},
$$

which rearranges to the desired inequality. $\qquad\square$

**Lemma B.8.** *Let $E_t = V_t - \nabla f(W_t)$. Then*

$$
\langle \nabla f(W_t), D_t \rangle \ge \|\nabla f(W_t)\|_{1,2} - 2\|E_t\|_{1,2}.
$$

*Proof.* We decompose the inner product by writing $\nabla f(W_t) = V_t - E_t$:

$$
\begin{aligned}
\langle \nabla f(W_t), D_t \rangle &= \langle V_t - E_t, D_t \rangle \\
&= \langle V_t, D_t \rangle - \langle E_t, D_t \rangle.
\end{aligned}
$$

By Lemma B.2(ii),

$$
\langle V_t, D_t \rangle = \|V_t\|_{1,2}.
$$

For the error term, by the duality between $\|\cdot\|_{1,2}$ and $\|\cdot\|_{\infty,2}$,

$$
\begin{aligned}
|\langle E_t, D_t \rangle| &\le \|E_t\|_{1,2}\|D_t\|_{\infty,2} \\
&= \|E_t\|_{1,2} \cdot 1,
\end{aligned}
$$

where we used Lemma B.2(i).

Since $V_t = \nabla f(W_t) + E_t$, the triangle inequality gives

$$
\begin{aligned}
\|V_t\|_{1,2} &= \|\nabla f(W_t) + E_t\|_{1,2} \\
&\geq \|\nabla f(W_t)\|_{1,2} - \|E_t\|_{1,2}.
\end{aligned}
$$

Combining all inequalities:

$$
\begin{aligned}
\langle \nabla f(W_t), D_t \rangle &= \langle V_t, D_t \rangle - \langle E_t, D_t \rangle \\
&\geq \|V_t\|_{1,2} - |\langle E_t, D_t \rangle| \\
&\geq \|V_t\|_{1,2} - \|E_t\|_{1,2} \\
&\geq (\|\nabla f(W_t)\|_{1,2} - \|E_t\|_{1,2}) - \|E_t\|_{1,2} \\
&= \|\nabla f(W_t)\|_{1,2} - 2\|E_t\|_{1,2}.
\end{aligned}
$$

$\square$

**Lemma B.9.** *Define the stochastic noise $\xi_t = G_t - \nabla f(W_t)$ for $t \geq 1$, which satisfies $\mathbb{E}_t[\xi_t \mid \mathcal{F}_{t-1}] = 0$ by Assumption 4.2. Then*

$$
\sum_{t=1}^{T} \mathbb{E}[\|E_t\|_{1,2}] \leq (T-1)\frac{L_{\infty,2}\eta\beta}{1-\beta} + T\frac{\sqrt{m}\sigma}{\sqrt{B}}\sqrt{\frac{1-\beta}{1+\beta}}.
$$

*Proof.* As in the proof of Lemma B.6, we have the recursion

$$
E_t = \beta E_{t-1} + \beta(\nabla f(W_{t-1}) - \nabla f(W_t)) + (1-\beta)\xi_t,
$$

which expands to

$$
E_t = \sum_{j=1}^{t-1} \beta^{t-j}(1-\beta)\xi_j + \sum_{j=1}^{t-1} \beta^{t-j}(\nabla f(W_j) - \nabla f(W_{j+1})).
$$

By the triangle inequality,

$$
\|E_t\|_{1,2} \leq \left\|\sum_{j=1}^{t-1} \beta^{t-j}(1-\beta)\xi_j\right\|_{1,2} + \left\|\sum_{j=1}^{t-1} \beta^{t-j}(\nabla f(W_j) - \nabla f(W_{j+1}))\right\|_{1,2}.
$$

For the gradient difference term, by the triangle inequality and Assumption 4.1(b),

$$\left\| \sum_{j=1}^{t-1} \beta^{t-j} (\nabla f(W_j) - \nabla f(W_{j+1})) \right\|_{1,2}$$

$$\leq \sum_{j=1}^{t-1} \beta^{t-j} \| \nabla f(W_j) - \nabla f(W_{j+1}) \|_{1,2}$$

$$\leq \sum_{j=1}^{t-1} \beta^{t-j} L_{\infty,2} \| W_j - W_{j+1} \|_{\infty,2}$$

$$= \sum_{j=1}^{t-1} \beta^{t-j} L_{\infty,2} \| W_j - (W_j - \eta D_j) \|_{\infty,2}$$

$$= \sum_{j=1}^{t-1} \beta^{t-j} L_{\infty,2} \eta \| D_j \|_{\infty,2}$$

$$= \sum_{j=1}^{t-1} \beta^{t-j} L_{\infty,2} \eta$$

$$= L_{\infty,2} \eta \sum_{j=1}^{t-1} \beta^{t-j}$$

$$= L_{\infty,2} \eta \sum_{k=1}^{t-1} \beta^{k}$$

$$= L_{\infty,2} \eta \cdot \beta \frac{1 - \beta^{t-1}}{1 - \beta}$$

$$\leq L_{\infty,2} \eta \cdot \frac{\beta}{1 - \beta}.$$

Summing over $t = 1, \ldots, T$ and changing the order of summation:

$$\sum_{t=1}^{T} \left\| \sum_{j=1}^{t-1} \beta^{t-j} (\nabla f(W_j) - \nabla f(W_{j+1})) \right\|_{1,2}$$

$$\leq L_{\infty,2} \eta \sum_{t=1}^{T} \sum_{j=1}^{t-1} \beta^{t-j}$$

$$= L_{\infty,2} \eta \sum_{j=1}^{T-1} \sum_{t=j+1}^{T} \beta^{t-j}$$

$$= L_{\infty,2} \eta \sum_{j=1}^{T-1} \sum_{k=1}^{T-j} \beta^{k}$$

$$= L_{\infty,2} \eta \sum_{j=1}^{T-1} \beta \frac{1 - \beta^{T-j}}{1 - \beta}$$

$$\leq L_{\infty,2} \eta \sum_{j=1}^{T-1} \frac{\beta}{1 - \beta}$$

$$= (T - 1) \frac{L_{\infty,2} \eta \beta}{1 - \beta}.$$

For the noise term, we use the fact that $\| \cdot \|_{1,2} \leq \sqrt{m} \| \cdot \|_F$ by Cauchy-Schwarz. Specifically, for any matrix $A$,

$$\|A\|_{1,2} = \sum_{i=1}^{m} \|A_{i,:}\|_2$$

$$\leq \sqrt{m} \sqrt{\sum_{i=1}^{m} \|A_{i,:}\|_2^2}$$

$$= \sqrt{m} \|A\|_F.$$

Therefore,

$$\left\| \sum_{j=1}^{t-1} \beta^{t-j}(1-\beta)\xi_j \right\|_{1,2} \leq \sqrt{m} \left\| \sum_{j=1}^{t-1} \beta^{t-j}(1-\beta)\xi_j \right\|_F.$$

By Cauchy-Schwarz inequality (for expectations) and Jensen's inequality,

$$\mathbb{E}\left[ \left\| \sum_{j=1}^{t-1} \beta^{t-j}(1-\beta)\xi_j \right\|_{1,2} \right]$$

$$\leq \sqrt{m}\, \mathbb{E}\left[ \left\| \sum_{j=1}^{t-1} \beta^{t-j}(1-\beta)\xi_j \right\|_F \right]$$

$$\leq \sqrt{m} \sqrt{ \mathbb{E}\left[ \left\| \sum_{j=1}^{t-1} \beta^{t-j}(1-\beta)\xi_j \right\|_F^2 \right] }.$$

From the proof of Lemma B.6, we know that

$$\mathbb{E}\left[ \left\| \sum_{j=1}^{t-1} \beta^{t-j}(1-\beta)\xi_j \right\|_F^2 \right] \leq \frac{\sigma^2}{B} \cdot \frac{1-\beta}{1+\beta}.$$

Therefore,

$$\mathbb{E}\left[ \left\| \sum_{j=1}^{t-1} \beta^{t-j}(1-\beta)\xi_j \right\|_{1,2} \right] \leq \sqrt{m} \cdot \frac{\sigma}{\sqrt{B}} \sqrt{\frac{1-\beta}{1+\beta}}.$$

Summing over $t = 1, \ldots, T$:

$$\sum_{t=1}^{T} \mathbb{E}\left[ \left\| \sum_{j=1}^{t-1} \beta^{t-j}(1-\beta)\xi_j \right\|_{1,2} \right] \leq T \frac{\sqrt{m}\sigma}{\sqrt{B}} \sqrt{\frac{1-\beta}{1+\beta}}.$$

Combining both bounds:

$$\sum_{t=1}^{T} \mathbb{E}[\|E_t\|_{1,2}] \leq (T-1)\frac{L_{\infty,2}\eta\beta}{1-\beta} + T\frac{\sqrt{m}\sigma}{\sqrt{B}} \sqrt{\frac{1-\beta}{1+\beta}}.$$

$\square$

**Lemma B.10.** *Under Assumption 4.1(a) (Frobenius smoothness), and under the same noise conditions as Lemma B.6, we have*

$$\sum_{t=1}^{T} \mathbb{E}[\|E_t\|_{1,2}] \leq (T-1)\frac{L_F \eta m \beta}{1-\beta} + T\frac{\sqrt{m}\sigma}{\sqrt{B}}\sqrt{\frac{1-\beta}{1+\beta}}.$$

*Proof.* As in the proof of Lemma B.6, we have the recursion

$$E_t = \beta E_{t-1} + \beta(\nabla f(W_{t-1}) - \nabla f(W_t)) + (1-\beta)\xi_t,$$

which expands to

$$E_t = \sum_{j=1}^{t-1} \beta^{t-j}(1-\beta)\xi_j + \sum_{j=1}^{t-1} \beta^{t-j}(\nabla f(W_j) - \nabla f(W_{j+1})).$$

By the triangle inequality,

$$\|E_t\|_{1,2} \leq \left\|\sum_{j=1}^{t-1} \beta^{t-j}(1-\beta)\xi_j\right\|_{1,2} + \left\|\sum_{j=1}^{t-1} \beta^{t-j}(\nabla f(W_j) - \nabla f(W_{j+1}))\right\|_{1,2}.$$

For the gradient difference term, we first use $\|\cdot\|_{1,2} \leq \sqrt{m}\|\cdot\|_F$, then the triangle inequality and Assumption 4.1(a):

$$\left\|\sum_{j=1}^{t-1} \beta^{t-j}(\nabla f(W_j) - \nabla f(W_{j+1}))\right\|_{1,2}$$

$$\leq \sqrt{m}\left\|\sum_{j=1}^{t-1} \beta^{t-j}(\nabla f(W_j) - \nabla f(W_{j+1}))\right\|_F$$

$$\leq \sqrt{m}\sum_{j=1}^{t-1} \beta^{t-j}\|\nabla f(W_j) - \nabla f(W_{j+1})\|_F$$

$$\leq \sqrt{m}\sum_{j=1}^{t-1} \beta^{t-j}L_F\|W_j - W_{j+1}\|_F$$

$$= \sqrt{m}\sum_{j=1}^{t-1} \beta^{t-j}L_F\eta\|D_j\|_F$$

$$= \sqrt{m}\sum_{j=1}^{t-1} \beta^{t-j}L_F\eta\sqrt{m}$$

$$= L_F\eta m\sum_{j=1}^{t-1} \beta^{t-j}$$

$$= L_F\eta m\sum_{k=1}^{t-1} \beta^{k}$$

$$= L_F\eta m \cdot \beta\frac{1-\beta^{t-1}}{1-\beta}$$

$$\leq L_F\eta m \cdot \frac{\beta}{1-\beta}.$$

Summing over $t = 1, \ldots, T$ and changing the order of summation:

$$\sum_{t=1}^{T} \left\| \sum_{j=1}^{t-1} \beta^{t-j}(\nabla f(W_j) - \nabla f(W_{j+1})) \right\|_{1,2}$$

$$\leq L_F \eta m \sum_{t=1}^{T} \sum_{j=1}^{t-1} \beta^{t-j}$$

$$= L_F \eta m \sum_{j=1}^{T-1} \sum_{t=j+1}^{T} \beta^{t-j}$$

$$= L_F \eta m \sum_{j=1}^{T-1} \sum_{k=1}^{T-j} \beta^{k}$$

$$= L_F \eta m \sum_{j=1}^{T-1} \beta \frac{1 - \beta^{T-j}}{1 - \beta}$$

$$\leq L_F \eta m \sum_{j=1}^{T-1} \frac{\beta}{1 - \beta}$$

$$= (T-1) \frac{L_F \eta m \beta}{1 - \beta}.$$

For the noise term, the analysis is identical to Lemma B.9. We use $\| \cdot \|_{1,2} \leq \sqrt{m} \| \cdot \|_F$ and the result from Lemma B.6:

$$\mathbb{E}\left[ \left\| \sum_{j=1}^{t-1} \beta^{t-j}(1-\beta)\xi_j \right\|_{1,2} \right] \leq \sqrt{m} \cdot \frac{\sigma}{\sqrt{B}} \sqrt{\frac{1-\beta}{1+\beta}}.$$

Summing over $t = 1, \ldots, T$:

$$\sum_{t=1}^{T} \mathbb{E}\left[ \left\| \sum_{j=1}^{t-1} \beta^{t-j}(1-\beta)\xi_j \right\|_{1,2} \right] \leq T \frac{\sqrt{m}\sigma}{\sqrt{B}} \sqrt{\frac{1-\beta}{1+\beta}}.$$

Combining both bounds:

$$\sum_{t=1}^{T} \mathbb{E}[\|E_t\|_{1,2}] \leq (T-1) \frac{L_F \eta m \beta}{1 - \beta} + T \frac{\sqrt{m}\sigma}{\sqrt{B}} \sqrt{\frac{1-\beta}{1+\beta}}.$$

$\square$

## B.4. Proof of Theorem

*Theorem.* 4.5 Under Assumptions 4.1(a), 4.2, 4.3, and 4.4, if Algorithm 2 uses constant step size $\eta_t = \eta$ for all $t$ and momentum parameter $\beta \in [0, 1)$, then

$$\frac{1}{T} \sum_{t=1}^{T} \mathbb{E}\left[ \|\nabla f(W_t)\|_F \right]$$

$$\leq \frac{\Delta}{T\eta} + (\sqrt{m} + 1) \left[ \left( 1 - \frac{1}{T} \right) \frac{L_F \eta \sqrt{m} \beta}{1 - \beta} \right.$$

$$\left. + \frac{\sigma}{\sqrt{B}} \sqrt{\frac{1-\beta}{1+\beta}} \right] + \frac{L_F \eta m}{2}.$$

*Proof.* We sum the descent inequality from Lemma B.4 over all iterations $t = 1, \ldots, T$:

$$\sum_{t=1}^{T} [f(W_t) - f(W_{t+1})] \geq \sum_{t=1}^{T} \left[ \eta \langle \nabla f(W_t), D_t \rangle - \frac{L_F \eta^2 m}{2} \right]$$

$$= \eta \sum_{t=1}^{T} \langle \nabla f(W_t), D_t \rangle - \sum_{t=1}^{T} \frac{L_F \eta^2 m}{2}$$

$$= \eta \sum_{t=1}^{T} \langle \nabla f(W_t), D_t \rangle - \frac{T L_F \eta^2 m}{2}.$$

The left-hand side is a telescoping sum:

$$\sum_{t=1}^{T} [f(W_t) - f(W_{t+1})] = [f(W_1) - f(W_2)] + [f(W_2) - f(W_3)] + \cdots + [f(W_T) - f(W_{T+1})]$$

$$= f(W_1) - f(W_{T+1}).$$

However, we need to account for the initial iteration. From the algorithm, $W_1$ is obtained from $W_0$ via $W_1 = W_0 - \eta D_0$. Including this, the telescoping sum gives:

$$\sum_{t=0}^{T-1} [f(W_t) - f(W_{t+1})] = f(W_0) - f(W_T).$$

For consistency with our indexing where we sum from $t = 1$ to $T$, we have:

$$\sum_{t=1}^{T} [f(W_t) - f(W_{t+1})] = f(W_1) - f(W_{T+1}).$$

To include the initial step, we note that

$$f(W_0) - f(W_1) \geq \eta \langle \nabla f(W_0), D_0 \rangle - \frac{L_F \eta^2 m}{2}.$$

For simplicity, we proceed with the standard formulation where we analyze iterations $t = 1, \ldots, T$ starting from $W_0$:

$$f(W_0) - f(W_T) \geq \eta \sum_{t=1}^{T} \langle \nabla f(W_t), D_t \rangle - \frac{T L_F \eta^2 m}{2}.$$

We now apply Lemma B.5 to bound the inner product from below. For each $t$, we have:

$$\langle \nabla f(W_t), D_t \rangle \geq \|\nabla f(W_t)\|_F - (\sqrt{m} + 1)\|E_t\|_F.$$

Summing over $t = 1, \ldots, T$:

$$\sum_{t=1}^{T} \langle \nabla f(W_t), D_t \rangle \geq \sum_{t=1}^{T} \left[ \|\nabla f(W_t)\|_F - (\sqrt{m} + 1)\|E_t\|_F \right]$$

$$= \sum_{t=1}^{T} \|\nabla f(W_t)\|_F - (\sqrt{m} + 1) \sum_{t=1}^{T} \|E_t\|_F.$$

Substituting this into our previous inequality:

$$f(W_0) - f(W_T) \geq \eta \left[ \sum_{t=1}^{T} \|\nabla f(W_t)\|_F - (\sqrt{m} + 1) \sum_{t=1}^{T} \|E_t\|_F \right] - \frac{TL_F\eta^2 m}{2}$$

$$= \eta \sum_{t=1}^{T} \|\nabla f(W_t)\|_F - \eta(\sqrt{m} + 1) \sum_{t=1}^{T} \|E_t\|_F - \frac{TL_F\eta^2 m}{2}.$$

Rearranging to isolate the gradient norm sum:

$$\eta \sum_{t=1}^{T} \|\nabla f(W_t)\|_F \leq f(W_0) - f(W_T) + \eta(\sqrt{m} + 1) \sum_{t=1}^{T} \|E_t\|_F + \frac{TL_F\eta^2 m}{2}.$$

Since $f(W_T) \geq f^* = \inf_W f(W)$ by definition, we have $f(W_0) - f(W_T) \leq f(W_0) - f^* = \Delta$. Thus:

$$\eta \sum_{t=1}^{T} \|\nabla f(W_t)\|_F \leq \Delta + \eta(\sqrt{m} + 1) \sum_{t=1}^{T} \|E_t\|_F + \frac{TL_F\eta^2 m}{2}.$$

Dividing both sides by $\eta$:

$$\sum_{t=1}^{T} \|\nabla f(W_t)\|_F \leq \frac{\Delta}{\eta} + (\sqrt{m} + 1) \sum_{t=1}^{T} \|E_t\|_F + \frac{TL_F\eta m}{2}.$$

Taking the expectation of both sides:

$$\sum_{t=1}^{T} \mathbb{E}[\|\nabla f(W_t)\|_F] \leq \frac{\Delta}{\eta} + (\sqrt{m} + 1) \sum_{t=1}^{T} \mathbb{E}[\|E_t\|_F] + \frac{TL_F\eta m}{2}.$$

We now apply Lemma B.6 to bound the error accumulation:

$$\sum_{t=1}^{T} \mathbb{E}[\|E_t\|_F] \leq (T-1)\frac{L_F\eta\sqrt{m}\beta}{1-\beta} + T\frac{\sigma}{\sqrt{B}}\sqrt{\frac{1-\beta}{1+\beta}}.$$

Substituting this bound:

$$\sum_{t=1}^{T} \mathbb{E}[\|\nabla f(W_t)\|_F] \leq \frac{\Delta}{\eta} + (\sqrt{m} + 1)\left[ (T-1)\frac{L_F\eta\sqrt{m}\beta}{1-\beta} + T\frac{\sigma}{\sqrt{B}}\sqrt{\frac{1-\beta}{1+\beta}} \right]$$

$$+ \frac{TL_F\eta m}{2}.$$

Dividing both sides by $T$:

$$\frac{1}{T}\sum_{t=1}^{T} \mathbb{E}[\|\nabla f(W_t)\|_F] \leq \frac{\Delta}{T\eta} + (\sqrt{m} + 1)\left[ \frac{T-1}{T} \cdot \frac{L_F\eta\sqrt{m}\beta}{1-\beta} \right.$$

$$\left. + \frac{\sigma}{\sqrt{B}}\sqrt{\frac{1-\beta}{1+\beta}} \right] + \frac{L_F\eta m}{2}.$$

Since $\frac{T-1}{T} = 1 - \frac{1}{T}$, we obtain:

$$\frac{1}{T}\sum_{t=1}^{T}\mathbb{E}[\|\nabla f(W_t)\|_F] \leq \frac{\Delta}{T\eta} + (\sqrt{m}+1)\left[\left(1 - \frac{1}{T}\right)\frac{L_F\eta\sqrt{m}\beta}{1-\beta}\right.$$

$$\left. + \frac{\sigma}{\sqrt{B}}\sqrt{\frac{1-\beta}{1+\beta}}\right] + \frac{L_F\eta m}{2}.$$

This completes the proof. $\qquad\square$

*Theorem.* 4.9 Under Assumptions 4.1(b), 4.2, 4.3, and 4.4, if Algorithm 2 uses constant step size $\eta_t = \eta$ for all $t$ and momentum parameter $\beta \in [0,1)$, then

$$\frac{1}{T}\sum_{t=1}^{T}\mathbb{E}\left[\|\nabla f(W_t)\|_{1,2}\right]$$

$$\leq \frac{\Delta}{T\eta} + 2\left[\left(1 - \frac{1}{T}\right)\frac{L_{\infty,2}\eta\beta}{1-\beta}\right.$$

$$\left. + \frac{\sqrt{m}\sigma}{\sqrt{B}}\sqrt{\frac{1-\beta}{1+\beta}}\right] + \frac{L_{\infty,2}\eta}{2}.$$

*Proof.* We sum the descent inequality from Lemma B.7 over all iterations $t = 1, \ldots, T$:

$$\sum_{t=1}^{T}[f(W_t) - f(W_{t+1})] \geq \sum_{t=1}^{T}\left[\eta\langle\nabla f(W_t), D_t\rangle - \frac{L_{\infty,2}\eta^2}{2}\right]$$

$$= \eta\sum_{t=1}^{T}\langle\nabla f(W_t), D_t\rangle - \sum_{t=1}^{T}\frac{L_{\infty,2}\eta^2}{2}$$

$$= \eta\sum_{t=1}^{T}\langle\nabla f(W_t), D_t\rangle - \frac{TL_{\infty,2}\eta^2}{2}.$$

The left-hand side is the telescoping sum $f(W_0) - f(W_T)$ (using the same argument as in the proof of Theorem B.4). Thus:

$$f(W_0) - f(W_T) \geq \eta\sum_{t=1}^{T}\langle\nabla f(W_t), D_t\rangle - \frac{TL_{\infty,2}\eta^2}{2}.$$

We now apply Lemma B.8 to bound the inner product from below. For each $t$, we have:

$$\langle\nabla f(W_t), D_t\rangle \geq \|\nabla f(W_t)\|_{1,2} - 2\|E_t\|_{1,2}.$$

Summing over $t = 1, \ldots, T$:

$$\sum_{t=1}^{T}\langle\nabla f(W_t), D_t\rangle \geq \sum_{t=1}^{T}[\|\nabla f(W_t)\|_{1,2} - 2\|E_t\|_{1,2}]$$

$$= \sum_{t=1}^{T}\|\nabla f(W_t)\|_{1,2} - 2\sum_{t=1}^{T}\|E_t\|_{1,2}.$$

Substituting this into our previous inequality:

$$f(W_0) - f(W_T) \geq \eta \left[ \sum_{t=1}^{T} \|\nabla f(W_t)\|_{1,2} - 2 \sum_{t=1}^{T} \|E_t\|_{1,2} \right] - \frac{T L_{\infty,2} \eta^2}{2}$$

$$= \eta \sum_{t=1}^{T} \|\nabla f(W_t)\|_{1,2} - 2\eta \sum_{t=1}^{T} \|E_t\|_{1,2} - \frac{T L_{\infty,2} \eta^2}{2}.$$

Rearranging to isolate the gradient norm sum:

$$\eta \sum_{t=1}^{T} \|\nabla f(W_t)\|_{1,2} \leq f(W_0) - f(W_T) + 2\eta \sum_{t=1}^{T} \|E_t\|_{1,2} + \frac{T L_{\infty,2} \eta^2}{2}.$$

Since $f(W_T) \geq f^*$, we have $f(W_0) - f(W_T) \leq \Delta$. Thus:

$$\eta \sum_{t=1}^{T} \|\nabla f(W_t)\|_{1,2} \leq \Delta + 2\eta \sum_{t=1}^{T} \|E_t\|_{1,2} + \frac{T L_{\infty,2} \eta^2}{2}.$$

Dividing both sides by $\eta$:

$$\sum_{t=1}^{T} \|\nabla f(W_t)\|_{1,2} \leq \frac{\Delta}{\eta} + 2 \sum_{t=1}^{T} \|E_t\|_{1,2} + \frac{T L_{\infty,2} \eta}{2}.$$

Taking the expectation of both sides:

$$\sum_{t=1}^{T} \mathbb{E}[\|\nabla f(W_t)\|_{1,2}] \leq \frac{\Delta}{\eta} + 2 \sum_{t=1}^{T} \mathbb{E}[\|E_t\|_{1,2}] + \frac{T L_{\infty,2} \eta}{2}.$$

We now apply Lemma B.9 to bound the error accumulation:

$$\sum_{t=1}^{T} \mathbb{E}[\|E_t\|_{1,2}] \leq (T-1) \frac{L_{\infty,2} \eta \beta}{1 - \beta} + T \frac{\sqrt{m}\sigma}{\sqrt{B}} \sqrt{\frac{1-\beta}{1+\beta}}.$$

Substituting this bound:

$$\sum_{t=1}^{T} \mathbb{E}[\|\nabla f(W_t)\|_{1,2}] \leq \frac{\Delta}{\eta} + 2 \left[ (T-1) \frac{L_{\infty,2} \eta \beta}{1 - \beta} + T \frac{\sqrt{m}\sigma}{\sqrt{B}} \sqrt{\frac{1-\beta}{1+\beta}} \right]$$

$$+ \frac{T L_{\infty,2} \eta}{2}.$$

Dividing both sides by $T$:

$$\frac{1}{T} \sum_{t=1}^{T} \mathbb{E}[\|\nabla f(W_t)\|_{1,2}] \leq \frac{\Delta}{T\eta} + 2 \left[ \frac{T-1}{T} \cdot \frac{L_{\infty,2} \eta \beta}{1 - \beta} \right.$$

$$\left. + \frac{\sqrt{m}\sigma}{\sqrt{B}} \sqrt{\frac{1-\beta}{1+\beta}} \right] + \frac{L_{\infty,2} \eta}{2}.$$

Since $\frac{T-1}{T} = 1 - \frac{1}{T}$, we obtain:

$$\frac{1}{T} \sum_{t=1}^{T} \mathbb{E}[\|\nabla f(W_t)\|_{1,2}] \leq \frac{\Delta}{T\eta} + 2\left[\left(1 - \frac{1}{T}\right) \frac{L_{\infty,2}\eta\beta}{1-\beta}\right.$$
$$\left. + \frac{\sqrt{m}\sigma}{\sqrt{B}} \sqrt{\frac{1-\beta}{1+\beta}}\right] + \frac{L_{\infty,2}\eta}{2}.$$

This completes the proof. $\square$

*Theorem.* 4.7 Under Assumptions 4.1(a), 4.2, 4.3, and 4.4, if Algorithm 2 uses constant step size $\eta_t = \eta$ for all $t$ and momentum parameter $\beta \in [0,1)$, then

$$\frac{1}{T} \sum_{t=1}^{T} \mathbb{E}\left[\|\nabla f(W_t)\|_{1,2}\right]$$
$$\leq \frac{\Delta}{T\eta} + 2\left[\left(1 - \frac{1}{T}\right) \frac{L_F\eta m\beta}{1-\beta}\right.$$
$$\left. + \frac{\sqrt{m}\sigma}{\sqrt{B}} \sqrt{\frac{1-\beta}{1+\beta}}\right] + \frac{L_F\eta m}{2}.$$

*Proof.* We sum the descent inequality from Lemma B.4 over all iterations $t = 1, \ldots, T$:

$$\sum_{t=1}^{T}[f(W_t) - f(W_{t+1})] \geq \sum_{t=1}^{T}\left[\eta\langle\nabla f(W_t), D_t\rangle - \frac{L_F\eta^2 m}{2}\right]$$
$$= \eta\sum_{t=1}^{T}\langle\nabla f(W_t), D_t\rangle - \sum_{t=1}^{T}\frac{L_F\eta^2 m}{2}$$
$$= \eta\sum_{t=1}^{T}\langle\nabla f(W_t), D_t\rangle - \frac{T L_F\eta^2 m}{2}.$$

The left-hand side is the telescoping sum $f(W_0) - f(W_T)$ (using the same argument as in the proof of Theorem B.4). Thus:

$$f(W_0) - f(W_T) \geq \eta\sum_{t=1}^{T}\langle\nabla f(W_t), D_t\rangle - \frac{T L_F\eta^2 m}{2}.$$

We now apply Lemma B.8 to bound the inner product from below. For each $t$, we have:

$$\langle\nabla f(W_t), D_t\rangle \geq \|\nabla f(W_t)\|_{1,2} - 2\|E_t\|_{1,2}.$$

Summing over $t = 1, \ldots, T$:

$$\sum_{t=1}^{T}\langle\nabla f(W_t), D_t\rangle \geq \sum_{t=1}^{T}[\|\nabla f(W_t)\|_{1,2} - 2\|E_t\|_{1,2}]$$
$$= \sum_{t=1}^{T}\|\nabla f(W_t)\|_{1,2} - 2\sum_{t=1}^{T}\|E_t\|_{1,2}.$$

Substituting this into our previous inequality:

$$f(W_0) - f(W_T) \geq \eta \left[ \sum_{t=1}^{T} \|\nabla f(W_t)\|_{1,2} - 2 \sum_{t=1}^{T} \|E_t\|_{1,2} \right] - \frac{T L_F \eta^2 m}{2}$$

$$= \eta \sum_{t=1}^{T} \|\nabla f(W_t)\|_{1,2} - 2\eta \sum_{t=1}^{T} \|E_t\|_{1,2} - \frac{T L_F \eta^2 m}{2}.$$

Rearranging to isolate the gradient norm sum:

$$\eta \sum_{t=1}^{T} \|\nabla f(W_t)\|_{1,2} \leq f(W_0) - f(W_T) + 2\eta \sum_{t=1}^{T} \|E_t\|_{1,2} + \frac{T L_F \eta^2 m}{2}.$$

Since $f(W_T) \geq f^*$, we have $f(W_0) - f(W_T) \leq \Delta$. Thus:

$$\eta \sum_{t=1}^{T} \|\nabla f(W_t)\|_{1,2} \leq \Delta + 2\eta \sum_{t=1}^{T} \|E_t\|_{1,2} + \frac{T L_F \eta^2 m}{2}.$$

Dividing both sides by $\eta$:

$$\sum_{t=1}^{T} \|\nabla f(W_t)\|_{1,2} \leq \frac{\Delta}{\eta} + 2 \sum_{t=1}^{T} \|E_t\|_{1,2} + \frac{T L_F \eta m}{2}.$$

Taking the expectation of both sides:

$$\sum_{t=1}^{T} \mathbb{E}[\|\nabla f(W_t)\|_{1,2}] \leq \frac{\Delta}{\eta} + 2 \sum_{t=1}^{T} \mathbb{E}[\|E_t\|_{1,2}] + \frac{T L_F \eta m}{2}.$$

We now apply Lemma B.10 to bound the error accumulation:

$$\sum_{t=1}^{T} \mathbb{E}[\|E_t\|_{1,2}] \leq (T-1)\frac{L_F \eta m \beta}{1-\beta} + T \frac{\sqrt{m}\sigma}{\sqrt{B}} \sqrt{\frac{1-\beta}{1+\beta}}.$$

Substituting this bound:

$$\sum_{t=1}^{T} \mathbb{E}[\|\nabla f(W_t)\|_{1,2}] \leq \frac{\Delta}{\eta} + 2 \left[ (T-1)\frac{L_F \eta m \beta}{1-\beta} + T \frac{\sqrt{m}\sigma}{\sqrt{B}} \sqrt{\frac{1-\beta}{1+\beta}} \right]$$

$$+ \frac{T L_F \eta m}{2}.$$

Dividing both sides by $T$:

$$\frac{1}{T} \sum_{t=1}^{T} \mathbb{E}[\|\nabla f(W_t)\|_{1,2}] \leq \frac{\Delta}{T\eta} + 2 \left[ \frac{T-1}{T} \cdot \frac{L_F \eta m \beta}{1-\beta} \right.$$

$$\left. + \frac{\sqrt{m}\sigma}{\sqrt{B}} \sqrt{\frac{1-\beta}{1+\beta}} \right] + \frac{L_F \eta m}{2}.$$

Since $\frac{T-1}{T} = 1 - \frac{1}{T}$, we obtain:

$$\frac{1}{T} \sum_{t=1}^{T} \mathbb{E}[\|\nabla f(W_t)\|_{1,2}] \leq \frac{\Delta}{T\eta} + 2\Bigg[ \left(1 - \frac{1}{T}\right) \frac{L_F \eta m \beta}{1 - \beta}$$
$$+ \frac{\sqrt{m}\sigma}{\sqrt{B}} \sqrt{\frac{1-\beta}{1+\beta}} \Bigg] + \frac{L_F \eta m}{2}.$$

This completes the proof. $\square$

## C. Analysis of MUON Preconditioner

This section provides implementation details for the diagonal dominance analysis presented in Section 2.2.

**Metric Computation**    For each matrix parameter $V_t \in \mathbb{R}^{m \times n}$ in the network, we compute the diagonal dominance metrics as follows:

1. **Gram Matrix Computation:** We first compute the Gram matrix $G = V_t V_t^T \in \mathbb{R}^{m \times m}$.

2. **Row-wise Ratio Calculation:** For each row $i \in \{1, \ldots, m\}$, we compute the ratio $r_i$ between the diagonal element and the average magnitude of off-diagonal elements:

$$r_i = \frac{G_{ii}}{\frac{1}{m-1} \sum_{j \neq i} |G_{ij}|} \tag{12}$$

   where $G_{ii} = \|V_{t,i:}\|_2^2$ is the squared norm of the $i$-th row of $V_t$.

3. **Per-Parameter Aggregation:** For each matrix parameter, we aggregate the row-wise ratios into three statistics:

$$r_{\mathrm{avg}} = \frac{1}{m} \sum_{i=1}^{m} r_i, \tag{13}$$

$$r_{\mathrm{min}} = \min_{i \in \{1, \ldots, m\}} r_i, \tag{14}$$

$$r_{\mathrm{max}} = \max_{i \in \{1, \ldots, m\}} r_i. \tag{15}$$

4. **Global Aggregation:** The global statistics $\bar{r}_{\mathrm{avg}}$, $\bar{r}_{\mathrm{min}}$, and $\bar{r}_{\mathrm{max}}$ are computed by averaging the corresponding per-parameter metrics across all $K$ matrix parameters in the network:

$$\bar{r}_{\mathrm{avg}} = \frac{1}{K} \sum_{k=1}^{K} r_{\mathrm{avg}}^{(k)}, \tag{16}$$

$$\bar{r}_{\mathrm{min}} = \frac{1}{K} \sum_{k=1}^{K} r_{\mathrm{min}}^{(k)}, \tag{17}$$

$$\bar{r}_{\mathrm{max}} = \frac{1}{K} \sum_{k=1}^{K} r_{\mathrm{max}}^{(k)}, \tag{18}$$

   where the superscript $(k)$ denotes the metric for the $k$-th matrix parameter.

**Logging Configuration**    We use Weights & Biases (wandb) for metric tracking. The diagonal dominance ratios are computed and logged at every training step. The metrics are computed within the optimizer's `step()` function, immediately after the momentum update and before the Newton-Schulz orthogonalization. In distributed training settings, the per-parameter metrics are computed locally on each GPU (parameters are distributed across GPUs), and the global statistics are synchronized via `all_reduce` operations.

**Model and Training Configuration**    We conduct the analysis on both GPT-2 and LLaMA model families to align with the main pre-training setting. For GPT-2, we analyze Small (125M), Medium (355M), and Large (770M) on OpenWebText; for LLaMA, we analyze 60M, 130M, 350M, and 1B on C4. Model scales, training steps, warm-up schedules, sequence length, and batch size follow the settings in Section D.2 (Appendix E.2). In particular, GPT-2 uses 10K/20K/40K steps with sequence length 1024 and batch size 480, while LLaMA uses 10K/20K/60K/90K steps with sequence length 256 and batch size 512. Optimization hyperparameters follow Appendix E.1; specifically, we use MUON with momentum $0.95$, weight decay $0.1$, and Newton-Schulz iteration steps of $5$.

**Visualization**    In all dominance figures of this appendix, transparent curves represent the raw logged values, while the solid curves are smoothed using simple moving average with a window size of 50. The red dashed line at $y = 1$ serves as a reference threshold—values above this line indicate that the diagonal elements dominate over the average off-diagonal magnitude, confirming diagonal dominance of the Gram matrix. Per-parameter $r_{\mathrm{avg}}, r_{\mathrm{min}}, r_{\mathrm{max}}$ for three representative matrix parameters of GPT-2 and LLaMA are shown in Figures 6 and 7; the cross-scale, cross-architecture comparison of the global ratios $\bar{r}_{\mathrm{avg}}, \bar{r}_{\mathrm{min}}, \bar{r}_{\mathrm{max}}$ is shown in the main-body Figure 3. Per-parameter ratios for the two largest models—GPT-2 XLarge (1.5B) and LLaMA 1B—are reported in Figure 8.

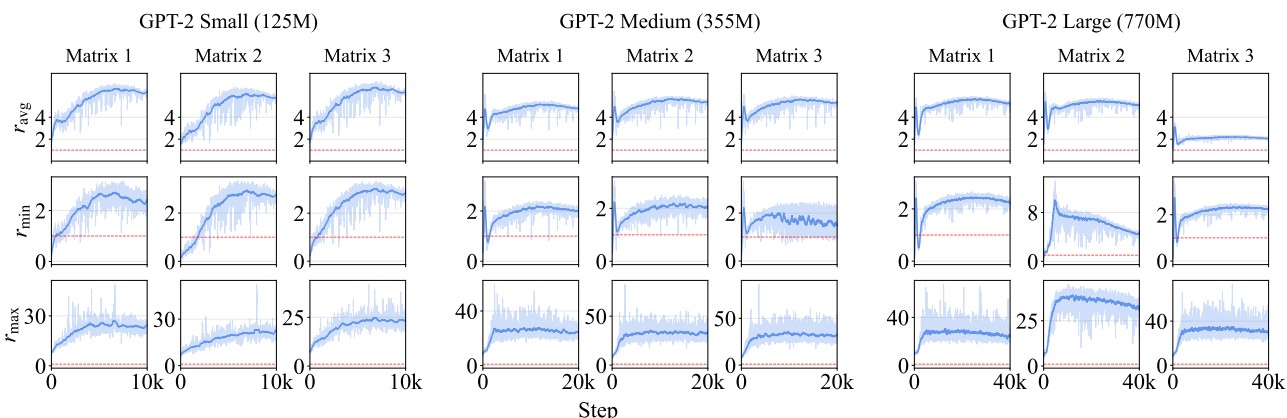

*Figure 6.* Per-parameter diagonal dominance ratios $r_{\mathrm{avg}}, r_{\mathrm{min}}, r_{\mathrm{max}}$ (rows) for three representative matrix parameters (columns) during GPT-2 Small (125M), GPT-2 Medium (355M), and GPT-2 Large (770M) pre-training. Transparent curves: raw values; solid curves: smoothed with window size 50. Red dashed line: $y = 1$ threshold.

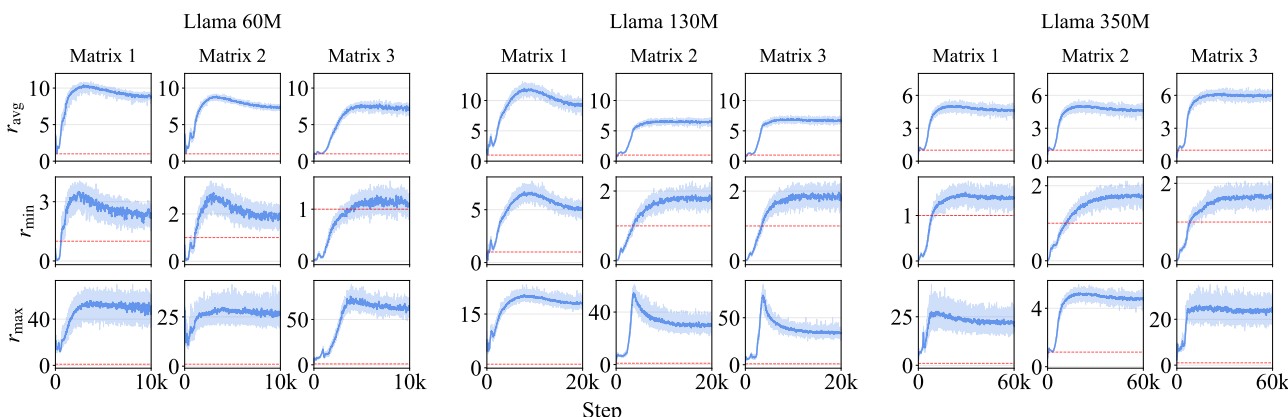

*Figure 7.* Per-parameter diagonal dominance ratios $r_{\mathrm{avg}}, r_{\mathrm{min}}, r_{\mathrm{max}}$ (rows) for three representative matrix parameters (columns) during LLaMA 60M, LLaMA 130M, and LLaMA 350M pre-training. Transparent curves: raw values; solid curves: smoothed with window size 50. Red dashed line: $y = 1$ threshold.

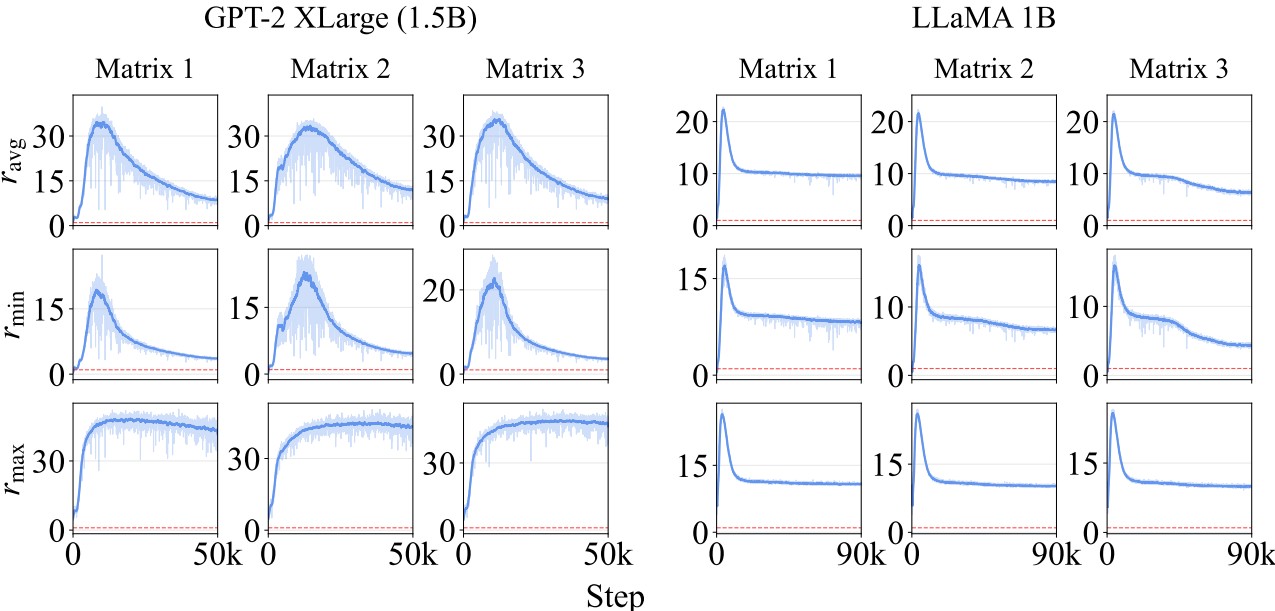

*Figure 8.* Per-parameter diagonal dominance ratios $r_{\text{avg}}$, $r_{\text{min}}$, $r_{\text{max}}$ (rows) for three representative matrix parameters (columns) on the largest scales evaluated in this paper: GPT-2 XLarge (1.5B) on FineWeb-Edu-100B (left block) and LLaMA 1B on C4 (right block). Transparent curves: raw values; solid curves: smoothed with window size 50. Red dashed line: $y = 1$ threshold. All metrics remain comfortably above the threshold throughout training, confirming that the row-wise block-diagonal dominance of the MUON preconditioner persists at the largest scales we evaluate.

## D. Preconditioning Process Wall-Clock Time

This section provides detailed efficiency measurements for the preconditioning time cost analysis presented in Section 3.2. As shown in Table 3, RMNP achieves significant speedups over MUON across all model sizes while maintaining identical memory usage. Specifically, RMNP reduces the preconditioner computation time by approximately $13\times$ to $43\times$, demonstrating its scalability advantage for large-scale training.

*Table 3.* Efficiency comparison between MUON (Newton-Schulz orthogonalization) and RMNP (row normalization) on GPT-2 models. Time and memory usage are measured over 100 steps with batch size 16 on one single RTX Pro 6000 GPU.

| Size | Time Cost (s) | | Memory (MB) | |
|---|---|---|---|---|
| | MUON | RMNP | MUON | RMNP |
| 60M | 1.480 | **0.115** | 7804 | 7804 |
| 125M | 2.975 | **0.201** | 11797 | 11797 |
| 200M | 4.140 | **0.260** | 15352 | 15352 |
| 355M | 7.380 | **0.401** | 23225 | 23225 |
| 500M | 15.720 | **0.462** | 30011 | 30011 |
| 770M | 27.070 | **0.611** | 41508 | 41508 |
| 1.3B | 30.570 | **0.783** | 61043 | 61043 |
| 1.5B | 36.650 | **0.855** | 69465 | 69465 |

### D.1. Model Configuration for Preconditioning Time Cost

Table 4 presents the detailed model configurations used for measuring preconditioning time cost in Section 3.2.

*Table 4.* GPT-2 Model Configurations for Preconditioning Time Cost

| Model | Params | Layers | Heads | $d_{\text{model}}$ |
|---|---|---|---|---|
| GPT-2 60M | 60M | 6 | 10 | 640 |
| GPT-2 Small | 125M | 12 | 12 | 768 |
| GPT-2 200M | 200M | 16 | 14 | 896 |
| GPT-2 Medium | 355M | 24 | 16 | 1024 |
| GPT-2 500M | 500M | 28 | 18 | 1152 |
| GPT-2 Large | 770M | 36 | 20 | 1280 |
| GPT-2 1.3B | 1.3B | 44 | 24 | 1536 |
| GPT-2 XL | 1.5B | 48 | 25 | 1600 |

## E. Hyperparameter Search for Pretraining Performance

This section provides detailed hyperparameter search results for the pretraining experiments described in Section 3.3. We perform a systematic hyperparameter grid search for both RMNP and MUON across GPT-2 (Small and Medium) and LLaMA (60M, 130M, and 350M) models. Following the standard MUON training protocol, RMNP is integrated with ADAMW, with the learning rate decoupled into $\text{lr}_{\text{AdamW}}$ and $\text{lr}_{\text{Matrix}}$. We fix $\text{lr}_{\text{AdamW}}$ and vary $\text{lr}_{\text{Matrix}}$ to evaluate its impact on convergence. For all LLaMA RMNP runs (60M / 130M / 350M / 1B), we further adopt a shared-LR convention $\text{lr}_{\text{AdamW}} = \text{lr}_{\text{Matrix}}$, i.e., the matrix LR reported in Tables 11–12 and 13 is also used as the ADAMW LR for the non-matrix parameters in those rows; this differs from the GPT-2 protocol, where $\text{lr}_{\text{AdamW}}$ is held fixed independently of $\text{lr}_{\text{Matrix}}$. The results are summarized in Tables 9 and 10 for GPT-2, and Tables 11, 12, and 13 for LLaMA. Due to compute constraints, we did not perform a full LR sweep for LLaMA-1B; instead, we use a fixed configuration: ADAMW with $\text{lr} = 6 \times 10^{-4}$, MUON with $\text{lr}_{\text{AdamW}} = 6 \times 10^{-4}$ and $\text{lr}_{\text{Matrix}} = 5 \times 10^{-3}$, and RMNP with $\text{lr}_{\text{AdamW}} = \text{lr}_{\text{Matrix}} = 5 \times 10^{-3}$, all with weight decay 0.1 and $\beta = (0.9, 0.95)$. All values reported are evaluation perplexity (lower is better). We also present GPT-2 experiments on FineWeb-Edu-100B (Penedo et al., 2024); see Tables 6, 7, and 17 in Appendix E.2.

### E.1. Hyperparameter Settings

This section provides detailed hyperparameter settings for the experiments described in Section 3.1.

**MUON**  For MUON, we set the momentum to 0.95 and weight decay to 0.1. Following Jordan et al. (2024), we apply an RMS scaling coefficient to the learning rate:

$$\eta = \text{lr}_{\text{Matrix}} \cdot \max\left(1, \sqrt{\frac{m}{n}}\right), \tag{19}$$

where $m$ and $n$ denote the number of rows and columns of the parameter matrix, respectively. During hyperparameter search, we exclusively tune $\text{lr}_{\text{Matrix}}$. For ADAMW, we set $\text{lr}_{\text{AdamW}} = 0.003$, $0.0015$, and $0.001$ for GPT-2 Small, medium, and large models, respectively.

**RMNP**  To ensure a fair comparison, we adopt the same RMS scaling as MUON:

$$\eta = \text{lr}_{\text{Matrix}} \cdot \max\left(1, \sqrt{\frac{m}{n}}\right), \tag{20}$$

as well as identical hyperparameters for ADAMW.

For GPT-2 experiments, the matrix optimizer is applied to all matrix parameters, including the LM head and token-embedding layers. For LLaMA experiments, the LM head and token-embedding parameters are handled by ADAMW in the main results (Tables 11, 12, and 13); an ablation on this choice is provided in Appendix E.4.

### E.2. Model Configurations

In this section, we present the model configurations and hyperparameters for GPT-2 (Table 5), GPT-2 on FineWeb-Edu-100B (Table 6), and LLaMA (Table 8). All GPT-2 models are trained with a maximum sequence length of 1024 and a batch size

of 480. All LLaMA models are trained with a maximum sequence length of 256 and a batch size of 512. GPT-2 Small and medium models are trained in parallel on 4 NVIDIA RTX Pro 6000 GPUs, while GPT-2 large models are trained on a single NVIDIA Blackwell B200 Tensor Core GPU. LLaMA-60M and LLaMA-130M are trained in parallel on 2 NVIDIA L40. LLaMA-350M is trained in parallel on 4 NVIDIA RTX Pro 6000. LLaMA-1B is trained in parallel on 8 NVIDIA GPUs. For FineWeb-Edu-100B (Penedo et al., 2024)[1], the GPT-2 Small, Medium, and Large configurations are identical to the OpenWebText setup, while the GPT-2 XLarge (1.5B) model is trained only on FineWeb-Edu-100B (no OpenWebText counterpart). Optimizer hyperparameters are listed in Table 7, and evaluation results in Figure 9 and Table 17.

*Table 5.* GPT-2 Model Configurations and specified hyperparameters for OpenWebText experiments.

| Model | Params | Layer | Heads | $d_{\text{emb}}$ | Steps | Warm-up | Token Count | Batch Size | LR schedule |
|---|---|---|---|---|---|---|---|---|---|
| GPT-2 Small | 125M | 12 | 12 | 768 | 10K | 1K | 5B | 480 | Cosine |
| GPT-2 Medium | 355M | 24 | 16 | 1024 | 20K | 2K | 10B | 480 | Cosine |
| GPT-2 Large | 770M | 36 | 20 | 1280 | 40K | 4K | 20B | 480 | Cosine |

*Table 6.* GPT-2 Model Configurations and specified hyperparameters for FineWeb-Edu-100B experiments.

| Model | Params | Layer | Heads | $d_{\text{emb}}$ | Steps | Warm-up | Token Count | Batch Size | LR schedule |
|---|---|---|---|---|---|---|---|---|---|
| GPT-2 Small | 125M | 12 | 12 | 768 | 10K | 1K | 5B | 480 | Cosine |
| GPT-2 Medium | 355M | 24 | 16 | 1024 | 20K | 2K | 10B | 480 | Cosine |
| GPT-2 Large | 770M | 36 | 20 | 1280 | 40K | 4K | 20B | 480 | Cosine |
| GPT-2 XLarge | 1.5B | 48 | 25 | 1600 | 50K | 5K | 25B | 480 | Cosine |

*Table 7.* Optimizer hyperparameters for GPT-2 pre-training on FineWeb-Edu-100B.

| Model | Optimizer | $\text{lr}_{\text{AdamW}}$ | $\text{lr}_{\text{Matrix}}$ | Weight Decay | $\beta$ | Schedule |
|---|---|---|---|---|---|---|
| | ADAMW | $6 \times 10^{-4}$ | — | 0.1 | (0.9, 0.95) | Cosine |
| Small (125M) | MUON | $3 \times 10^{-3}$ | $2 \times 10^{-2}$ | 0.1 | (0.9, 0.95) | Cosine |
| | RMNP | $3 \times 10^{-3}$ | $2 \times 10^{-2}$ | 0.1 | (0.9, 0.95) | Cosine |
| | ADAMW | $3 \times 10^{-4}$ | — | 0.1 | (0.9, 0.95) | Cosine |
| Medium (355M) | MUON | $1.5 \times 10^{-3}$ | $1 \times 10^{-2}$ | 0.1 | (0.9, 0.95) | Cosine |
| | RMNP | $1.5 \times 10^{-3}$ | $1 \times 10^{-2}$ | 0.1 | (0.9, 0.95) | Cosine |
| | ADAMW | $2 \times 10^{-4}$ | — | 0.1 | (0.9, 0.95) | Cosine |
| Large (770M) | MUON | $1 \times 10^{-3}$ | $6.67 \times 10^{-3}$ | 0.1 | (0.9, 0.95) | Cosine |
| | RMNP | $1 \times 10^{-3}$ | $6.67 \times 10^{-3}$ | 0.1 | (0.9, 0.95) | Cosine |
| | ADAMW | $2 \times 10^{-4}$ | — | 0.1 | (0.9, 0.95) | Cosine |
| XLarge (1.5B) | MUON | $1 \times 10^{-3}$ | $6.67 \times 10^{-3}$ | 0.1 | (0.9, 0.95) | Cosine |
| | RMNP | $1 \times 10^{-3}$ | $2 \times 10^{-3}$ | 0.1 | (0.9, 0.95) | Cosine |

*Table 8.* LLaMA Model Configurations and specified hyperparameters.

| Params | Hidden | Intermediate | Heads | Blocks | Steps | Warm-up | Token Count | Batch Size | LR schedule |
|---|---|---|---|---|---|---|---|---|---|
| 60M | 512 | 1376 | 8 | 8 | 10K | 1K | 1B | 512 | Cosine |
| 130M | 768 | 2048 | 12 | 12 | 20K | 2K | 2B | 512 | Cosine |
| 350M | 1024 | 2736 | 16 | 24 | 60K | 6K | 6B | 512 | Cosine |
| 1B | 2048 | 5461 | 32 | 24 | 90K | 9K | 9B | 512 | Cosine |

---

[1]We use the shuffled version by Lozhkov et al. (2024): https://huggingface.co/datasets/karpathy/fineweb-edu-100b-shuffle.

*Table 9.* Hyperparameter search on GPT-2 Small with ADAMW learning rate fixed at $3 \times 10^{-3}$.

| Matrix LR | 0.01 | 0.015 | 0.02 | 0.025 |
|---|---|---|---|---|
| MUON | 23.62 | 26.74 | **22.86** | 22.87 |
| Matrix LR | 0.002 | 0.003 | 0.004 | 0.005 |
| RMNP | 23.58 | 22.95 | **22.82** | 26.42 |

*Table 10.* Hyperparameter search on GPT-2 Medium with ADAMW learning rate fixed at $1.5 \times 10^{-3}$.

| Matrix LR | 0.005 | 0.01 | 0.02 | 0.03 |
|---|---|---|---|---|
| MUON | 18.33 | 18.26 | **17.38** | 17.44 |
| Matrix LR | 0.001 | 0.002 | 0.003 | 0.005 |
| RMNP | 18.58 | **17.31** | 17.42 | 17.88 |

*Table 11.* Hyperparameter search on LLaMA-60M (LM head and token-embedding parameters handled by ADAMW). Validation perplexity is reported.

| Matrix LR | 0.005 | 0.01 | 0.02 | 0.03 | 0.04 |
|---|---|---|---|---|---|
| MUON | 29.90 | **29.58** | 30.46 | 30.49 | 30.03 |
| Matrix LR | 0.001 | 0.004 | 0.005 | 0.01 | 0.02 |
| RMNP | 31.00 | 28.99 | **28.95** | 29.26 | 29.64 |
| Matrix LR | 0.005 | 0.01 | 0.02 | 0.03 | 0.04 |
| SHAMPOO | 31.04 | 30.69 | 31.07 | **29.74** | 30.61 |
| Matrix LR | 0.001 | 0.002 | 0.003 | 0.004 | 0.005 |
| SOAP | 30.85 | 29.30 | **29.14** | 29.36 | 29.57 |

*Table 12.* Hyperparameter search on LLaMA-130M (LM head and token-embedding parameters handled by ADAMW). Validation perplexity is reported.

| Matrix LR | 0.005 | 0.01 | 0.02 | 0.03 |
|---|---|---|---|---|
| MUON | 22.51 | **22.42** | 22.47 | 22.51 |
| Matrix LR | 0.01 | 0.02 | 0.03 | 0.04 |
| RMNP | 22.42 | 22.49 | **22.14** | 23.31 |
| Matrix LR | 0.005 | 0.01 | 0.03 | 0.04 |
| SHAMPOO | 23.22 | 22.70 | **22.69** | 23.49 |
| Matrix LR | 0.001 | 0.002 | 0.003 | 0.005 |
| SOAP | 23.13 | **22.61** | 22.78 | 23.11 |

*Table 13.* Hyperparameter search on LLaMA-350M (LM head and token-embedding parameters handled by ADAMW). Validation perplexity is reported.

| Matrix LR | 0.003 | 0.004 | 0.005 |
|---|---|---|---|
| MUON | 17.01 | **16.87** | 16.89 |
| Matrix LR | 0.003 | 0.004 | 0.005 |
| RMNP | 17.02 | 16.86 | **16.85** |

### E.3. Extended Training Budget

To verify that the advantage of RMNP over MUON and ADAMW persists at longer training horizons, we additionally extend the training budget to $2\times$ the standard length for three model-dataset combinations: GPT-2 Small on OpenWebText (20K steps), LLaMA-60M on C4 (20K steps), and LLaMA-130M on C4 (40K steps). Final validation perplexity is reported in Table 14. RMNP achieves the lowest perplexity in every cell, indicating that its advantage is not a short-horizon artifact.

*Table 14.* Final validation PPL ($\downarrow$) under an extended training budget ($2\times$ standard). Lower is better.

| Optimizer | LLaMA 60M | LLaMA 130M | GPT-2 Small (OWT) |
|---|---|---|---|
| ADAMW | 28.23 | 21.35 | 20.97 |
| MUON | 27.03 | 20.84 | 20.88 |
| RMNP | **26.44** | **20.53** | **20.41** |

### E.4. LM Head and Embedding Ablation

We additionally study the effect of extending the matrix-aware optimizer to cover the LM head and token-embedding parameters (rather than letting ADAMW handle them). Tables 15 and 16 report the LR-sweep results for MUON and RMNP on LLaMA-60M and LLaMA-130M when LM head and embedding parameters are included in the matrix-optimizer parameter group.

*Table 15.* Hyperparameter search on LLaMA-60M with LM head and embedding parameters optimized by the matrix optimizer. Validation perplexity is reported.

| Matrix LR | 0.005 | 0.01 | 0.02 | 0.03 | 0.04 |
|---|---|---|---|---|---|
| MUON | 30.41 | 29.49 | 29.63 | **29.38** | 30.57 |
| Matrix LR | 0.001 | 0.004 | 0.005 | 0.01 | 0.02 |
| RMNP | 34.92 | 29.56 | 29.28 | **29.03** | 31.45 |

*Table 16.* Hyperparameter search on LLaMA-130M with LM head and embedding parameters optimized by the matrix optimizer. Validation perplexity is reported.

| Matrix LR | 0.005 | 0.01 | 0.02 | 0.03 |
|---|---|---|---|---|
| MUON | 22.89 | **22.55** | 22.80 | 22.87 |
| Matrix LR | 0.01 | 0.02 | 0.03 | 0.04 |
| RMNP | 22.16 | 22.11 | **22.06** | 23.62 |

Overall, as shown in Tables 15 and 16, including the LM head and token-embedding parameters in the matrix-optimizer group has a negligible effect on final perplexity: the differences across all settings are within 0.13 PPL and show no consistent trend across model scales or optimizers. For the GPT-2 experiments reported in the main body, the LM head and embedding parameters are optimized together with the other matrix parameters using the matrix optimizer.

## F. Full Training Curves

This section presents the complete set of training-loss, validation-loss, and gradient clip-rate curves for every model-dataset combination evaluated in this paper, comparing ADAMW, MUON, and RMNP. In every plot RMNP is drawn on top so that it is never occluded by the other two curves. All curves use the canonical hyperparameters reported in Appendix E.2.

### F.1. Final Validation Perplexity Summary

Before presenting the full training curves, we summarize the final validation perplexity for the three main pre-training settings as bar charts paired with the corresponding numeric tables. RMNP attains the lowest final perplexity in every cell. The three settings are jointly visualized in Figure 9, reproduced here from the main body for self-containment.

*Table 17.* Final validation perplexity ($\downarrow$) across all three pre-training settings – LLaMA on C4, GPT-2 on FineWeb-Edu-100B, and GPT-2 on OpenWebText. The RMNP row is highlighted; per-column best values are shown in **bold**.

| | LLaMA on C4 | | | | GPT-2 on FineWeb-Edu-100B | | | | GPT-2 on OpenWebText | | |
|---|---|---|---|---|---|---|---|---|---|---|---|
| | 60M | 130M | 350M | 1B | Small | Medium | Large | XLarge | Small | Medium | Large |
| ADAMW | 33.28 | 23.24 | 17.08 | 15.33 | 23.85 | 18.19 | 14.81 | 13.12 | 24.19 | 18.80 | 15.27 |
| MUON | 29.58 | 22.42 | 16.87 | 14.13 | 22.71 | 17.13 | 14.16 | 12.97 | 22.86 | 17.38 | 14.67 |
| RMNP | **28.95** | **22.14** | **16.85** | **13.75** | **22.60** | **17.07** | **13.75** | **12.58** | **22.82** | **17.31** | **14.43** |

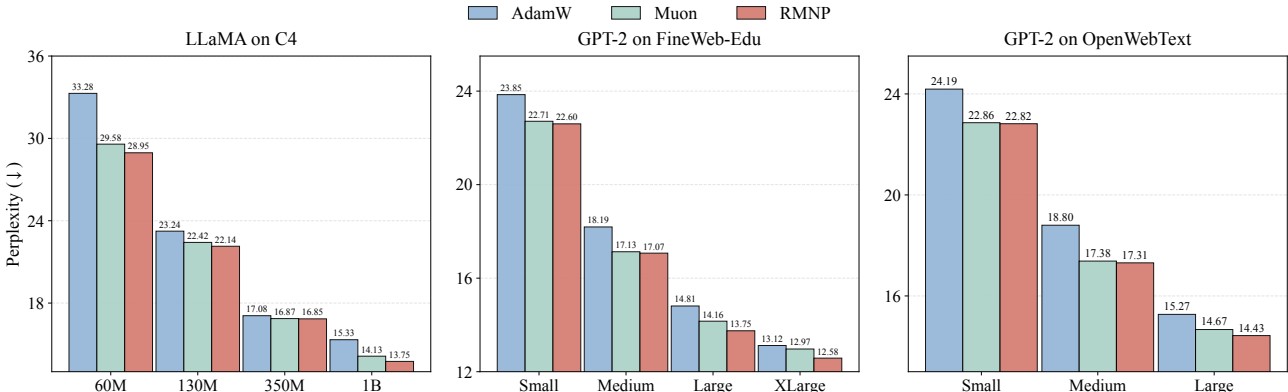

*Figure 9.* Final validation perplexity (↓) across three pretraining settings (reproduced from Figure 5 for self-containment). **Left:** LLaMA on C4. **Middle:** GPT-2 on FineWeb-Edu-100B. **Right:** GPT-2 on OpenWebText. Numeric values are reported in Table 17.

## F.2. GPT-2 on OpenWebText

Figures 10–12 show the training and validation loss for GPT-2 SMALL, MEDIUM, and LARGE pre-trained on OpenWebText. Across all three scales RMNP consistently matches or slightly improves upon MUON, while both clearly outperform ADAMW.

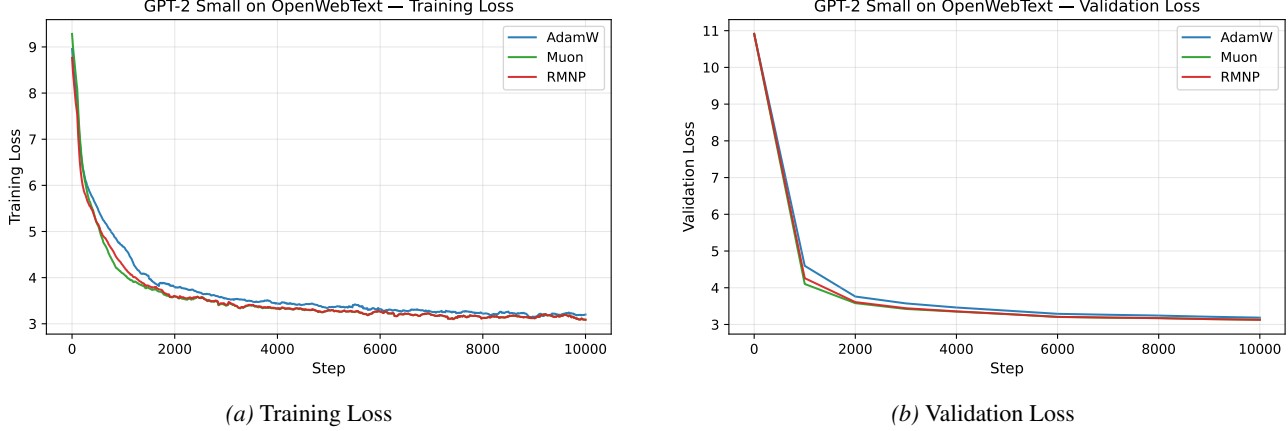

*(a)* Training Loss          *(b)* Validation Loss

*Figure 10.* GPT-2 SMALL (125M) on OpenWebText. Training loss is smoothed with a 20-step rolling window. RMNP ends with the lowest training and validation loss.

## F.3. GPT-2 on FineWeb-Edu-100B

Figures 13–16 present the training and validation loss curves for GPT-2 SMALL, MEDIUM, LARGE, and XLARGE pre-trained on FineWeb-Edu-100B. Across all four scales RMNP again matches or surpasses MUON and clearly outperforms ADAMW, demonstrating that the trend observed on OpenWebText extends to a more competitive corpus and a larger token budget.

## F.4. LLaMA on C4

Figures 17–20 report the training and validation loss curves for the four LLaMA scales pretrained on C4. RMNP consistently delivers a slight but stable improvement over MUON across all sizes, and the gap between matrix-aware optimizers and ADAMW widens as model scale grows.

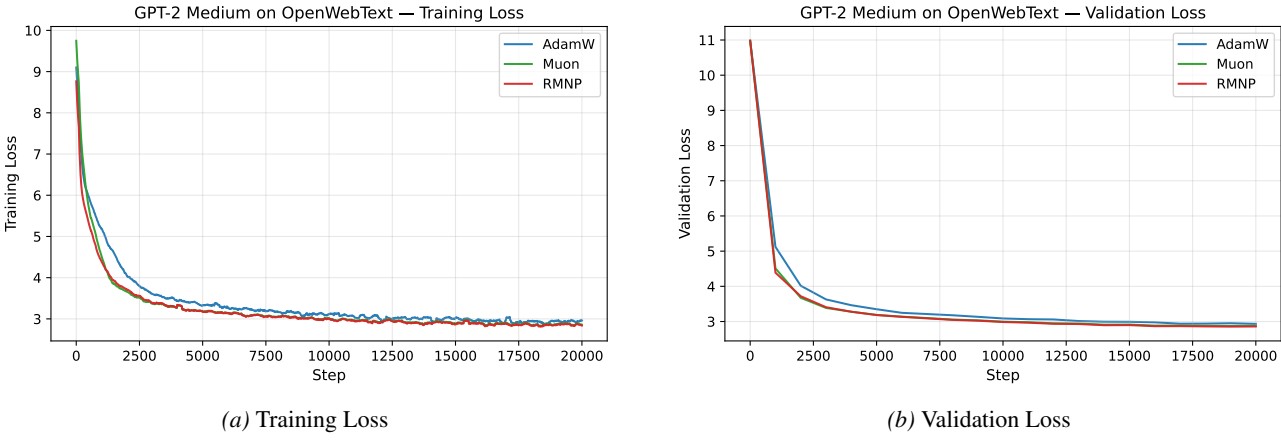

*(a)* Training Loss                     *(b)* Validation Loss

*Figure 11.* GPT-2 MEDIUM (355M) on OpenWebText. RMNP achieves the lowest validation loss among the three optimizers.

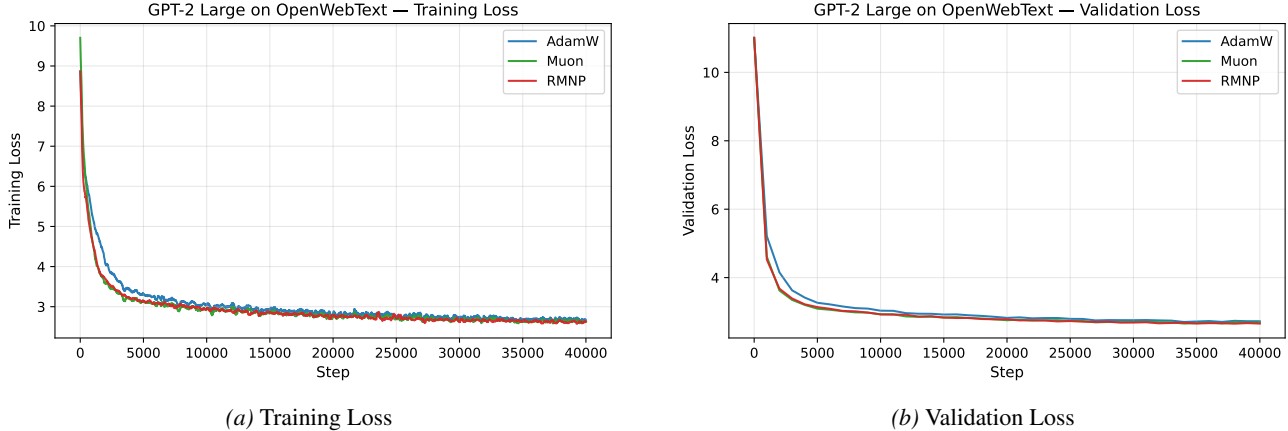

*(a)* Training Loss                     *(b)* Validation Loss

*Figure 12.* GPT-2 LARGE (770M) on OpenWebText. RMNP's lead over MUON grows with model scale.

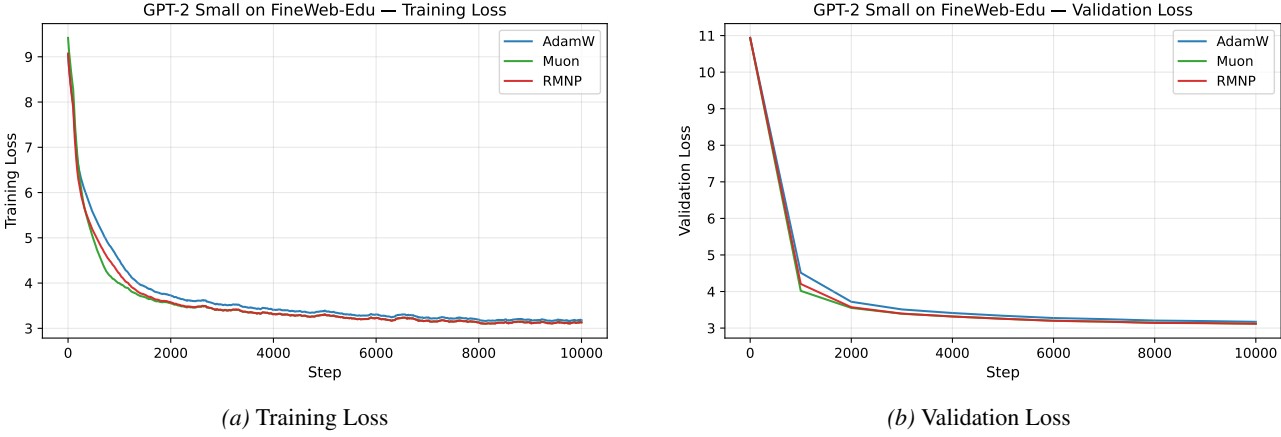

*(a)* Training Loss                     *(b)* Validation Loss

*Figure 13.* GPT-2 SMALL (125M) on FineWeb-Edu-100B. RMNP attains the lowest training and validation loss.

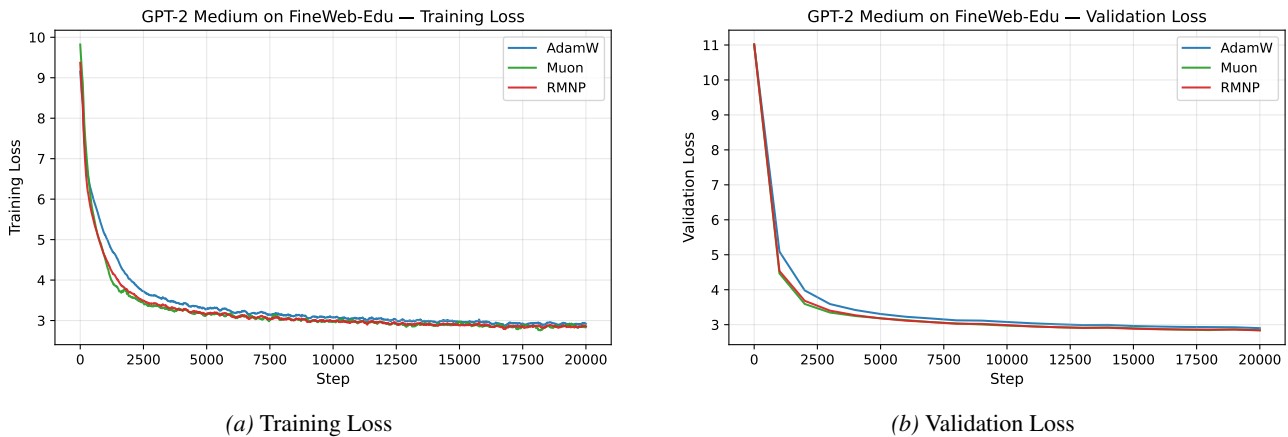

*(a)* Training Loss                                            *(b)* Validation Loss

*Figure 14.* GPT-2 MEDIUM (355M) on FineWeb-Edu-100B. RMNP maintains a slight but consistent edge over MUON on validation loss while ADAMW lags throughout training.

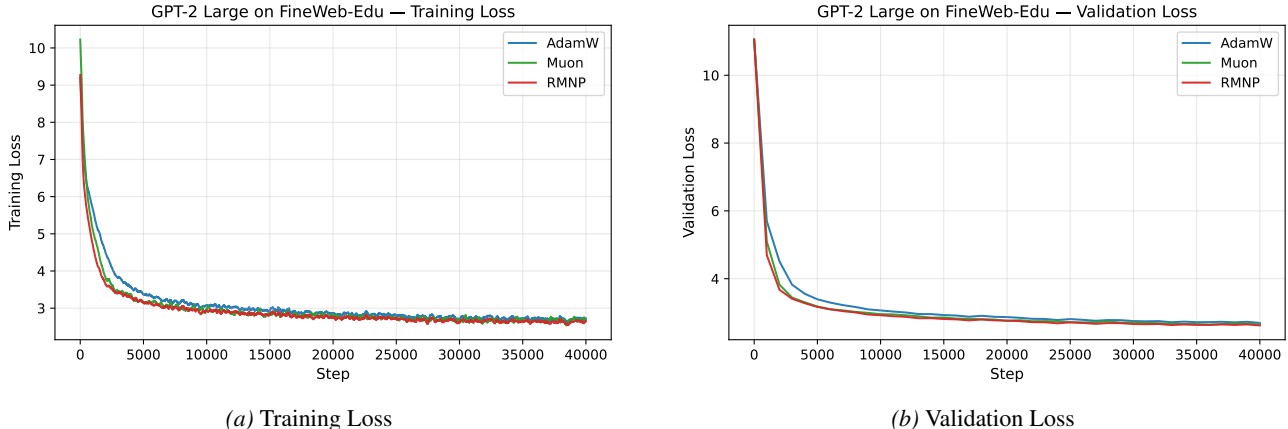

*(a)* Training Loss                                            *(b)* Validation Loss

*Figure 15.* GPT-2 LARGE (770M) on FineWeb-Edu-100B. RMNP's lead over MUON grows with model scale, while ADAMW converges to a noticeably higher validation loss.

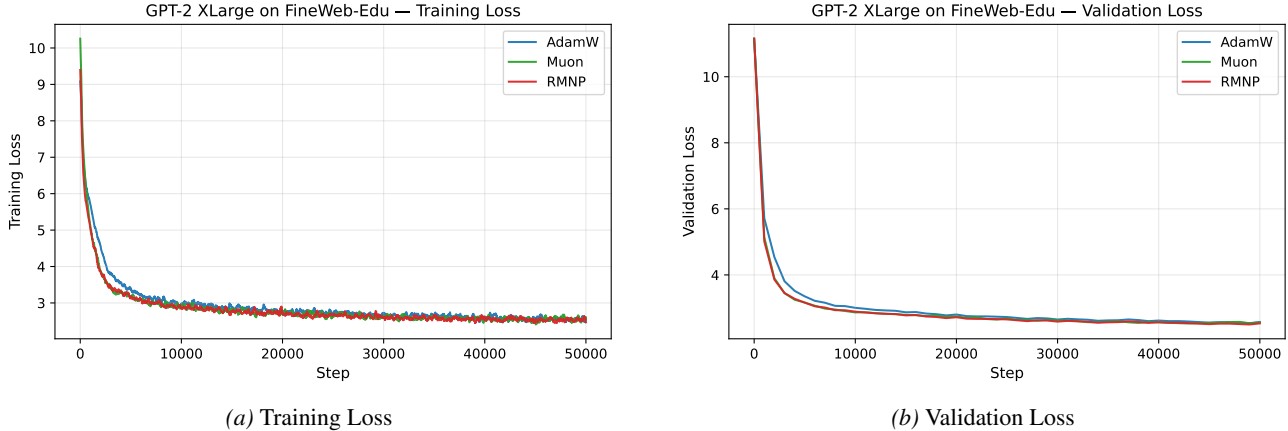

*(a)* Training Loss                                            *(b)* Validation Loss

*Figure 16.* GPT-2 XLARGE (1.5B) on FineWeb-Edu-100B. RMNP continues to track MUON closely and surpasses it in late training, while delivering an order-of-magnitude reduction in preconditioning wall-clock cost (Appendix D).

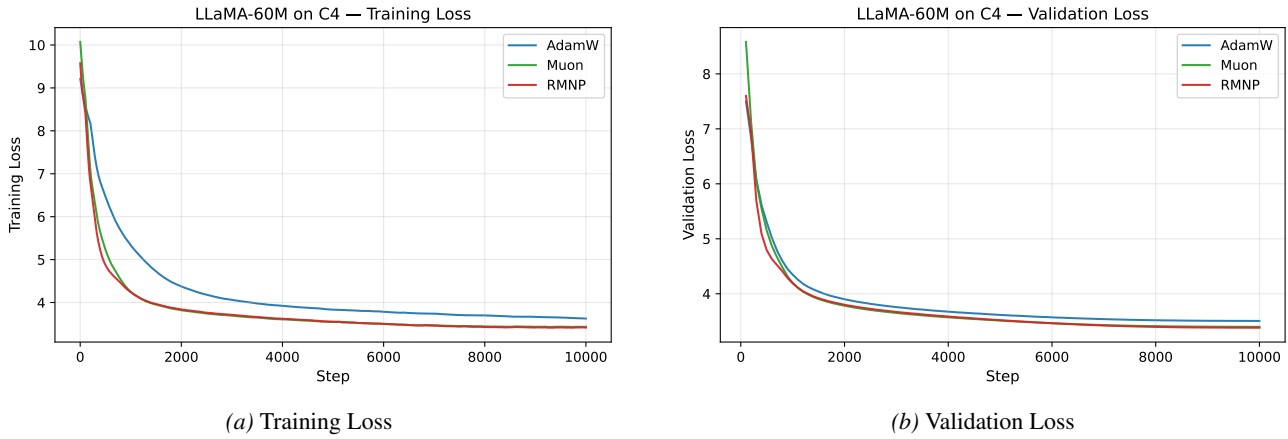

*(a)* Training Loss

*(b)* Validation Loss

*Figure 17.* LLAMA-60M on C4. The available ADAMW log extends beyond the canonical training horizon and has been clipped to match MUON and RMNP on a shared x-range. RMNP achieves the lowest validation loss; MUON is close behind, while ADAMW converges to a clearly higher value.

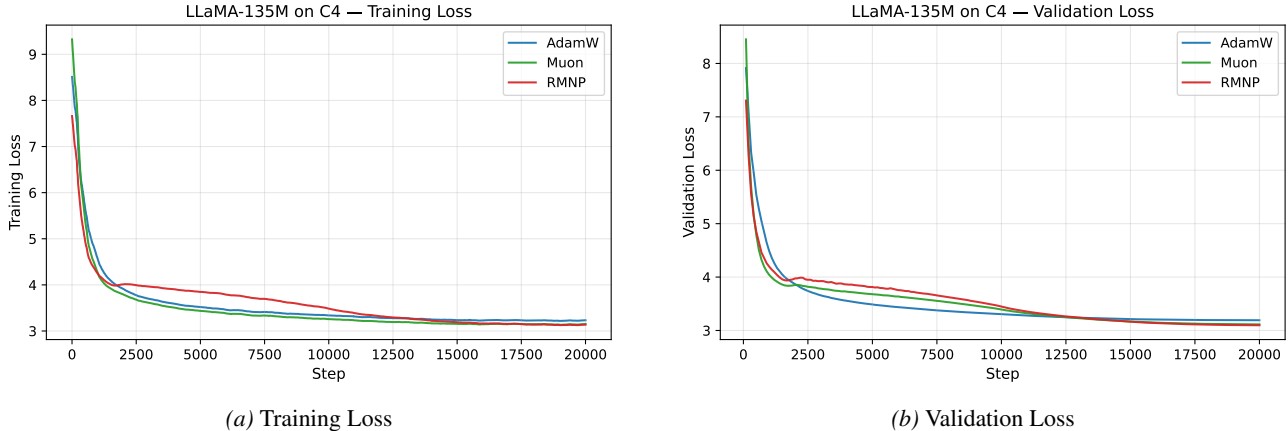

*(a)* Training Loss

*(b)* Validation Loss

*Figure 18.* LLAMA-130M on C4. RMNP outperforms both baselines in validation loss throughout training.

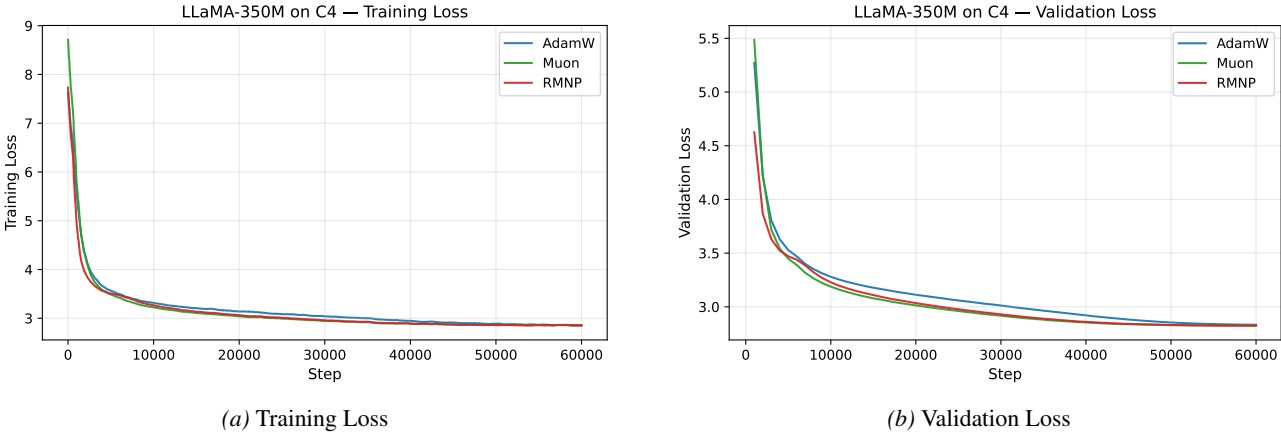

*(a)* Training Loss

*(b)* Validation Loss

*Figure 19.* LLAMA-350M on C4. RMNP matches MUON on training loss and edges ahead on validation loss in late training.

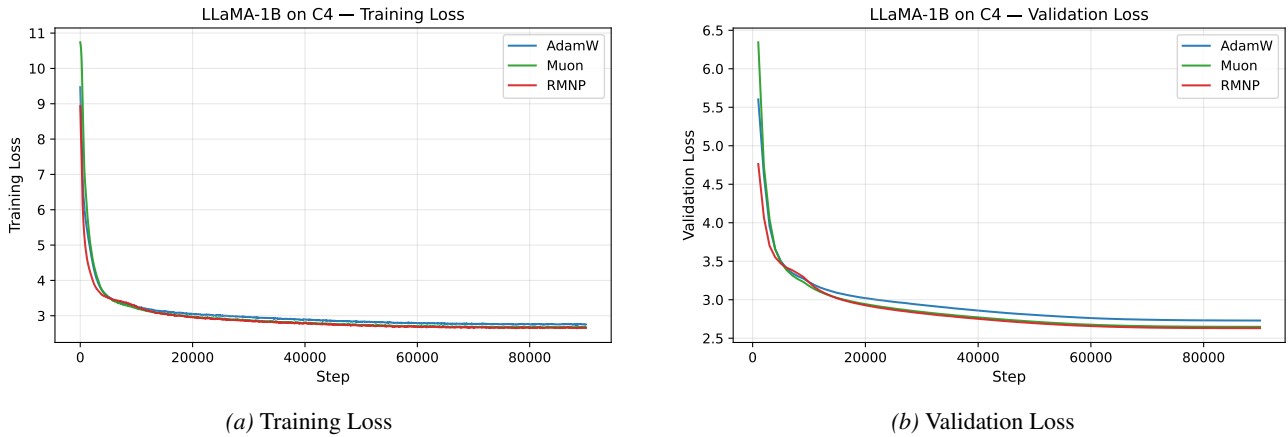

*(a)* Training Loss

*(b)* Validation Loss

*Figure 20.* LLAMA-1B on C4. The RMNP curve tracks MUON closely on both training and validation loss while delivering a substantially lower preconditioning cost.

### F.5. Mamba on FineWeb-Edu

We additionally evaluate RMNP on a Mamba state-space language model trained on FineWeb-Edu to verify that the row-wise normalized preconditioner generalizes beyond Transformer attention. Figure 21 reports the training loss and validation perplexity, comparing ADAMW, MUON, and RMNP. Despite the architectural difference, RMNP tracks MUON essentially in lockstep and both clearly outperform ADAMW.

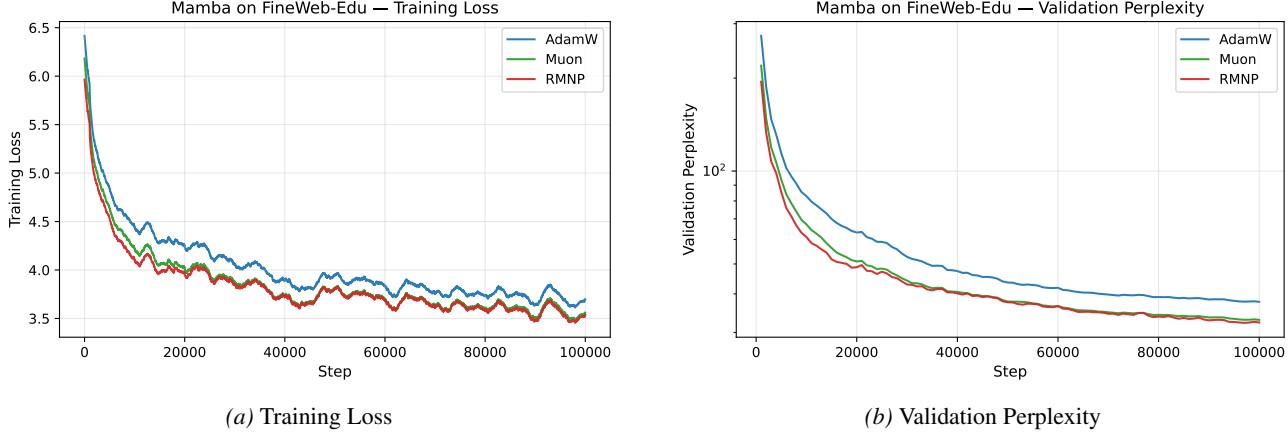

*(a)* Training Loss

*(b)* Validation Perplexity

*Figure 21.* Mamba on FineWeb-Edu. Validation perplexity is shown on a log scale. RMNP matches MUON throughout training and clearly outperforms ADAMW, demonstrating that the row-wise normalized preconditioner generalizes beyond Transformer architectures to state-space models.

The same diagonal-dominance property observed for Transformer-family models continues to hold for Mamba's matrix parameters. Figure 22 reports both the global aggregate metrics (panel (a)) and the per-parameter metrics for three representative matrix parameters (panel (b)) of Mamba; all three ratio metrics rise above the threshold $r = 1$ shortly after warm-up and remain there throughout training.

The learning-rate sweep underlying the Mamba experiment is reported in Table 18. We fix the ADAMW learning rate at $1 \times 10^{-4}$ and sweep the matrix learning rate; the table reports final validation perplexity (lower is better).

### F.6. ResNet-18 on CIFAR-10

To verify that RMNP is competitive on architectures and modalities outside of language modeling, we compare RMNP and MUON on the canonical RESNET-18 / CIFAR-10 image-classification benchmark. Figure 23 reports the training/test loss and training/test accuracy for both optimizers (ADAMW omitted to keep the comparison focused on the two matrix-aware

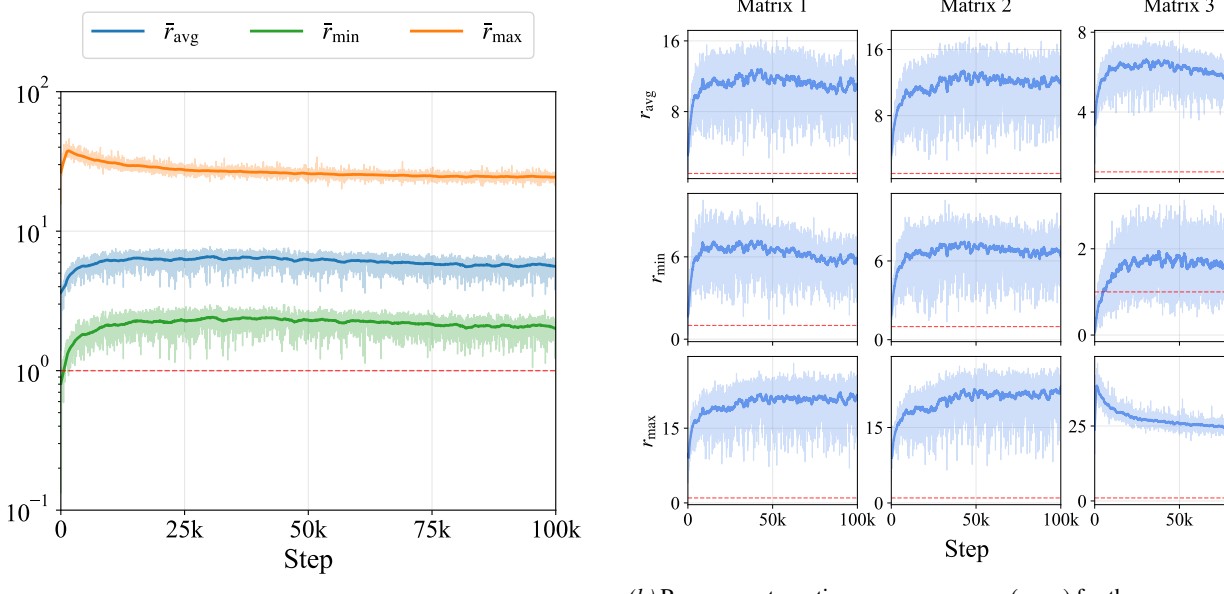

*(a)* Global ratios $\bar{r}_{\text{avg}}, \bar{r}_{\text{min}}, \bar{r}_{\text{max}}$ (log-scale y-axis).

*(b)* Per-parameter ratios $r_{\text{avg}}, r_{\text{min}}, r_{\text{max}}$ (rows) for three representative matrix parameters (columns).

*Figure 22.* Diagonal dominance ratios for Mamba pre-training on FineWeb-Edu. Transparent curves: raw values; solid curves: smoothed with window size 50. Red dashed line: $y = 1$ threshold. All metrics remain above the threshold throughout training, demonstrating that the row-wise block-diagonal dominance property holds for the Mamba state-space architecture both at the global aggregate level (panel (a)) and at the per-parameter level (panel (b)).

*Table 18.* Hyperparameter search on Mamba (FineWeb-Edu) with ADAMW learning rate fixed at $1 \times 10^{-4}$. Validation perplexity is reported.

| Matrix LR | $6.67 \times 10^{-4}$ | 0.008 | 0.009 |
|---|---|---|---|
| MUON | 36.55 | **32.95** | 33.02 |
| Matrix LR | $6.67 \times 10^{-4}$ | $8 \times 10^{-4}$ | $1 \times 10^{-3}$ |
| RMNP | 32.56 | **32.32** | 32.33 |

methods). RMNP closely tracks MUON throughout training and converges to essentially identical final accuracy, indicating that the row-wise normalized preconditioner is effective in the convolutional regime as well.

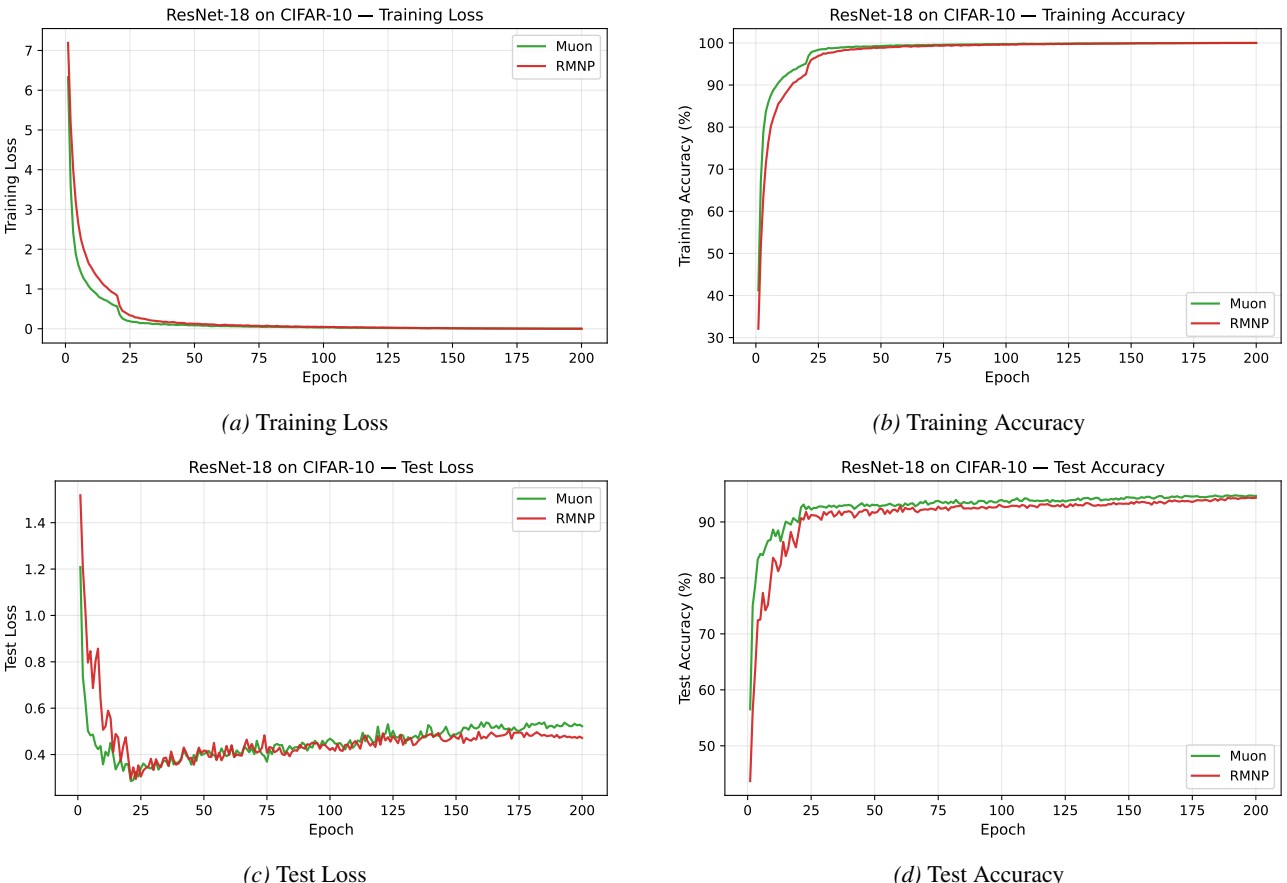

*(a)* Training Loss

*(b)* Training Accuracy

*(c)* Test Loss

*(d)* Test Accuracy

*Figure 23.* RESNET-18 on CIFAR-10, comparing MUON and RMNP. The two matrix-aware optimizers track each other closely and converge to essentially identical final accuracy, demonstrating that RMNP extends to convolutional vision tasks without loss of optimization quality.

We also extend the diagonal-dominance analysis of Section C to RESNET-18: the row-wise block-diagonal dominance property continues to hold beyond fully-connected matrix parameters. Figure 24 reports both the global aggregate metrics (panel (a)) and the per-parameter metrics for three representative matrix parameters (panel (b)).

The matrix learning-rate sweep for RESNET-18 is reported in Table 19. We fix the ADAMW learning rate at 0.006 and sweep the matrix learning rate; the table reports final test accuracy (higher is better).

*Table 19.* Test accuracy (%) on CIFAR-10 for RESNET-18: matrix LR search with ADAMW learning rate fixed at 0.006. Higher is better.

| Matrix LR | 0.01 | 0.04 | 0.05 |
|---|---|---|---|
| MUON | 94.57 | **94.65** | 94.39 |
| Matrix LR | 0.006 | 0.008 | 0.01 |
| RMNP | **94.33** | 93.93 | 94.31 |

## F.7. Gradient Clip-Rate Trajectories

We additionally report the gradient clip rate (the per-step fraction of times the gradient norm exceeds the clip threshold) for the GPT-2 pre-training runs on OpenWebText and FineWeb-Edu-100B. Two views are provided per dataset:

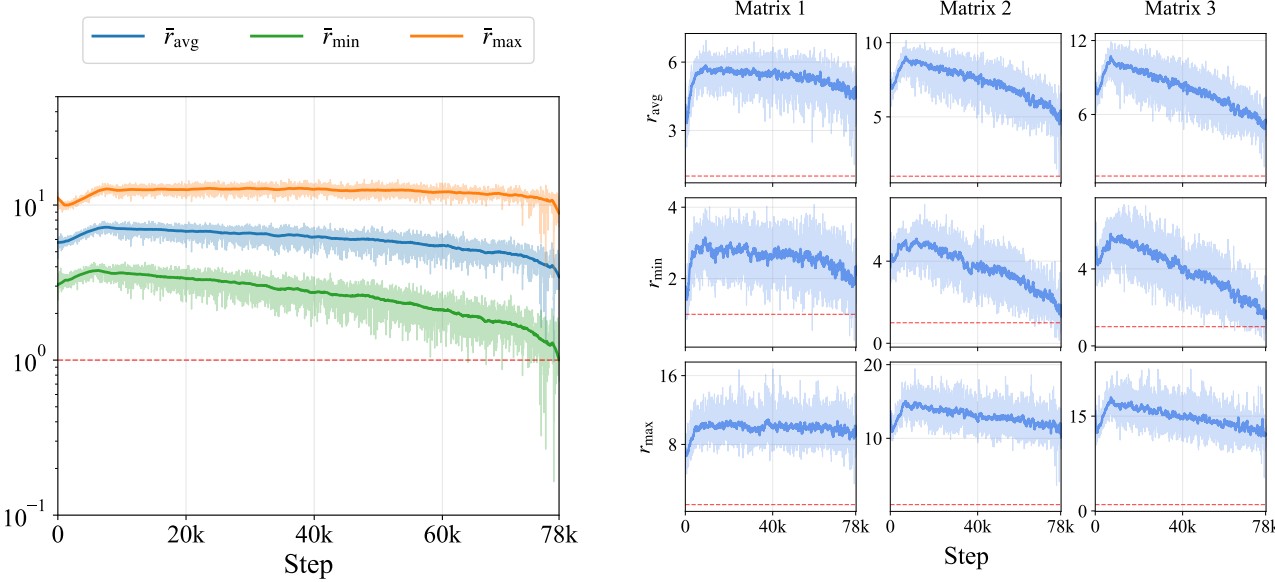

*(a)* Global ratios $\overline{r}_{\text{avg}}, \overline{r}_{\text{min}}, \overline{r}_{\text{max}}$ (log-scale y-axis).

*(b)* Per-parameter ratios $r_{\text{avg}}, r_{\text{min}}, r_{\text{max}}$ (rows) for three representative matrix parameters (columns).

*Figure 24.* Diagonal dominance ratios for RESNET-18 training on CIFAR-10. Transparent curves: raw values; solid curves: smoothed with window size 50. Red dashed line: $y = 1$ threshold. All metrics remain above the threshold throughout training, demonstrating that the row-wise block-diagonal dominance property holds for the convolutional vision architecture both at the global aggregate level (panel (a)) and at the per-parameter level (panel (b)).

- *Per-size grid* (Figures 25 and 27) overlays ADAMW, MUON, and RMNP within each cell, with the raw step on the x-axis. RMNP is drawn on top.

- *Cross-scale comparison* (Figures 26 and 28) places the four model scales together within each optimizer panel, with the x-axis rescaled to relative training progress (%). Within each panel a single hue is used and shade encodes model scale, so a darker line is a larger model.

Across both datasets, larger models keep their gradients clipped for a longer fraction of training, with ADAMW on GPT-2 XLARGE an extreme case where every step is clipped throughout the run. RMNP consistently begins to release the clip threshold earliest of the three optimizers, indicating that its row-normalized update reduces gradient-norm volatility relative to both ADAMW and MUON.

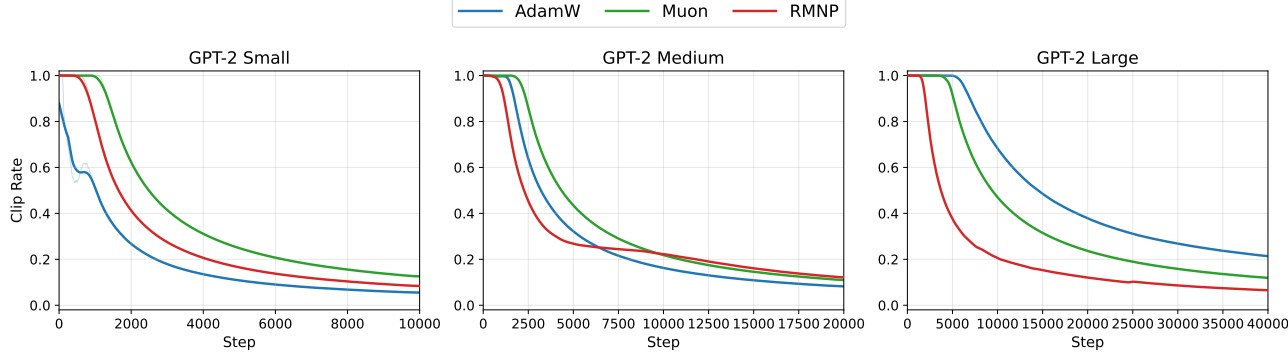

*Figure 25.* Gradient clip rate during GPT-2 pre-training on OpenWebText, one panel per model size. Transparent line: raw values; solid line: 50-step rolling mean. RMNP (red) is drawn last so it sits on top of ADAMW (blue) and MUON (green).

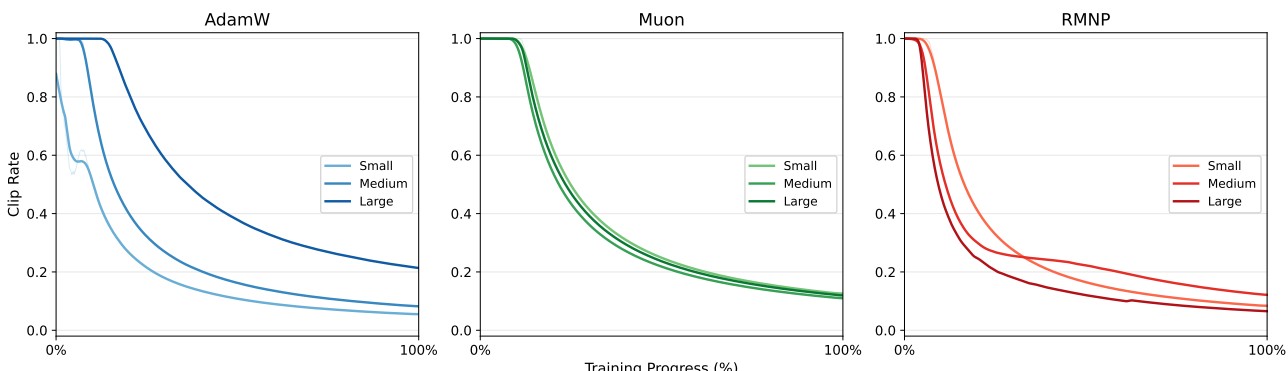

*Figure 26.* Gradient clip rate during GPT-2 pre-training on OpenWebText, with x-axis rescaled to relative training progress (%). Each panel shows one optimizer; within a panel, lighter to darker shades encode SMALL/MEDIUM/LARGE. The clip rate falls below 1.0 progressively later for larger models.

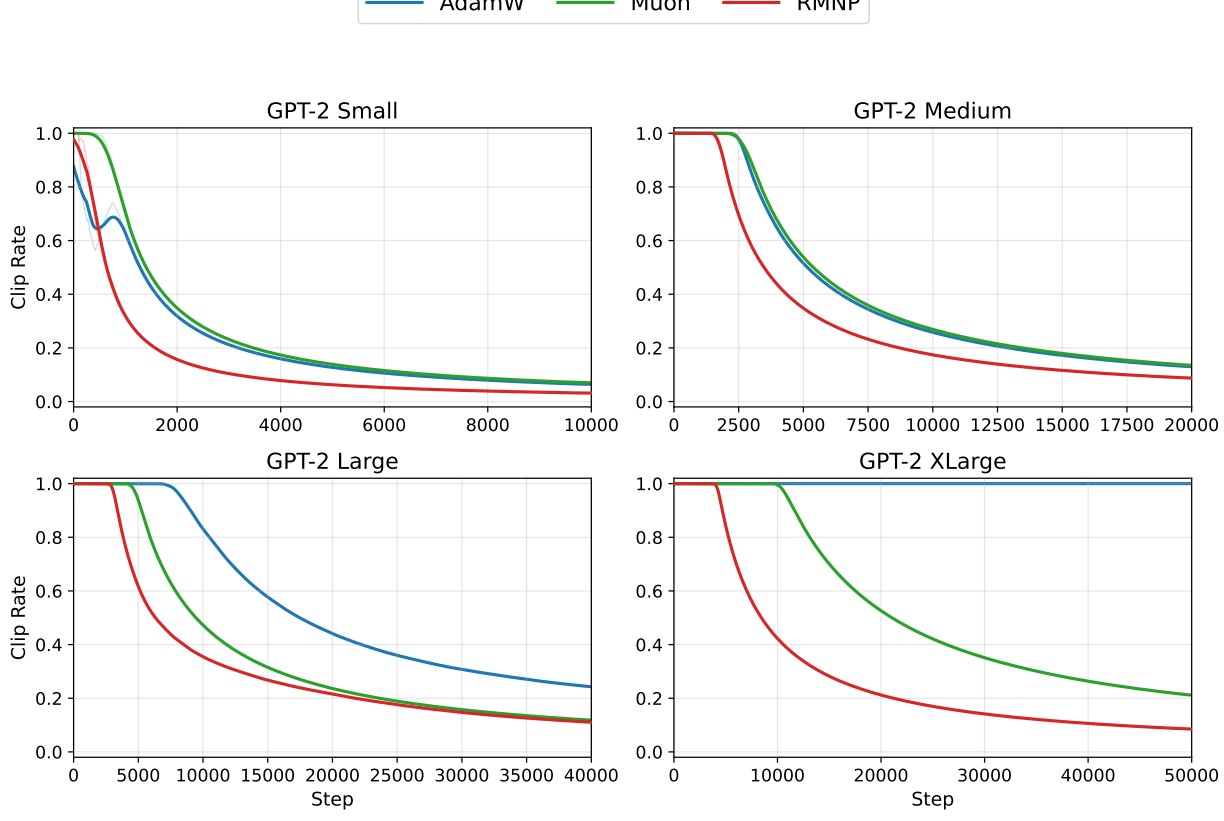

*Figure 27.* Gradient clip rate during GPT-2 pre-training on FineWeb-Edu-100B, one panel per model size. Transparent line: raw values; solid line: 50-step rolling mean. ADAMW on the XLARGE (1.5B) model has its gradients clipped at every step throughout the entire run; both MUON and RMNP progressively reduce the clip rate.

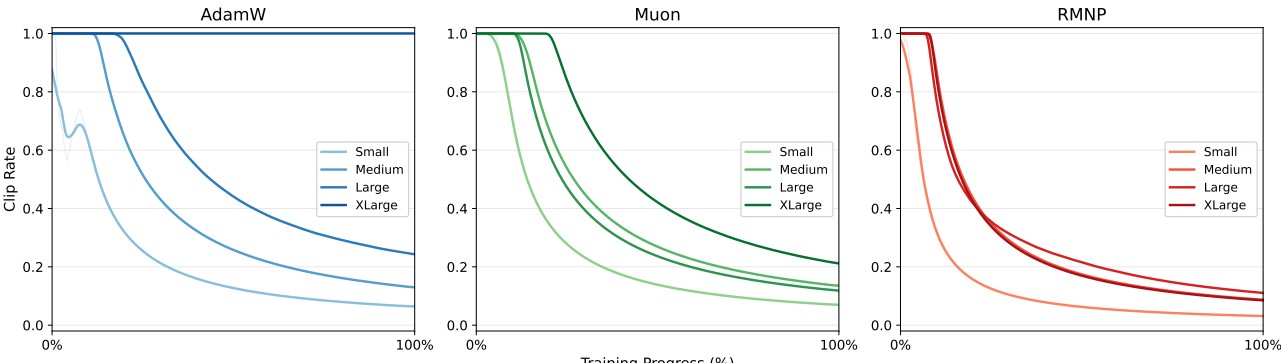

*Figure 28.* Gradient clip rate during GPT-2 pre-training on FineWeb-Edu-100B, with x-axis rescaled to relative training progress (%). Each panel shows one optimizer; lighter to darker shades within a panel encode SMALL/MEDIUM/LARGE/XLARGE. The size-dependent delay before the clip rate begins to drop is most pronounced under ADAMW (left) and least pronounced under RMNP (right).

