# OpenReview forum: "RMNP: Row-Momentum Normalized Preconditioning for Scalable Matrix-Based Optimization"
_ICML.cc/2026/Conference — ICML 2026 regular_

### Official Review · Reviewer_fjwp · 2026-03-04

**Soundness:** 3
**Presentation:** 3
**Significance:** 2
**Originality:** 2
**Overall Recommendation:** 4
**Confidence:** 4

**Summary:**

This paper proposes the RMNP  optimizer. Motivated by the observation that the Transformer Hessian exhibits a row block-diagonal dominance structure, the authors replace the computationally expensive Newton–Schulz iteration used in the MUON optimizer with a simple row-wise $l_2$ normalization. This reduces the per-iteration preprocessing complexity from $\mathcal{O}(mn \cdot \min(m,n))$ to $\mathcal{O}(mn)$. The paper also provides a non-convex convergence guarantee, matching the latest theoretical results established for MUON. Experiments on GPT-2 models (up to 1.5B parameters) and LLaMA models show that the optimization performance (in terms of perplexity) is comparable to, and sometimes slightly better than, MUON and AdamW.

**Compliance With Llm Reviewing Policy:**

Affirmed.

**Final Justification:**

This work identifies the most computationally expensive and scalability-limiting subroutine in MUON and proposes a minimal yet practical alternative. The experimental evaluation extends beyond toy settings, demonstrating meaningful empirical support. I believe the core idea has genuine value, and the results suggest that RMNP is not merely a trick. While I do have some concerns regarding the methodological novelty, the strong motivation of the paper largely compensates for this. Therefore, I recommend acceptance.

**Key Questions For Authors:**

Question:
1.Robustness on non-Transformer architectures: Has RMNP been tested on architectures that may not exhibit strong row-wise Hessian dominance?

**Limitations:**

Yes

**Strengths And Weaknesses:**

**Strengths**

- The core idea is simple and practical. Starting from the MUON preconditioning form, the paper explicitly makes a structured approximation that keeps only the row block–diagonal structure, and shows that the resulting update reduces to row-wise normalization (Eq. (3)(4)). This makes the method much lighter computationally and easier to implement in practice.

- Strong empirical results. Experiments on GPT-2 (Small, Medium, Large) and LLaMA (60M, 135M, 350M) show that RMNP consistently achieves lower perplexity than both AdamW and MUON.

- Comprehensive theoretical guarantees. The paper develops a solid theoretical framework and proves convergence under both the Frobenius norm ($L_F$) and mixed norm ($L_{\infty,2}$) smoothness assumptions, matching the best-known theoretical guarantees for MUON.


**Weaknesses**

- Limited comparison with a broader set of optimizers. The experiments mainly focus on MUON and AdamW. From the perspective of large-scale model training, it would be interesting to see how RMNP compares with methods such as Shampoo, SOAP, or other recent engineering-oriented distributed second-order approximations, especially in terms of the throughput–performance trade-off under the same memory and compute budget.

- Dependence on the “row block–diagonal dominance” property of the Hessian. The method largely relies on this structural assumption. While the paper cites evidence that it holds for Transformers, it is unclear whether the same geometric property generally appears in non-Transformer architectures (e.g., Mamba or CNNs).

- Potential novelty concerns. Although the motivation is clearly presented, the method essentially performs row-wise normalization on the momentum, and similar normalization operations have been discussed in prior work. This raises some questions about the level of novelty.


[1] NorMuon: Making Muon More Efficient and Scalable

[2] A Minimalist Optimizer Design for LLM Pretraining

[3] Training Deep Learning Models with Norm-Constrained LMOs

---

> ### Author Rebuttal · Authors · 2026-03-31
>
> We appreciate the reviewer's valuable comments. We provide additional evidence to support our findings and address the reviewer's concerns through the following link: https://anonymous.4open.science/r/RMNP-317C
>
>
>
>
> **W1**
>
>
> We have supplemented additional results(Table 14,16) of Shampoo and SOAP to further support the performance of our optimizer. The experimental results indicate that RMNP still achieved the lowest PPL .
>
>
>
> **W2&Q1**
> A.We have supplemented additional experimental evidence（Figure 12，13，14，15） for the row block diagonal dominance of the preconditioner on CNN, Mamba, and Transformers at different scales. The experimental results support the following:
> 1. Across Transformers of different scales, MUON's preconditioner consistently exhibits the row block diagonal dominance property. **Moreover, we additionally find that this dominance becomes more pronounced as the model scale increases.**
> 2. For weight matrices in CNN and Mamba architectures, we also observe the row block diagonal dominance phenomenon in their MUON preconditioners.
>
>
>
> B.We have conducted additional training comparisons on other architectures and tasks (Mamba:Table 18 and CNN: Table 17). The conclusions show that RMNP delivers comparable performance, while our method remains the simplest and most efficient.
>
>
> **W3**
>
>
> We thank the reviewer oqBk and reviewer fjwp for pointing out concurrent works [1-3]. We include some discussion in our paper (Line 246) and We will incorporate more discussion on concurrent works in the revised draft. We like to provide a unified clarification here first:
>
> 1. First, [1] is fundamentally different from [2-5] and our work. The algorithm in [1] combines MUON and Adam: it first applies Newton-Schulz (NS) iteration to the parameters via MUON, and then performs optimization using the improved Adam proposed in [7], where row normalization occurs within this improved Adam. (Specifically, the Adam variant in [7] replaces the standard per-parameter normalization with row-wise normalization.) In contrast, both [2-6] and our work apply row normalization directly to the parameters without such a two-stage procedure. As a result, our algorithm has significantly lower complexity than [1].
>
>
> 2.For [2-6]:
> **Open question in concurrent works:** As discussed in our paper (Line 246), while this operation has been mentioned or utilized in several concurrent works (for instance, row normalization can be viewed as a special case of LMO, as in [6]), these related papers[2-6] propose a class of optimizers from a highly abstract perspective (i.e., a general class of LMO-based optimizers). **However, the community still lacks a concrete understanding of exactly why certain specific norms benefit Transformer optimization.** MUON's spectral norm is the one particular norm that has been extensively verified empirically to work well across most tasks and scales.
>
> **Our core contribution:** What significantly differentiates our work from these concurrent studies is that we start directly from the curvature structure of Transformer weight matrices. Specifically, we verify that MUON's preconditioner is row-block diagonally dominant for Transformers, **which implies that the preconditioners of MUON and RMNP are approximately equivalent.** Furthermore, we observe that this diagonal dominance becomes more pronounced as model size increases(See Figure 12), making RMNP an efficient and highly scalable alternative to MUON for ultra-large-scale training.
>
>
>
> [1]NorMuon: Making Muon more efficient and scalable
>
> [2]A Minimalist Optimizer Design for LLM Pretraining
>
> [3]Training deep learning models with norm-constrained LMOs
>
>
> [4]Adam-mini: Use Fewer Learning Rates To Gain More
>
> [5]Why Transformers Need Adam: A Hessian Perspective
>
> [6]Towards Quantifying the Hessian Structure of Neural Networks

---

> > ### Author Rebuttal · Reviewer_fjwp · 2026-04-01
> >
> > Thank you for the author's reply. I will keep my score. Overall, I think the article has a clear motivation and an effective method.

---

> > > ### Author Response · Authors · 2026-04-01
> > >
> > > We sincerely thank Reviewer fjwp for their time, constructive feedback, and encouraging recognition of our work. If you think we have addressed all you concerns please consider raising your score in the final decision.
> > >
> > > To further strengthen the paper, we will incorporate the experimental evidence and discussions provided in this rebuttal into the final manuscript. Furthermore, we will acknowledge Reviewer in the Acknowledgments section for their insightful discussions. Specifically, we will add:
> > >
> > > 1. Additional baseline comparisons with Shampoo and Soap on LLMs, including a hyperparameter search over 5 learning rates for each algorithm.
> > > 2. Evidence demonstrating the row block diagonal dominance of the Muon preconditioner across diverse architectures (Mamba, CNN).
> > > 3. An expanded discussion section to clearly differentiate our contributions from recent related works.

---

### Official Review · Reviewer_oqBk · 2026-03-12

**Soundness:** 3
**Presentation:** 2
**Significance:** 3
**Originality:** 3
**Overall Recommendation:** 5
**Confidence:** 4

**Summary:**

This paper introduces RMNP (Row Momentum Normalized Preconditioning), an optimizer improving the computational efficiency of matrix-based optimization. Motivated by the empirical observation that Transformer layerwise Hessians exhibit a block-diagonal structure, RMNP simplifies the preconditioning step of the Muon. Specifically, it replaces Muon's Newton-Schulz iteration with a row-wise $\ell_{2}$ normalization. The authors provide non-convex convergence guarantees that match the information-theoretic minimax optimal complexity previously established for Muon. Empirically, RMNP reduces the per-iteration complexity from $\mathcal{O}(mn\cdot min(m,n))$ to $\mathcal{O}(mn)$. Results on GPT-2 and LLaMA demonstrates that it speeds up the preconditioning process while achieving comparable perplexity than Muon.

**Compliance With Llm Reviewing Policy:**

Affirmed.

**Final Justification:**

Good paper. I raise the score to 5.

**Key Questions For Authors:**

1. Can you show results on image tasks like CIFAR-10?
2. Can you provide the loss curve for Figure 5 and 6 experiments? I want to check if the training converges.
3. In Table 3, the memory of RMNP and Muon is the same. This is counter-intuitive. Muon requires 5 steps of Newton-Schulz iterations, which typically requires storing intermediate matrices for matrix multiplication. Row normalization is a simple reduction operation that should theoretically require less peak memory. Can the authors explain this?
4. An existing work [1] looks very relevant and analyzes the row or column normalization. Could you discuss the difference with RMNP?

[1] Glentis, Athanasios, et al. "A minimalist optimizer design for LLM pretraining." _arXiv preprint arXiv:2506.16659_ (2025).

**Limitations:**

yes

**Strengths And Weaknesses:**

**Strengths:**
1. The motivation is clear. This paper is inspired by the row-wise block-diagonal dominance of layer-wise Hessians in Transformers. It's interesting.
2. This method reduces the per-iteration complexity significantly while keeping the competitive performance.
3. The paper defines $r_i$ to quantify the ratio of the diagonal element to the average magnitude of off-diagonal entries. This shows that the diagonal element dominates in most layers.
4. The paper establishes non-convex convergence guarantees.
5. Empirical results on GPT-2 and LLaMA show comparable performance of RMNP with Muon and its significant speedup.


**Weakness:**
1. The experiments only test on language tasks. Can you show results on image tasks like CIFAR-10?
2. (Minor) In Figure 4 and 8, are $r_{avg}$ and $r_{min}$ flipped?

---

> ### Author Rebuttal · Authors · 2026-03-31
>
> We appreciate the reviewer's valuable comments. We provide additional evidence to support our findings and address the reviewer's concerns through the following link: https://anonymous.4open.science/r/RMNP-317C
>
>
>
> **W1 & Q1**
>
>
> We have conducted additional experiments (Figure 18, Table 17) with Resnet 18 on CIFAR10, and the results show that RMNP slightly underperforms Muon (<0.4%), it delivers comparable performance while maintaining low computational complexity.
>
>
> **W2**
>
> We thank the reviewer for pointing this out. We have fixed this issue, and it will not appear in the final version of the manuscript.
>
> **Q2**
>
> We have provided the corresponding curves in Figure 19 and Figure 20.
>
>
> **Q3**
> Equal peak memory is expected, for two reasons:
>
> **1. NS intermediates are tiny and transient.** Newton–Schulz creates temporary matrices inside `optimizer.step()` that are freed immediately rather than stored as optimizer state. For GPT-2 Medium’s largest 4096×1024 weight, the largest such matrix is only about **2 MB** in bf16; across all layers, NS intermediates total only $\sim$ **26 MB**, far smaller than the shared `updates_flat` buffer ($\sim$ **700 MB**). With `torch.compile`, some are fused away entirely.
>
> **2. Peak memory is dominated by activations, not optimizer internals.** The GPU high-water mark is dominated by forward/backward activations ($\sim$ **10–20 GB** for GPT-2 Medium), which are identical for Muon and RMNP. PyTorch’s caching allocator reuses memory for the small transient NS buffers, so they typically do not increase `max_memory_allocated`. The persistent optimizer states are also structurally the same.
>
> So the difference between RMNP and Muon is in **compute**, not **memory**.
>
>
> **Q4**
>
> We thank the reviewer oqBk and reviewer fjwp for pointing out concurrent works [1-2]. We include some discussion in our paper (Line 246) and We will incorporate more discussion on concurrent works in the revised draft. We like to provide a unified clarification here first:
>
> **Open question in concurrent works:** As discussed in our paper (Line 246), while this operation has been mentioned or utilized in several concurrent works (for instance, row normalization can be viewed as a special case of LMO, as in [1]), these related papers[1-2] propose a class of optimizers from a highly abstract perspective (i.e., a general class of LMO-based optimizers). **However, the community still lacks a concrete understanding of exactly why certain specific norms benefit Transformer optimization.** MUON's spectral norm is the one particular norm that has been extensively verified empirically to work well across most tasks and scales.
>
> **Our core contribution:** What significantly differentiates our work from these concurrent studies is that we start directly from the curvature structure of Transformer weight matrices（related work[3-5]) to understand why specific norm work. Specifically, we verify that MUON's preconditioner is row-block diagonally dominant for Transformers, **which implies that the preconditioners of MUON and RMNP are approximately equivalent.** Furthermore, we observe that this diagonal dominance becomes more pronounced as model size increases(See Figure 12), making RMNP an efficient and highly scalable alternative to MUON for ultra-large-scale training.
>
> [1]A Minimalist Optimizer Design for LLM Pretraining
>
> [2]Training deep learning models with norm-constrained LMOs
>
> [3]Adam-mini: Use Fewer Learning Rates To Gain More
>
> [4]Why Transformers Need Adam: A Hessian Perspective
>
> [5]Towards Quantifying the Hessian Structure of Neural Networks

---

> > ### Author Rebuttal · Reviewer_oqBk · 2026-04-04
> >
> > Thanks for the authors’ detailed reply. My concerns are fully addressed.

---

> > > ### Author Response · Authors · 2026-04-05
> > >
> > > We sincerely thank Reviewer oqBk for their time, constructive feedback, and encouraging recognition of our work. **If you think we have addressed all you concerns please consider raising your score in the final decision.**
> > >
> > > To further strengthen the paper, we will incorporate the experimental evidence and discussions provided in this rebuttal into the final manuscript. Furthermore, we will acknowledge Reviewer in the Acknowledgments section for their insightful discussions. Specifically, we will add:
> > >
> > >
> > > 1.Report on the performance of RMNP in CV task.
> > >
> > > 2.Clarification on Memory Usage
> > >
> > > 3.Corrections to figures and other minor errors
> > >
> > > 4.An expanded discussion section to clearly differentiate our contributions from recent related works.

---

### Official Review · Reviewer_wDnB · 2026-03-13

**Soundness:** 3
**Presentation:** 4
**Significance:** 3
**Originality:** 3
**Overall Recommendation:** 5
**Confidence:** 4

**Summary:**

This paper proposes a very efficient preconditioner for optimizers, replacing the Muon msign with a simple row-wise L2 norm. The idea is motivated by empirical observations that Transformer layer-wise Hessians exhibit strong row-wise block diagonal dominance. This method holds up well in performance compared to more traditional optimizers like AdamW and Muon, optimizing models efficiently while saving both computing resources and training time.

**Compliance With Llm Reviewing Policy:**

Affirmed.

**Key Questions For Authors:**

See weaknesses.

**Limitations:**

yes

**Strengths And Weaknesses:**

**Strengths**
1. This paper studies an important problem and provides an interesting solution.
2. The paper is well-written with clear motivation and good presentation flow.
3. Extensive experiments are conducted to justify RMNP's effectiveness.

**Weaknesses**
1. The stability of the optimizer over long training durations has not been thoroughly evaluated. Conducting longer training runs would provide greater confidence in its reliability.
2. The paper relies heavily on the hypothesis regarding the Transformer Hessian's row-wise block diagonal dominance property and uses a unified learning rate scaler for different modules. Given that the scaling factor $\max(1, \sqrt{m/n})$ depends heavily on matrix size, the partitioning strategy for different layers (such as attention layers vs. MLP layers) is critical. It remains unclear what impact alternative splitting methods (e.g., splitting attention QKV projections separately) would have on the results.
3. How does RMNP perform when applied to the LM head or embedding layers? These modules are often row/column-wise independence. It would be helpful to discuss whether RMNP's assumptions remain applicable  for such special cases.

---

> ### Author Rebuttal · Authors · 2026-03-31
>
> We appreciate the reviewer's valuable comments. We provide additional evidence to support our findings and address the reviewer's concerns through the following link: https://anonymous.4open.science/r/RMNP-317C
>
> **W1:**
> We conducted longer training runs on GPT-2/Llama, the results are in Figure 10, Table 12. As shown in our reported experiments, our algorithm achieves stable results over extended training durations.
>
> **W2**
>
>
> A. Regarding the concern about the scaling coefficient, the experiments in our original paper were primarily designed to maintain consistency with MUON's setup. We have conducted additional ablation experiments with and without QKV parameter stacking, using the scaling factor $\max(1, \sqrt{m/n})$. The results shows that the difference is very small.
>
> B.Regarding the concern about assumption dependency, we have supplemented additional experiments to support our claims, extending beyond the Transformer architecture (though we have also included more evidence across different scales of Transformers), to CNN and Mamba architectures.The results are in Figture 12，13，14，15. The experimental results support the following:
> 1. Across Transformers of different scales, MUON's preconditioner consistently exhibits the row block diagonal dominance property. Moreover, we additionally find that this dominance becomes more pronounced as the model scale increases.
> 2. For weight matrices in CNN and Mamba architectures, we also observe the row block diagonal dominance phenomenon in their MUON preconditioners.
>
> **W3**
>
> We thank the reviewer for raising a very interesting question. We have conducted additional ablation experiments(Figture 16,17;Table 13,14,15,16) that include the LM head and embedding layers. The conclusions are as follows:
>
> 1.When training with Muon that includes the LM Head and Embedding, the network's overall diagonal dominance is preserved. Additionally, the Muon preconditioners for both the LM Head and Embedding matrices continue to demonstrate diagonal dominance.
>
> 2.Including LM Head and Embedding in Muon and RMNP training yields no significant performance changes, with variations remaining within a negligible error margin.

---

> > ### Author Rebuttal · Reviewer_wDnB · 2026-04-02
> >
> > Thank you to the authors for their additional clarifications which were helpful. Given the high quality of both the work and the response, I will maintain my initial score.

---

> > > ### Author Response · Authors · 2026-04-02
> > >
> > > We sincerely thank Reviewer wDnB  for their time, constructive feedback, and encouraging recognition of our work.
> > >
> > > To further strengthen the paper, we will incorporate the experimental evidence and discussions provided in this rebuttal into the final manuscript. Furthermore, we will acknowledge Reviewer in the Acknowledgments section for their insightful discussions. Specifically, we will add:
> > >
> > > 1.Extended training results on GPT2 and LLaMA (2× standard steps), confirming that RMNP achieves stable performance over longer training durations.
> > >
> > > 2.Ablation experiments on QKV parameter stacking with the scaling factor max(1, √(m/n)), showing that the performance difference between stacked and split configurations is negligible.
> > >
> > > 3.Broader empirical evidence that the row block diagonal dominance property of the Muon preconditioner holds consistently across different scale of Transformer and different architectures(CNN and Mamba).

---

### Decision · Program_Chairs · 2026-04-30

**Decision:**

Accept (regular)

**Comment:**

# Summary

This paper proposes Row Momentum Normalized Preconditioning (RMNP), which is a matrix-based optimizer that applies row-wise normalization to momentum. RMNP can be viewed as a diagonal-preconditioner approximation to Muon, which is motivated by the observation that the Transformer layerwise Hessians have diagonal structure. RMNP runs much faster than Muon and achieves generalization performance comparable to Muon. Convergence theorems on nonconvex smooth functions are also provided.

# Comments

The paper is well-written and easy to follow. All reviewers gave positive assessments of the paper, and the authors addressed the concerns raised by the reviewers well. This paper is a solid contribution to the literature on matrix-based optimizers, and I would like to recommend acceptance.

Although I recommend acceptance, one slight remaining concern is the novelty and the positioning of the paper relative to many existing and concurrent works studying row-norm/column-norm-based optimizers. It is essential that the paper offers a thorough coverage of these existing results and contextualizes the key contributions relative to them. The reviewers brought up some important references, and I believe the paper should also cite [1-4]. Also, the paper should discuss why their theoretical analyses are not simply subsumed by existing papers on more general LMOs, such as Pethick et al. (2025).

[1] SRON: State-free LLM Training via Row-wise Gradient Normalization

[2] A Minimalist Optimizer Design for LLM Pretraining

[3] SWAN: SGD with Normalization and Whitening Enables Stateless LLM Training

[4] Gradient Multi-Normalization for Efficient LLM Training

Some other comments that I have:
* Algorithm 1, Line 5 is not exactly orthogonalization. NS5 is just an approximation to it.
* If Eq. (3) is intended to represent row-wise normalization, then the current expression does not seem correct. In general, $\operatorname{diag}\left((V_tV_t^\top)^{1/2}\right) \neq \operatorname{diag}(V_tV_t^\top)^{1/2}$. The former takes the diagonal after applying the matrix square root, while the latter corresponds to taking the square root of the row-wise squared norms. Therefore, Eq. (3) should be corrected to $H_{\mathrm{RMNP}} = \operatorname{diag}(V_tV_t^\top)^{1/2} \otimes I_n$.
* This applies to most papers studying matrix-valued optimizers, but it is really important to specify whether you are using the $d_{out} \times d_{in}$ or $d_{in} \times d_{out}$ convention in your neural network experiments, because the convention may differ depending on the implementation style.